# The Effect of Attention Head Count on Transformer Approximation

**Penghao Yu**
Department of Mathematics
National University of Singapore
`penghaoyu@u.nus.edu`

**Haotian Jiang**
Institute for Functional Intelligent Materials
National University of Singapore
`haotian@nus.edu.sg`

**Zeyu Bao**
Department of Mathematics
National University of Singapore
`zeyu@u.nus.edu`

**Ruoxi Yu**
Center for Data Science
Peking University
`yuruoxi@stu.pku.edu.cn`

**Qianxiao Li**
Department of Mathematics
Institute for Functional Intelligent Materials
National University of Singapore
`qianxiao@nus.edu.sg`

## Abstract

Transformer has become the dominant architecture for sequence modeling, yet a detailed understanding of how its structural parameters influence expressive power remains limited. In this work, we study the approximation properties of transformers, with particular emphasis on the role of the number of attention heads. Our analysis begins with the introduction of a generalized $D$-retrieval task, which we prove to be dense in the space of continuous functions, thereby providing the basis for our theoretical framework. We then establish both upper and lower bounds on the parameter complexity required for $\epsilon$-approximation. Specifically, we show that transformers with sufficiently many heads admit efficient approximation, whereas with too few heads, the number of parameters must scale at least as $O(1/\epsilon^{cT})$, for some constant $c$ and sequence length $T$. To the best of our knowledge, this constitutes the first rigorous lower bound of this type in a nonlinear and practically relevant setting. We further examine the single-head case and demonstrate that an embedding dimension of order $O(T)$ allows complete memorization of the input, where approximation is entirely achieved by the feed-forward block. Finally, we validate our theoretical findings with experiments on both synthetic data and real-world tasks, illustrating the practical relevance of our results.

## 1 Introduction

The transformer architecture (Vaswani et al., 2017) has become the foundation of modern sequence modeling, driving progress in natural language processing (Devlin et al., 2019; Brown et al., 2020), computer vision (Dosovitskiy et al., 2021), and multi-modal learning (Radford et al., 2021). Its ability to scale has enabled breakthroughs such as BERT, GPT, and ViT, making it the dominant paradigm across domains. Despite this remarkable empirical success, the theoretical principles underlying transformer expressivity remain incomplete. In particular, while universal approximation results establish that transformers can approximate arbitrary sequence-to-sequence mappings (Yun et al., 2020a; Pérez et al., 2021), much less is known about how their structural hyperparameters influence approximation efficiency.

Among transformer hyperparameters, the number of attention heads plays a central role. In practice, large models often adopt head counts such as 32, 64, or 128 (e.g., Devlin et al. (2019); Dosovitskiy et al. (2021); Touvron et al. (2023); Jiang et al. (2023); Grattafiori et al. (2024) see Table 10 for

more ), yet this choice is largely heuristic: there is no principled understanding of how many heads are needed for a given task, nor of the costs incurred when the head count is insufficient. Theoretical progress on this question has so far been limited. Most existing results focus on upper bounds, showing that transformers with sufficiently many heads or with extremely large embedding dimension in the single-head case can achieve universal approximation or good approximation rate, but offering little insight into the limitations that arise when the head count is insufficient. Moreover, many analyses rely on strong simplifications—such as restricting to linear embeddings, isolating the attention block, or linearizing the architecture. While these assumptions make the problem more tractable, they severely restrict the model's expressive power and prevent the derivation of rigorous lower bounds in realistic nonlinear settings.

In this work, we address this gap by analyzing single-layer transformers on sequence-to-vector tasks. To this end, we introduce a new function class, the *generalized D-retrieval tasks*, which we design as a structured but expressive family motivated by retrieval problems. Each coordinate is defined by $\bar{z}_i(X_T) = \min_{t \in S_i} f_i(x(t)), i = 1, \ldots, D$ for subsets $S_i \subseteq [T]$, and the overall target takes the form $H(X_T) = F_0(\bar{z}_1(X_T), \ldots, \bar{z}_D(X_T))$. By construction, this class extends retrieval-style problems while being dense in the space of continuous sequence-to-vector mappings, ensuring that results obtained in this setting reflect general approximation behavior.

The central challenge arises when the number of heads $h$ is smaller than the intrinsic dimension $D$ of the target. In this case, multiple coordinates $z_i(X_T)$ must be represented by the same head, creating an information bottleneck: the attention layer maps distinct sequences to nearly indistinguishable representations, forcing the feed-forward network to perform the separation. We show that overcoming this bottleneck requires parameter growth exponential in the sequence length $T$, namely $O(1/\epsilon^{cT})$ parameters for $\epsilon$-accuracy, thus establishing the first rigorous lower bounds for transformers in nonlinear settings. In contrast, when $h \geq D$, heads can specialize to distinct coordinates $z_i$, eliminating the bottleneck and enabling efficient approximation.

Our results advance the theoretical understanding of attention by showing, that insufficient head count provably limits expressivity in realistic regimes. Experiments on both synthetic tasks and real-world retrieval data confirm that the predicted scaling laws persist in practice.

**Contributions.** Our main contributions are as follows:

First, we establish the first rigorous lower bounds for transformers in nonlinear settings, showing that when $h < D$, parameter complexity grows exponentially with sequence length.

Second, we provide constructive upper bounds, proving that $h \geq D$ enables efficient approximation with parameter growth independent of sequence length $T$.

Third, in the memorization regime, single-head transformers with embedding dimension $n \geq Td$ approximate by memorizing sequences, with the complexity residing in the feed-forward block.

## 2 RELATED WORK

Several works have studied the approximation and expressivity properties of transformers. The universal approximation property was first established in Yun et al. (2020a), and later extended to transformers with sparse attention matrices in Yun et al. (2020b). The approximation rate of single-layer transformers with one head was analyzed in Jiang & Li (2024). Amsel et al. (2024) investigated how the rank of the attention matrix influences expressivity for a specific nearest-neighbor target can be constructed. They showed that when the rank is insufficient, the number of heads required for approximation grows exponentially, independent of sequence length. In a related direction, Bhojanapalli et al. (2020) argued that setting the rank of the attention matrix equal to the sequence length enhances expressivity. Beyond finite-dimensional settings, Takakura & Suzuki (2023) considered sequences of infinite dimension, characterizing approximation rates in terms of target function smoothness. Similarly, Wang et al. (2024) studied special classes of target functions and demonstrated that approximation error scales polynomially with the number of heads. In addition to these approximation-theoretic results, several works have investigated broader notions of expressivity. Dehghani et al. (2019); Pérez et al. (2021) established the Turing completeness of transformers, and Giannou et al. (2023) showed that transformers can represent arbitrary computer programs in

a looped setting. Finally, Mahdavi et al. (2023) examined memorization capacity, proving that the number of samples that can be stored scales linearly with the number of heads.

## 3 PRELIMINARIES

**Input and Output** We consider the input space

$$\mathcal{X}_T = \big\{ \, x(s) \in [0,1]^d \; : \; s \in [T] \, \big\}, \tag{1}$$

where $[T] = \{1, \ldots, T\}$. We call $T$ the length of the input sequence. The output is a single vector $y \in \mathbb{R}^l$, where $l$ is independent of $T$ and specified by the task.

For example, in a text retrieval task one may take $d$ to be the max number of tokens per candidate, $T$ the number of candidates, and $l$ the size of the output representation.

**Input Representation.** Each token is mapped to an $E$-dimensional vector by a trainable encoder

$$P_\phi : [0,1]^d \times [T] \to \mathbb{R}^E, \qquad (\boldsymbol{x}, s) \mapsto P_\phi(\boldsymbol{x}, s),$$

which jointly incorporates the content $\boldsymbol{x}$ and its position $s$. Given $X_T = \{x(s)\}_{s=1}^T \in \mathcal{X}_T$, the embedded sequence is

$$\hat{X}_T = \{ \, \hat{x}(s) = P_\phi(x(s), s) \in \mathbb{R}^E : s \in [T] \, \}. \tag{2}$$

This formulation subsumes common designs where $P_\phi$ combines a content embedding with either learned or deterministic positional encoding.

For example, if $\mathrm{Emb}(x)$ is a content embedding map and $p(t)$ a positional code, then common schemes correspond to

$$P_\phi(x(t), t) = \mathrm{Emb}(x(t)) + p(t) \quad \text{(additive PE)},$$

or

$$P_\phi(x(t), t) = \big(\mathrm{Emb}(x(t)), \, p(t)\big) \quad \text{(concatenated PE)}.$$

Following common practice, we append a trainable *classification token* $\hat{c}_0 \in \mathbb{R}^E$ to the sequence. The final input to the transformer is

$$\hat{X}[T] = \{\hat{x}(1), \ldots, \hat{x}(T), \hat{c}_0\} \in \mathbb{R}^{E \times (T+1)}, \tag{3}$$

and the output $\hat{y}$ is taken from the $(T{+}1)$-th position corresponding to $\hat{c}_0$.

**Transformer Hypothesis Class** With the input space and embedding defined, we then formulate the transformer hypothesis space.

We consider a single-layer transformer based on the standard architecture (Vaswani et al., 2017), with two modifications introduced for analytical simplicity. Firstly, we omit layer normalization to simplify the analysis, while acknowledging its practical importance, and we conjecture that our key lower bound (Theorem 2 (2)) still holds when layer normalization is present. Secondly, we exclude residual connections outside the feed-forward network. In the single-layer sequence-to-vector setting, where the output is read from a designated classification token, the residual branch can be merged into the feed-forward transformation by reparameterization, thus these likewise do not alter the expressive power of the architecture.

For an $h$-head, single-layer transformer, let $n$ denote the embedding dimension *per head* and $E = nh$ the total embedding dimension. The output is

$$\hat{y} = \hat{H}(\hat{X}[T]) = \hat{F}\Big(\hat{c}_0 + W_O \, \mathrm{Concat}_{i=1}^h \Big(\sum_{t=1}^T \sigma\big[(W_{Q,i}\hat{c}_0)^\top W_{K,i}\hat{x}(t)\big] \, W_{V,i}\hat{x}(t)\Big)\Big), \tag{4}$$

where for each head $i$, $W_{Q,i}, W_{K,i} \in \mathbb{R}^{n \times E}$ are the query/key projection matrices, $W_{V,i} \in \mathbb{R}^{n \times E}$ is the value projection, $W_O \in \mathbb{R}^{E \times E}$ is the output projection applied to the concatenated heads, and

$\hat{F} : \mathbb{R}^E \to \mathbb{R}^l$ is a feed-forward network which we call it the *feed-forward block*. The softmax with scaling factor $\beta$ is defined by

$$\sigma[\rho](t) = \frac{\exp\big(\beta\,\rho(t)\big)}{\sum_{t'=1}^{T} \exp\big(\beta\,\rho(t')\big)}, \qquad \beta > 0. \tag{5}$$

and $\beta > 0$ may be chosen arbitrarily large in order to make the softmax attention mechanism approximate a hardmax.

We denote this family by

$$\mathcal{H}(h, n, d, T, M), \tag{6}$$

the class of single-layer transformers with $h$ heads, per-head embedding dimension $n$, input dimension $d$, sequence length $T$, and parameter count $M$. Each $H \in \mathcal{H}(h, n, d, T, M)$ is a mapping

$$H : \mathbb{R}^{d \times T} \to \mathbb{R}^l,$$

implemented by the encoder $P_\phi : [0,1]^d \times [T] \to \mathbb{R}^{nh}$, concatenation of the classification token $\hat{c}_0$, a multi-head attention layer with projections $\{W_{Q,i}, W_{K,i}, W_{V,i}\}_{i=1}^{h}, W_O$, and a feed-forward network $\hat{F} : \mathbb{R}^{nh} \to \mathbb{R}^l$. Thus $H$ has the form equation 4, with parameter count $k$ referring to the weights and biases in FFNs $(P_\phi, \hat{F})$.

**Approximation Problem**    With the hypothesis class specified, we now formalize the approximation problem, which provides the framework for analyzing the expressive power of transformers.

**Definition 1** ($\epsilon$-approximation)**.** *Let $\mathcal{X}_T \subset \mathbb{R}^{d \times T}$ be a compact domain, and let $F : \mathcal{X}_T \to \mathbb{R}^l$ be a target function. We say that the hypothesis class $\mathcal{H}(h, n, d, T, M)$ $\epsilon$-approximates $F$ on $\mathcal{X}_T$ if there exists $\hat{H} \in \mathcal{H}(h, n, d, T, M)$ such that*

$$\sup_{X_T \in \mathcal{X}_T} \|\hat{H}(X_T) - F(X_T)\|_\infty < \epsilon.$$

# 4    GENERALIZED $D$-RETRIEVAL TASKS

**Target functions.**    To motivate our construction, consider a simple one-dimensional example:
$$\mathcal{X}_T = \{\, X_T = (x(1), \dots, x(T)) : x(t) \in [0,1] \,\},$$
with target

$$H(X_T) = \max_{1 \le t \le T} x(t) + \min_{1 \le t \le T} x(t). \tag{7}$$

This task requires the model to extract two distinct features from the sequence—the maximum and the minimum—before combining them. It can thus be viewed as a retrieval problem with two independent features being aggregated.

This example illustrates the broader idea behind our target class: retrieval-style problems where multiple salient features must be identified and combined. We now formalize this intuition by defining the family of *generalized $D$-retrieval tasks*.

**Mathematical Formulation**    Formally, for each $i = 1, \dots, D$, let $f_i : [0,1]^d \to [0,1]$ be $C^2$ and define

$$\bar{z}_i(X_T) = \min_{t \in S_i} f_i(x(t)), \qquad S_i \subseteq [T], \;\; |S_i| \ge \frac{1}{4}T, \tag{8}$$

so that $\bar{z}(X_T) = (\bar{z}_1(X_T), \dots, \bar{z}_D(X_T)) \in [0,1]^D$. The target is then

$$H(X_T) = F_0\big(\bar{z}(X_T)\big), \tag{9}$$

where $F_0 : [0,1]^D \to \mathbb{R}$ is $C^1$. For vector-valued targets $H : [0,1]^{d \times T} \to \mathbb{R}^l$ defined with the same functions $f_i$, subsets $S_i$, and an outer map $F_0 : [0,1]^D \to \mathbb{R}^l$, the extension is applied coordinate-wise, since each coordinate function of $F_0$ can be approximated independently. Therefore, it suffices to consider the scalar-valued case. We denote by $\mathcal{F}_D^{d,T}$ the class of all such functions $H$.

Related sparse sequence-to-sequence retrieval tasks, such as the $q$-sparse token regression (qSTR) model of (Mousavi-Hosseini et al., 2025), where each output position depends on at most $q$ relevant input tokens, can be viewed as sequence-to-sequence analogues of our formulation. Their results on the sample complexity of single-layer Transformers with at least $q$ heads are complementary to our approximation-theoretic guarantees in the generalized $D$-retrieval setting.

**Assumptions on the target class** For the theoretical analysis to be tractable we impose the following conditions:

**Assumption 1** (Model constraints). The model constraints are as follows:

(1.1) The embedding $P_\phi$ satisfies

$$\|P_\phi(x(s), s)\|_2 \leq 1, \qquad \forall\, s \in [T],\; X_T \in \mathcal{X}_T,$$

ensuring embedded inputs remain uniformly bounded.

(1.2) The post-attention mapping $\hat{F}$ is a two-layer feed-forward network with 1-Lipschitz activation, hence a universal approximator on compact domains.

(1.3) All weights in $\hat{F}$ and entries of the attention matrices $\{W_{Q,i}, W_{K,i}, W_{V,i}\}, W_O$ are bounded in magnitude by 1, ensuring stability of the model.

**Assumption 2** (Target class constraint). The target functions defined in equation 9 satisfy the following:

(2.1) Each $f_i : [0,1]^d \to [0,1]$ attains its unique global minimum $z_i$ at a point $x^{(i)} \in [0,1]^d$.

(2.2) The minimizers $\{x^{(i)}\}_{i=1}^D$ are pairwise distinct.

(2.3) The Hessian $\nabla_x^2 f_i(x^{(i)})$ is positive definite for all $i = 1, \ldots, D$.

(2.4) The gradient $\nabla_z F_0(z_1, \ldots, z_D)$ has all coordinates strictly nonzero.

*Remark.* Assumption 2 excludes only degenerate cases while preserving broad generality for both the functions $f_i$ and the outer map $F_0$. More specifically: Assumptions (2.1) and (2.4) ensure that each $f_i$ behaves regularly around its minimizer. A degenerate example excluded by these assumptions is $f_i(x) \equiv c_0$ for constant $c_0$, which is totally independent of the input; Assumption (2.2) requires distinct minimizers, which allows partitioning the space into $D$ disjoint basins around each minimizer $x^{(i)}$. A degenerate example excluded by this assumption is $f_1 = f_2 = \cdots = f_D$; Assumption (2.3) enforces sensitivity of the target to small perturbations near the minimizers, ruling out trivial flat cases (such as $F_0 \equiv c_0$ for constant $c_0$) where no meaningful separation can be made.

Having introduced the generalized $D$-retrieval tasks, it remains to ask whether this family is sufficiently expressive. To address this, we now establish the *universality of the target class*: the family is dense in the space of continuous sequence-to-vector mappings.

**Theorem 1** (Density of the target class). *For fixed $d, T$, the family $\{\mathcal{F}_D\}_{D=1}^\infty$ is dense in $C(\mathcal{X}_T)$. That is, for every $F \in C(\mathcal{X}_T)$ and every $\epsilon > 0$, there exists $D$ and $f \in \mathcal{F}_D$ such that*

$$\max_{X \in \mathcal{X}_T} |F(X) - f(X)| \leq \epsilon.$$

The proof is deferred to Appendix A.1. This density property highlights that our specially designed target family is not overly restrictive; rather, it forms a sufficiently general class to capture arbitrary continuous sequence-to-vector mappings. Beyond density, we show that $D$ is indeed the *intrinsic dimension* of this target, which means that it is the unique $D \ll T$ for which the target $H$ can be represented in the generalized $D$-retrieval task form.

**Corollary 1** (Uniqueness of intrinsic dimension). *If task $H$ can be represented by $(\{f_i, S_i\}_{i=1}^{D_1}, F_0)$ and $(\{\tilde{f}_i, \tilde{S}_i\}_{i=1}^{D_2}, \tilde{F}_0)$, satisfying Assumption 1 and 2 with $D_1^2 + D_2^2 \leq \frac{1}{50}T$, then $D_1 = D_2$.*

This corollary justifies that $D$ comes from the intrinsic property of the target and is invariant across its form of representation. The proof is deferred to Appendix A.2

## 5 APPROXIMATION RATE OF GENERALIZED $D$-RETRIEVAL TASKS

Theorem 1 establishes that the generalized $D$-retrieval tasks form a dense family in the space of continuous sequence-to-vector functions. The next step is to analyze the efficiency with which transformers approximate these functions. To this end, we begin by stating two standard approximation assumptions regarding how well the fundamental building blocks of the target can be approximated.

**Assumption 3** (Approximation of components). We assume the following approximation properties hold.

(A1) There exist constants $C_1 > 0$ and $\gamma > 0$ such that for every $\delta > 0$, the function $F_0 : [0,1]^D \to \mathbb{R}$ can be $\delta$-approximated by a two-layer feed-forward network $\Phi_\delta$ of width at most $C_1/\delta^\gamma$, i.e.,

$$\sup_{x \in [0,1]^D} |F_0(x) - \Phi_\delta(x)| \leq \delta.$$

(A2) There exist constants $C_2 > 0$ and $\gamma > 0$ (possibly different from (A1)) such that for each $i = 1, \ldots, D$ and every $\delta > 0$, the function $f_i : [0,1]^d \to [0,1]$ can be $\delta$-approximated by a two-layer feed-forward network $\Psi_{i,\delta}$ of width at most $C_2/\delta^\gamma$, i.e.,

$$\sup_{x \in [0,1]^d} |f_i(x) - \Psi_{i,\delta}(x)| \leq \delta.$$

These assumptions are reasonable: by the classical result of (Cybenko, 1989), two-layer networks can approximate continuous functions on compact domains. In particular, if the Barron norm is finite, one may take $\gamma = 2$ (Barron, 1993); even in the worst case, setting $\gamma = \max(d, D)$ yields approximation rates comparable to uniform grid discretizations, which still suffices for our analysis.

We now present our main theoretical result. It establishes upper and lower bounds on the approximation rates of transformers within the generalized $D$-retrieval framework. In particular, the lower bound in part (2) provides the first rigorous evidence that insufficient head count $h < D$ leads to exponential parameter complexity, revealing a fundamental expressivity bottleneck.

**Theorem 2** (Approximation rates of transformers). *Fix $d, T$. Under Assumption 3, the following hold for any target $H \in \mathcal{F}_D^{d,T}$:*

> (1) **Sufficient expressivity with $D$ heads.** *For $h = D$ and embedding dimension $n = 2$ per head, there exists a constant $C_{d,D,T} > 0$ such that $\forall M > \frac{C_{d,D,T}}{\epsilon^\gamma}$. the hypothesis class $\mathcal{H}(h, n, d, T, M)$ $\epsilon$-approximates $H$.*
>
> (2) **Lower bound with $s < D$ heads.** *For $h = s < D$, define*
> $$k = \frac{(\frac{1}{4}T - s - D + 1)}{(n+1)s + 1} - 1,$$
> *then*
> $$\min\big\{ M \;:\; \mathcal{H}(h, n, d, T, M) \text{ $\epsilon$-approximates } H \big\} \;=\; \Omega\big(\tfrac{1}{\epsilon^k}\big).$$
>
> (3) **Single-head large embedding dimension.** *For $h = 1$ and per-head embedding dimension $n \geq Td$, if the feed-forward block is a 5-layer ReLU neural network, then there exists a constant $C_{d,D,T} > 0$ such that for all $M > \frac{C_{d,D,T}}{\epsilon^{1+\gamma}}$, the hypothesis class $\mathcal{H}(h, n, d, T, M)$ can $\epsilon$-approximate $H$.*

*Remark.* We clarify the precise role of the assumptions and constants in Theorem 2.
(1)Theorems 2 (1) and 2 (3) require only Assumption 3, while Theorem 2 (2) relies only on Assumptions 1 and 2.
(2) The constant in Theorem 2 can be made explicit as $C_{d,D,T} = C_{d,D}(rT)^{-\alpha_{d,D}T}$, where $r > 0$ is determined by the local behavior of the functions $f_i$ around $x^{(i)}$ and of $F_0$, and $\alpha_{d,D}$ depends only on $d$ and $D$. This form is valid in the regime $d, D \ll T \ll 1/\epsilon$.
(3) The exponent coefficients in Theorems 2 (1) and 2 (3) differ because, in Theorem 2 (3), the network $\hat{F}$ also needs $\Omega(T/\epsilon)$ parameters to approximate the "max-like" operation. This yields a bound of the form $M \leq \frac{1}{\epsilon^{\max(1,\gamma)}}$, and for notational simplicity we write $M \leq 1/\epsilon^{1+\gamma}$.
We provide the detailed proof in Appendix A.2. We also justify the tightness of Theorem 2 (2) in Appendix A.2.2.

Theorem 2 highlights how approximation efficiency depends on head count: enough heads allow specialization, too few force inefficient compression, and a single large head can rely on memorization. To illustrate these cases concretely, we now revisit the toy example from equation 7 and discuss how each part of the theorem works in that setting.

**Case (1): $h \geq D$ heads.** Theorem 2 (1) shows that when the number of heads matches the intrinsic dimension $D$ of the target, the transformer can allocate one head per component feature, allowing each head to specialize and leaving the feed-forward block to aggregate their outputs. This yields efficient approximation with $O(M^{-1/\gamma})$ error for parameter count $M$, independent of sequence length $T$.

In the toy example with $D = 2$, one head naturally tracks the maximum and the other the minimum, so the task is solved directly without incurring inefficiency. This illustrates how having "enough heads" removes the unfavorable scaling in $T$ and explains the practical advantage of multiple heads beyond universal approximation results (e.g., Kajitsuka & Sato (2023)).

**Case (2):** $h < D$ **heads.** Theorem 2 (2) establishes that when the number of heads is smaller than the intrinsic dimension $D$, the parameter count required to achieve a given accuracy can grow exponentially in the sequence length $T$. This lower bound highlights why insufficient heads lead to severe inefficiency. Intuitively, each head can be viewed as specializing in one coordinate of the minima structure in equation 9. When $h < D$, a single head must encode multiple roles simultaneously.

In the toy example with $D = 2$, one head is forced to capture both the maximum and the minimum across all $T$ positions. Since softmax attention only produces weighted averages, the head must effectively encode information from multiple sequence elements simultaneously. As $T$ increases, the number of relevant elements to distinguish grows linearly with $T$, yet they are compressed into an $n$-dimensional vector whose size does not scale with $T$. The feed-forward block must then disentangle these increasingly entangled representations, which requires parameters exponential in $T$. This explains why the parameter requirement scales as $\Omega(1/\epsilon^{cT})$ and why the scaling improves dramatically once $h \geq D$.

Theorem 2 (2) is proved with the following idea: (i) each head selects its most responsive locations $(y_j, t_j)$ in $D$ disjoint minima basins around $x^{(i)}$; (ii) because $s < D$, there is at least one segment $G_i \subset B(x^{(i)}, r)$ that no head focuses on. We then consider the segment $G_i$ in it; (iii) along this segment (suppose it is $G_1$), we construct many candidate subsequences and, by a pigeonhole argument, obtain two subsequences $Z_1, Z_2$ whose post-attention representations are almost identical but whose contribution to $f_1$ differs; (iv) these subsequences are then embedded into full sequences $W_1, W_2$, which the target function separates by at least $3\epsilon$, while the attention block maps them within $O(\epsilon^{k+1})$, forcing a large feed-forward network.

The intuition by which we derive the large network is different from geometric arguments in existing works such as (Yehudai & Shamir, 2019). We directly made use of the fact that the network must be either large or have large parameters to approximate a function with large Lipschitz norm.

**Case (3): single head with large embedding.** Theorem 2 (3) shows that when the embedding dimension scales with the sequence length, $E = n \geq Td$, the model can encode the entire sequence into the classification token $\hat{c}_0$. Concretely, each input can be embedded as $e_t \otimes x(t) \in \mathbb{R}^{Td}$, where $e_t$ is the $t$-th standard basis vector, so that trivial attention aggregates to

$$\frac{1}{T}\left(x(1), \ldots, x(T)\right) \in [0, 1]^T,$$

which preserves the full sequence. The feed-forward block $\hat{F}$ can then recover the target relation efficiently. Unlike memorization of training data (Mahdavi et al., 2023), this mechanism can generalize since it captures the relation itself. Moreover, approximating extrema functions such as $\max$ and $\min$ with a shallow ReLU network is straightforward (see Lemma 9), requiring width $O(T/\epsilon)$ for $\epsilon$ accuracy. However, this regime is impractical, as it demands embedding dimensions that grow linearly with $T$.

As deeper transformer are more commonly used in practice, here we conjecture the extension of Theorem 2 (2) onto the $L$-layer case.

**Conjecture 1** (Multilayer transformer case). A necessary condition for efficient approximation is $L \cdot h \geq D$. When the head number is insufficient across layers, we conjecture the lower-bound scaling for some constants $a_L, b_L > 0$ depending only on depth $L$ to be

$$\log(\text{ParamCount}) = \Omega\Big(|\log \epsilon| \cdot \frac{a_L T^{b_L}}{nh}\Big),$$

Experiments on the synthetic dataset in Section 6.1 with 2-layer transformer with no positional encoding and no layer norm has also verified this transition at $D = L \cdot h$. (See Table 5 in Appendix)

## 6 EXPERIMENTS

Theorem 2 provides theoretical insights into how the approximation ability of transformers depends on the number of heads. In this section, we illustrate these insights empirically. We begin with synthetic tasks that mirror the structure of the generalized $D$-retrieval tasks, and then turn to real datasets (MS MARCO and CIFAR-10) to examine whether similar scaling behaviors arise in practice.

## 6.1 NUMERICAL VERIFICATION OF THEOREM 2 WITH SYNTHETIC DATASET

We design a synthetic task aligned with the target class analyzed in Theorem 2. Given a sequence $X = \{x(1), \ldots, x(T)\}$ of length $T$ with $x(t) \in \mathbb{R}^4$, the output is

$$y = \sum_{i=1}^{4} \max_{1 \leq t \leq T} a_i^\top x(t),$$

where $a_1, \ldots, a_4 \in \mathbb{R}^4$ are fixed. Inputs are sampled i.i.d. from $x(t) \sim \mathcal{N}(0, I_4)$. For $T \in \{8, 16, 32, 64, 128\}$ we generate 8000 training and 2000 validation examples.

On this task, we evaluate single-layer transformers with head numbers $h \in \{1, 2, 3, 4, 5\}$ and fixed per-head embedding dimension. Each $x(t)$ is embedded via a two-layer ReLU MLP and concatenated with a trainable classification token $c_0$, after which a single-layer multi-head attention block (without residuals or normalization) processes the sequence. A two-layer GELU MLP then outputs the scalar prediction. Both MLPs have the same hidden dimension $N$.

Then for each $(h, T)$, models are trained under multiple random seeds. We report the *minimal normalized mean squared error (NMSE)* across seeds to reduce optimization noise and highlight expressivity. NMSE, equivalent to $1 - R^2$, corrects for the variance shrinkage of maxima as $T$ grows, thus enabling fair comparison across lengths. Further training details and explanations are given in Appendix B.1.1.

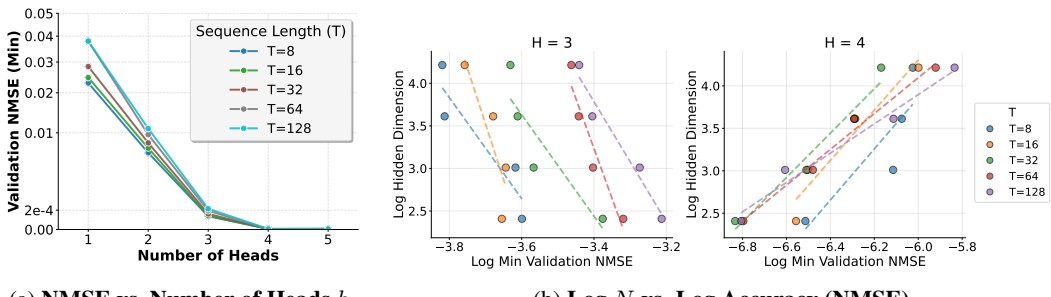

(a) **NMSE vs. Number of Heads $h$.**    (b) **Log $N$ vs. Log Accuracy (NMSE)**

Figure 1: Results on the synthetic example. (a) NMSE vs. number of heads $h$ for sequence lengths $T \in \{8, 16, 32, 64, 128\}$, hidden dimension fixed at $N = 32$. Note that there is a transition at $h = 4$. (A table of mean and variance values corresponding to these curves is provided in Table 2.) (b) Log Hidden Dimension $N$ vs. Log Accuracy for different sequence lengths $T$. The parameter count $k$ for the MLPs change linearly with $N$. (Plots for $h = 1$ and $h = 2$ are in Figure 4.)

Figure 1a shows minimal validation NMSE versus head number $h$ across sequence lengths $T$. Performance improves monotonically with $h$ and exhibits a clear transition near the intrinsic dimension $D = 4$. For $h < D$, NMSE grows with $T$, as limited heads must encode multiple extrema and the FFN becomes inefficient. Once $h \geq D$, curves flatten across $T$, indicating that heads specialize to different coordinates and the FFN aggregates them very effectively. Normalization by NMSE ensures comparability across $T$, despite the increasing concentration of the max-of-Gaussians target.

Figure 1b highlights a phase transition between $h = 3$ and $h = 4$, with $h = D = 4$ equal to the intrinsic dimension of the target. When $h \leq 3$, the negative log NMSE scales approximately linearly with the log parameter count (proportional to the MLP hidden dimension $N$), in agreement with Theorem 2 (2). Moreover, for a fixed parameter count, larger $T$ yields higher NMSE (worse approximation). Equivalently, as indicated by the fitted scaling lines, achieving the same error requires larger parameter counts when $T$ increases, in line with Theorem 2 (2). In contrast, for $h = 4$ these trends change qualitatively. Validation error reaches the order of $10^{-6}$, indicating near-perfect generalization, yet the slope with respect to parameter count reverses: larger MLPs yield slightly higher validation NMSE, a signature of tiny overfitting. The dependence on $T$ also changes in this regime; see Remark B.1.1 in Appendix for details.

We also conducted experiments on synthetic data with the widely used scheme of fixing $E = nh = 32$ constant(For $h = 3, 5$, we choose per-head embedding dimension to be $\lceil 32/h \rceil$, and total embedding dimension becomes $E = 33, 35$. See Table 3 in Appendix for details), as well as experiments

on synthetic datasets with $D = 3$ (See Table 4 in Appendix for details). Both of the above experiments demonstrate similar trends to the $D = 4$ experiments, implying the robustness of our results.

## 6.2 EXPERIMENTS ON REAL DATASETS

We conduct two additional experiments on real datasets to assess the practical relevance of our theoretical findings. The first is a text retrieval task based on MS MARCO, and the second is an image classification task based on CIFAR-10. As there is no natural NMSE-like metric on retrieval tasks and accuracy is most widely used, we focus on training accuracy to isolate architectural expressivity from issues related to optimization or data scarcity. For completeness, we also report test accuracy for both experiments in Table 7 and 9 in Appendix. The experiments examine whether the phase transition around the intrinsic dimension $D$, predicted by Theorem 2, also manifests in practice.

**MS MARCO (text retrieval).** We construct retrieval-style datasets from the MS MARCO passage ranking collection (Bajaj et al., 2016), where each query is paired with one positive passage and $T-1$ mined hard negatives ($T \in \{8, 16, 32, 64\}$). We train a two-layer transformer encoder with per-head embedding dimension fixed at 32, varying the number of heads across $\{1, 2, 4, 6, 8, 10, 12, 14, 16\}$. Input text is tokenized using the BERT tokenizer, and word, positional, and segment embeddings from pretrained BERT (Devlin et al., 2019) are kept frozen. These 768-dimensional embeddings are linearly projected to the embedding size $E = \text{heads} \times 32$, after which only the projection and transformer layers are trained. Full dataset construction and training details are given in Appendix B.2.

**CIFAR-10 (image classification).** We further evaluate on the CIFAR-10 dataset (Krizhevsky, 2009) using a four-layer Vision transformer (ViT) (Dosovitskiy et al., 2021). Each image of size $32 \times 32$ is divided into non-overlapping $8 \times 8$ patches (patch size $= 8$), which are linearly embedded. The transformer encoder has per-head embedding dimension fixed at 16, and we vary the number of heads across $h \in \{1, 2, 4, 8, 10, 11, 12, 13, 14, 16, 20, 24\}$. To vary the sequence length, we extend the border with interpolation around each image to enlarge its side length, after which the sequence length is given by $\left(\frac{\text{image side length}}{\text{patch size}}\right)^2 + 1$, including the classification token. Figure 6 shows some of the examples. Full dataset preprocessing and training details are provided in Appendix B.3.

**Result analysis.** Both experiments exhibit the same qualitative trend as in the synthetic setting. Figure 2a shows that in the text retrieval experiment, when $h < 12$, accuracy declines as the sequence length $T$ increases, consistent with Theorem 2 (2). Once $h > 12$, this dependence on $T$ disappears, and performance remains stable. Taking $Err(h, T) = 1 - \text{Accuracy}(h, T)$ as error, by using $cT^\beta \exp(\alpha h/T^\delta)$ to approximate $(Err(h, T))$ in log scale under MAE and drop-outs ($h = 1, 12, 14, 16$ are dropped out as outliers, $\delta = 0.25 > 0, \alpha = -1.40 < 0$), figure 2b illustrates that when $h < 12$, $-\log(Err(h, T)) \propto h/T^\delta$, highly consistent with the order in Theorem 2 (2) under fixed parameter count $M$. The flattening of curves after $h > 12$ is also consistent with theory.

Figure 2c shows similar trend in image classification, with intrinsic dimension at $h = 10$. Figures 2d and 2e illustrates weighted reversal score, calculated by $R(h) = \frac{1}{w_h} \sum_{T_1 < T_2} \max((err(T_1) - err(T_2)), 0)$ with normalization factor $w_h = \max_T err(T) - \min_T err(T)$, detects the existence of longer $T$ yielding smaller error for this head number $h$. Such phenomenon leads to positive $R(h)$, and it also indicates phase transition as explained in remark B.1.1. Figure 2e further verified this.

## 7 CONCLUSION

In this work we investigated the approximation properties of single-layer transformers. We first introduced a structured target family, the generalized $D$-retrieval task, that is broad enough to capture general sequence-to-vector mappings (Theorem 1). Within this setting, we analyzed how the approximation efficiency of transformers depends on architectural choices, especially the number of head. Our results indicate that having a sufficient number of heads leads to efficient approximation, while an insufficient number of heads forces the parameter count to grow exponentially with sequence length $T$. We also examined the single-head case, where large embedding dimension allows sequence memorization but shifts the complexity to the feed-forward

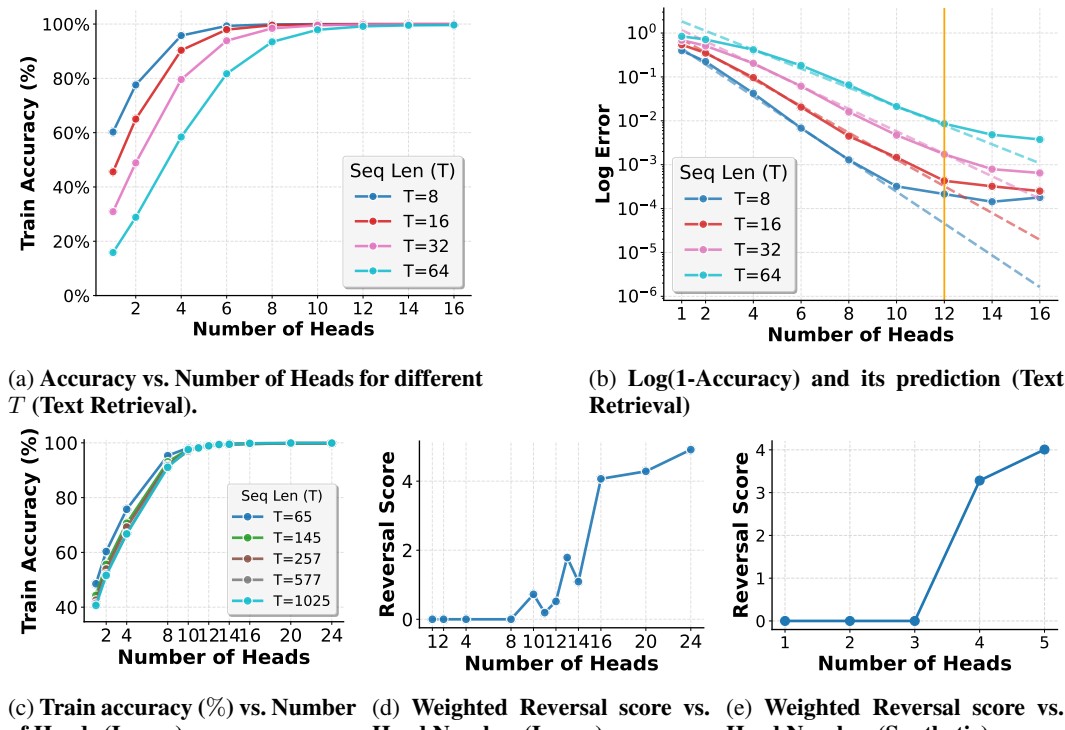

(a) **Accuracy vs. Number of Heads for different** $T$ **(Text Retrieval).**

(b) **Log(1-Accuracy) and its prediction (Text Retrieval)**

(c) **Train accuracy (%) vs. Number of Heads (Image).**

(d) **Weighted Reversal score vs. Head Number (Image)**

(e) **Weighted Reversal score vs. Head Number (Synthetic)**

Figure 2: **Experiments on real datasets.** Training performance with different numbers of heads $h$ across different sequence lengths $T$. (a) Accuracy vs. number of heads for different $T$ in text retrieval; phase transition near $h = 12$. Mean and standard deviation see Table 6. MRR shows a similar trend, see Fig. 5 in the appendix. (b) Phase transition for text retrieval. (c) Accuracy vs. number of heads for different $T$ in image classification; phase transition near $h = 10$. Mean and standard deviation see Table 8. (d) Weighted Reversal Score for Image Classification, $err = 1 - Accuracy$. The plot becomes positive when $h \geq 10$, indicating phase transition. (e) Weighted Reversal Score for Synthetic Experiment, it becomes positive at $h = 4$, exactly the intrinsic dimension of the task.

block (Theorem 2). These findings clarify the roles played by head count in transformer expressivity.

Our experiments on both synthetic and real datasets reveal a non-trivial phase transition around the intrinsic dimension $D$, consistent with theoretical analysis. When the number of heads is below $D$, models exhibit higher error for the same parameter count as sequence length $T$ increases. Once the head count reaches or exceeds $D$, approximation rate becomes independent of sequence lengths $T$. This transition is also observed in real-world datasets with deeper architectures, indicating that the notion of intrinsic dimension is not only theoretical but also practically relevant. In particular, beyond fully training models, analyzing head contributions early in training to estimate how many heads meaningfully affect the output, or training multiple models with varying depths and head counts while tracking how error scales with $T$ are potential ways to probe the task's intrinsic dimension. These experiments might demonstrate whether the inferred intrinsic dimension is stable across architectures, thereby informing head-count selection and the head number to retain under pruning.

**Limitations.** We conclude by noting several limitations of this study. Firstly, although the analyzed target class is dense, the phenomena of interest are most naturally manifested in retrieval-style tasks aligned with our setting. Secondly, our analysis is restricted to single-layer transformers; while experiments on real datasets supports Conjecture 1 in deeper architectures, a rigorous multilayer theory remains open. Finally, the tradeoff between sequence memorization and pattern learning—observed for shorter sequences (cf. Remark B.1.1)—has not yet been established rigorously and warrants further investigation.

ACKNOWLEDGMENTS

This research is supported by the National Research Foundation, Singapore, under the NRF fellowship (project No. NRF-NRFF13-2021-0005). The computational work for this article was fully performed on resources of the National Supercomputing Centre, Singapore (https://www.nscc.sg).

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

# A    Proofs of Main Theorems

## A.1    Proof of Theorem 1

**Proof Sketch.**   The proof proceeds in three steps. First, by Lemma 1, we approximate a broader function class that relaxes the smoothness requirements and the assumptions in Assumption 2. Second, Lemmas 2 and 3 show that by constructing appropriate $S_i$, we can faithfully recover all information from the original input sequence with simple $f_i$. Finally, the outer function $F_0$ can be applied to approximate an arbitrary sequence-to-vector target within this class. Together, these steps establish the result.

*Proof of Theorem 1.*   To prove Theorem 1, we first establish a few auxiliary lemmas.

**Lemma 1** (Relaxed target class and closure equivalence). *Let $\widetilde{\mathcal{F}}_D^{d,T}$ be defined as in  9 but with only $f_i \in C([0,1]^d, [0,1])$, i.e., we drop "unique minimizer", "pairwise distinct"and "PD Hessian".Set $\mathcal{F}^{d,T} := \bigcup_{D \geq 1} \mathcal{F}_D^{d,T}$ and $\widetilde{\mathcal{F}}^{d,T} := \bigcup_{D \geq 1} \widetilde{\mathcal{F}}_D^{d,T}$. Then*

$$\overline{\mathcal{F}^{d,T}}^{\|\cdot\|_\infty} = \overline{\widetilde{\mathcal{F}}^{d,T}}^{\|\cdot\|_\infty} \quad \text{on } \mathcal{X}_T.$$

*Proof.* Fix $\widetilde{H}(X_T) = \widetilde{F}_0(\widetilde{z}_1(X_T), \ldots, \widetilde{z}_D(X_T)) \in \widetilde{\mathcal{F}}_D^{d,T}$ with $\widetilde{z}_i(X_T) = \min_{t \in S_i} \widetilde{f}_i(x(t))$ and $\widetilde{f}_i \in C([0,1]^d, [0,1])$. Let $\varepsilon > 0$. We will construct $H \in \mathcal{F}^{d,T}$ with $\|H - \widetilde{H}\|_\infty \leq \varepsilon$. Firstly, by Stone–Weierstrass, choose $p_i \in C^\infty([0,1]^d)$ so that $\|p_i - \widetilde{f}_i\|_\infty \leq \eta$, where $\eta > 0$ will be fixed later. Because the uniform approximation can slightly leave $[0,1]$, compose with a smooth strictly increasing squashing $s : [-c, 1+c] \to [0,1]$ with $s(u) = u$ on $[0,1]$ and $\|s \circ p_i - p_i\|_\infty \leq \eta$ (for small enough $c > 0$), and replace $p_i$ by $s \circ p_i$. We still write $p_i$ and retain $\|p_i - \widetilde{f}_i\|_\infty \leq 2\eta$.

Secondly, let $\xi_i \in \arg\min_{x \in [0,1]^d} p_i(x)$ (nonempty by compactness). Pick $r \in (0, \frac{1}{4})$ small and a $C^\infty$ bump $\phi_i$ supported in $B(\xi_i, 2r) \cap [0,1]^d$, with $\phi_i(\xi_i) = 1$, $\nabla \phi_i(\xi_i) = 0$, and with $\nabla^2 \phi_i(\xi_i)$ *negative definite*.[1] Define, for parameters $\delta_{i,1}, \delta_{i,2} > 0$ to be fixed,

$$g_i(x) := p_i(x) - \delta_{i,1} \phi_i(x) + \delta_{i,2} \phi_i(x) \|x - \xi_i\|^2.$$

(i) Since $g_i(\xi_i) = p_i(\xi_i) - \delta_{i,1}$ while $g_i(x) \geq p_i(x)$ whenever $\phi_i(x) = 0$ and $g_i(x) > p_i(x)$ for $x \in B(\xi_i, 2r) \setminus \{\xi_i\}$, we get that $\xi_i$ *is the unique global minimizer* of $g_i$.

(ii) At $\xi_i$, because $\nabla \phi_i(\xi_i) = 0$,

$$\nabla^2 g_i(\xi_i) = \nabla^2 p_i(\xi_i) - \delta_{i,1} \nabla^2 \phi_i(\xi_i) + 2\delta_{i,2} I.$$

Here $-\nabla^2 \phi_i(\xi_i) \succ 0$, so choosing $(\delta_{i,1}, \delta_{i,2})$ suitably makes $\nabla^2 g_i(\xi_i) \succ 0$ (PD). Because $\|\phi_i\|_\infty \leq 1$ and $\| \|x - \xi_i\|^2 \|_\infty \leq d$ on $[0,1]^d$,

$$\|g_i - p_i\|_\infty \leq \delta_{i,1} + \delta_{i,2} (2r)^2.$$

Hence, by taking $r$ small and then $\delta_{i,1}, \delta_{i,2}$ small (using the $r^{-2}$ scaling in $\nabla^2 \phi_i(\xi_i)$ to keep the Hessian PD), we can ensure both PD at $\xi_i$ and $\|g_i - p_i\|_\infty \leq \eta$.

Thirdly, we use tiny translation to remove distinctiveness. It may happen that $\xi_i = \xi_{i'}$ for some $i \neq i'$. Choose pairwise distinct small vectors $v_i \in \mathbb{R}^d$ and fix a smooth cutoff $\chi \in C^\infty([0,1]^d, [0,1])$ that equals 1 on $[r, 1-r]^d$ and vanishes near the boundary. Define a $C^\infty$ diffeomorphism of the cube,

$$\Phi_i(x) := x - \varepsilon_i \chi(x) v_i, \qquad \text{with } \varepsilon_i > 0 \text{ small}.$$

Then $\Phi_i$ is arbitrarily close to the identity in $C^1$ for small $\varepsilon_i$, maps $[0,1]^d$ to itself, and $h_i := g_i \circ \Phi_i$ has a (unique) minimizer at $x^{(i)} := \Phi_i^{-1}(\xi_i)$. For different $i$, these points are distinct if the $v_i$'s are distinct and $\varepsilon_i$'s are small but nonzero. Moreover, because $\nabla g_i(\xi_i) = 0$, the Hessian at $x^{(i)}$ satisfies

$$\nabla^2 h_i(x^{(i)}) = D\Phi_i(x^{(i)})^\top \nabla^2 g_i(\xi_i) D\Phi_i(x^{(i)}) \succ 0,$$

---

[1] For instance take $\phi_i(x) = \psi(\|x - \xi_i\|^2 / r^2)$ with $\psi(0) = 1$, $\psi' < 0$ near 0, $\psi \equiv 0$ on $[1, \infty)$; then $\nabla^2 \phi_i(\xi_i) \prec 0$ and its norm scales like $r^{-2}$.

so PD is preserved. Since $g_i$ is Lipschitz on the compact cube, $\|h_i - g_i\|_\infty \le L_i\,\varepsilon_i$ for some $L_i$, hence by taking $\varepsilon_i$ small we get $\|h_i - g_i\|_\infty \le \eta$.

Finally, if needed, compose with the same strictly increasing squashing $s$ as in Step 1 and set

$$f_i := s \circ h_i \in C^2([0,1]^d, [0,1]).$$

Because $s$ is strictly increasing, it preserves the minimizer location and, at the minimizer $x^{(i)}$, $\nabla^2(s \circ h_i)(x^{(i)}) = s'\big(h_i(x^{(i)})\big)\,\nabla^2 h_i(x^{(i)}) \succ 0$. Also $\|f_i - h_i\|_\infty \le \eta$ by construction.

Collecting the bounds from previous deduction:

$$\|f_i - \widetilde{f}_i\|_\infty \;\le\; \underbrace{\|p_i - \widetilde{f}_i\|_\infty}_{\le 2\eta} + \underbrace{\|g_i - p_i\|_\infty}_{\le \eta} + \underbrace{\|h_i - g_i\|_\infty}_{\le \eta} + \underbrace{\|f_i - h_i\|_\infty}_{\le \eta} \;\le\; 5\eta.$$

For each $i$, the map $u \mapsto \min_{t \in S_i} u_t$ is 1-Lipschitz in $\|\cdot\|_\infty$. Hence the corresponding features $\bar{z}_i(X_T) := \min_{t \in S_i} f_i(x(t))$ and $\widetilde{z}_i(X_T) := \min_{t \in S_i} \widetilde{f}_i(x(t))$ satisfy $\|\bar{z}_i - \widetilde{z}_i\|_\infty \le 5\eta$. Let $\omega_{\widetilde{F}_0}$ be a modulus of continuity of $\widetilde{F}_0$ on $[0,1]^D$. Choose $\eta$ so small that $\omega_{\widetilde{F}_0}(5\eta) \le \varepsilon/2$. Then

$$\big\|\widetilde{F}_0\big(\bar{z}(X_T)\big) - \widetilde{F}_0\big(\widetilde{z}(X_T)\big)\big\|_\infty \le \varepsilon/2.$$

Finally, approximate $\widetilde{F}_0$ uniformly on $[0,1]^D$ by some $F_0 \in C^1([0,1]^D)$ within $\varepsilon/2$ (Stone–Weierstrass). Setting

$$H(X_T) := F_0\big(\bar{z}_1(X_T), \ldots, \bar{z}_D(X_T)\big) \in \mathcal{F}^{d,T},$$

we obtain

$$\|H - \widetilde{H}\|_\infty \;\le\; \underbrace{\|F_0(\bar{z}) - \widetilde{F}_0(\bar{z})\|_\infty}_{\le \varepsilon/2} + \underbrace{\|\widetilde{F}_0(z) - \widetilde{F}_0(\widetilde{z})\|_\infty}_{\le \varepsilon/2} \;\le\; \varepsilon.$$

This shows $\widetilde{\mathcal{F}}^{d,T} \subset \overline{\mathcal{F}^{d,T}}^{\|\cdot\|_\infty}$. The reverse inclusion is simple, hence we have the lemma.

$\square$

*Remark.* Thus, we now focus on the relaxed class $\widetilde{\mathcal{F}}^{d,T}$ and Lemma 1 lifts the result to the original class $\mathcal{F}^{d,T}$.

**Lemma 2** (Order-statistic in the relaxed class). *Without loss of generation, suppose $4|T$. Let $m = \frac{T}{4}$. For each $j \in [d]$ and $X_T = \{x(1), \ldots, x(T)\}$, define*

$$U_j(X_T) := \max_{\substack{B \subseteq [T] \\ |B|=m}} \min_{u \in B} x(u)_j,$$

*and for each fixed $t \in [T]$,*

$$Y_{t,j}(X_T) := \max_{\substack{A \subseteq [T], \ t \in A \\ |A|=m}} \min_{u \in A} x(u)_j, \qquad Z_{t,j}(X_T) := \max_{\substack{A \subseteq [T], \ t \in A \\ |A|=m}} \min_{u \in A}(1 - x(u)_j).$$

*Let $v_{1,j} \ge \cdots \ge v_{T,j}$ be the sorted values of $\{x(1)_j, \ldots, x(T)_j\}$ and set $U_j = v_{m,j}$. For the multi-set $\{x(u)_j : u \in [T]\}$, let $v_{1,j} \ge \cdots \ge v_{T,j}$ (nonincreasing) and $w_{1,j} \le \cdots \le w_{T,j}$ (nondecreasing). Then we have*

$$Y_{t,j} = \min\{x(t)_j, \ v_{m,j}\}, \qquad 1 - Z_{t,j} = \max\{x(t)_j, \ w_{m,j}\}.$$

*In particular:*

$$x(t)_j \le v_{m,j} \ \Rightarrow\ Y_{t,j} = x(t)_j, \qquad x(t)_j \ge w_{m,j} \ \Rightarrow\ 1 - Z_{t,j} = x(t)_j.$$

*Proof.* Among all $m$-subsets $B$, the maximum of $\min_{u \in B} x(u)_j$ is attained by picking the $m$ largest coordinates, so $U_j = v_{m,j}$. Forcing $t \in A$, the choice of the other $m-1$ indices to maximize the minimum is the $m-1$ largest among $\{x(u)_j : u \ne t\}$, hence $Y_{t,j} = \min\{x(t)_j, v_{m,j}\}$. For $Z_{t,j}$, note that $\min_{u \in A}(1 - x(u)_j) = 1 - \max_{u \in A} x(u)_j$, so maximizing it over $A$ is the same as minimizing $\max_{u \in A} x(u)_j$, which picks $t$ plus the $(m-1)$ smallest leading to $1 - Z_{t,j}$. The particular statements follow immediately. $\square$

**Lemma 3** (Smooth selector). *For $q > 0$ define*

$$\widehat{x}_q(t)_j \; := \; \frac{e^{q(\, 1-Z_{t,j}-w_{m,j}\,)^2}\,(1-Z_{t,j}) + e^{q(\, Y_{t,j}-v_{m,j}\,)^2}\,Y_{t,j}}{e^{q(\, 1-Z_{t,j}-w_{m,j}\,)^2} + e^{q(\, Y_{t,j}-v_{m,j}\,)^2}}.$$

*Then $\widehat{x}_q(t)_j \to x(t)_j$ uniformly on $\mathcal{X}_T$ as $q \to \infty$.*

*Proof.* From Lemma 2, if $x(t)_j \geq U_j$, we have $Y_{t,j} = v_{m,j}$ and $(Y_{t,j} - v_{m,j})^2 = 0$ while $1 - Z_{t,j} = x(t)_j$ and $(1 - Z_{t,j} - w_{m,j})^2 > 0$. Thus, as $q \to \infty$, the weight concentrates on $(1 - Z_{t,j}) = x(t)_j$. If $x(t)_j \leq U_j$ and $w_{m,j} \geq x(t)_j$, we have $(1 - Z_{t,j} - w_{m,j})^2 = 0$ while $Y_{t,j} = x(t)_j$ and $(Y_{t,j} - v_{m,j})^2 > 0$ concentrating on $Y_{t,j} = x(t)_j$. If $w_{m,j} \leq x(t)_j \leq U_j$, we have $1 - Z_{t,j} = Y_{t,j} = x(t)_j$, so either of three settings leads to $x(t)_j$. The compactness of $\mathcal{X}_T$ gives us the uniform property. $\qquad\square$

Now, we begin our formal proof for Theorem 1.

By Lemma 1, for any $F \in C(\mathcal{X}_T)$ and $\varepsilon > 0$, it suffices to construct an $\widetilde{H} \in \widetilde{\mathcal{F}}^{d,T}$ with $\|F - \widetilde{H}\|_\infty \leq \varepsilon/2$, since the lemma 1 could lift it to $\mathcal{F}^{d,T}$ with another $\varepsilon/2$.

Fix $m = \frac{T}{4}$. For each coordinate $j$ and each $S \subseteq [T]$ with $|S| = m$, include the relaxed primitives

$$z^{(j,S)}(X_T) := \min_{t \in S} x(t)_j, \qquad \bar{z}^{(j,S)}(X_T) := \min_{t \in S}(1 - x(t)_j).$$

To form the thresholds $v_{m,j}$ and $w_{m,j}$ needed in Lemma 2, additionally include:

$$\min_{t \in S} x(t)_j \quad \text{for all } S \subseteq [T] \setminus \{t\} \text{ with } |S| = m,$$

and

$$\min_{t \in S}(1 - x(t)_j) \quad \text{for all } S \subseteq [T] \setminus \{t\} \text{ with } |S| = T - m,$$

Using smooth log-sum-exp (softmax) in the outer function $\widetilde{F}_0$, we can recover the subset-wise maximum required to compute $U_j, Y_{t,j}, Z_{t,j}, v_{m,j}$ and $w_{m,j}$ from these primitives.

By Lemma 3, for any $\delta > 0$ there exists $q$ such that

$$\max_{X_T \in \mathcal{X}_T} \max_{t \in [T],\, j \in [d]} \left| \widehat{x}_q(t)_j - x(t)_j \right| \; \leq \; \delta.$$

By uniform continuity of $F$ on the compact $\mathcal{X}_T$, choose $\delta$ so that this implies $|F(X_T) - F(\widehat{X}_q(T))| \leq \varepsilon/4$ for all $X_T$, where $\widehat{X}_q(T)$ stacks the coordinates $\widehat{x}_q(t)_j$. We approximate the continuous map $u \mapsto F(u)$ on $[0,1]^{dT}$ uniformly by a polynomial $P$ within $\varepsilon/4$ (Stone–Weierstrass). Define

$$\widetilde{H}(X_T) := \big(P \circ \mathrm{vec}\big)(\widehat{X}_q(T)),$$

which is a $C^1$ function of the inner features. Hence $\widetilde{H} \in \widetilde{\mathcal{F}}^{d,T}$ and

$$\|F - \widetilde{H}\|_\infty \leq \underbrace{\|F - F \circ \widehat{X}_q\|_\infty}_{\leq \varepsilon/4} + \underbrace{\|F \circ \widehat{X}_q - P \circ \widehat{X}_q\|_\infty}_{\leq \varepsilon/4} \leq \varepsilon/2.$$

Apply Lemma 1 to replace each relaxed primitives by admissible $C^2$ functions with unique minimizers and to replace $\widetilde{F}_0$ by function $F_0$ so that the final error increases by at most $\varepsilon/2$. This leads to $f \in \mathcal{F}_D^{d,T}$ with $\|F - f\|_\infty \leq \varepsilon$.

$\qquad\square$

## A.2 Proof of Theorem 2

### A.2.1 Proof of Theorem 2 (1)

Here we prove Theorem 2 (1)

**Proof Sketch.** The idea is straightforward. Each attention head is assigned to approximate one term $\min_{t \in S_i} f_i(x(t))$. Once these components are extracted, the outer function $F_0$ can be approximated by a suitable $\hat{F}$, completing the construction.

*Proof of Theorem 2 (1): Sufficient expressivity with D heads.* Fix $d, T$ and $\alpha \in (0, 1)$, and let $\varepsilon > 0$ be given. Throughout the proof, constants depending only on $(d, D, T)$ are absorbed into $C_{d,D,T} > 0$ which may change from line to line.

In the display equation 4, each head produces an $n$-dimensional vector and $\text{Concat}_{i=1}^{h}$ gives a vector in $\mathbb{R}^{nh}$ before $\hat{F}$. For the construction, we realize the usual *block-by-head* parameterization, which means that the encoder outputs a block-decomposed embedding

$$\hat{x}(t) = \big(\hat{x}^{(1)}(t), \ldots, \hat{x}^{(D)}(t)\big) \in \mathbb{R}^{2D}, \qquad \hat{x}^{(i)}(t) \in \mathbb{R}^2,$$

and the $i$-th head only reads the $i$-th block via block-diagonal $W_{Q,i}, W_{V,i}$ (entries bounded by 1). This keeps the parameter counts within the same order. We therefore set the *per-head embedding dimension* to $n = 2$.

Firstly, from Assumption (A2), for any $\delta > 0$ there exist two-layer FFNs $\Psi_{i,\delta} : [0, 1]^d \to [0, 1]$ such that

$$\max_{x \in [0,1]^d} |f_i(x) - \Psi_{i,\delta}(x)| \leq \delta, \qquad \text{width}(\Psi_{i,\delta}) \leq \frac{C_2}{\delta^{\gamma_f}}, \tag{10}$$

where $\gamma_f > 0$ is the exponent from (A2). Define for each head $i$, the position gate

$$r_i(s) := \begin{cases} 0, & s \in S_i, \\ -1, & s \notin S_i, \end{cases} \qquad s \in [T].$$

(Recall $S_i \subset [T]$ with $|S_i| \geq \alpha T$ by equation 8.) We implement the encoder $P_\phi$ so that its $i$-th block is

$$\hat{x}^{(i)}(t) = \big(\Psi_{i,\delta}(x(t)), r_i(t)\big) \in [0, 1] \times \{-1, 0\} \subset [-1, 1]^2. \tag{11}$$

This choice follows $\|\hat{x}(t)\|_2 \leq \sqrt{2D}$. After a fixed rescaling (absorbed into $\beta$), this meets the norm constraint.

Secondly, we would like to use head-wise attention to isolate the minimum on $S_i$. For each head $i$, we take a single attention logit ($m_h = 1$) by choosing

$$W_{O,i} = I, \qquad W_{K,i} = [-1 \quad 1], \qquad W_{Q,i}\hat{c}_0 = 1, \qquad W_{V,i} = I_2.$$

All entries are within the allowed bound 1. With the block equation 11, the (pre-softmax) score of token $t$ in head $i$ is

$$\rho_i(t) = (W_{K,i}\hat{x}^{(i)}(t))^\top (W_{Q,i}\hat{c}_0) = -\Psi_{i,\delta}\big(x(t)\big) + r_i(t). \tag{12}$$

Let $\sigma[\rho_i]$ be the softmax equation 5 with $\beta > 0$. Define the head-$i$ value readout (first coordinate of the head output)

$$\tilde{z}_i(X_T) := \sum_{t=1}^{T} \sigma[\rho_i](t) \, \Psi_{i,\delta}\big(x(t)\big). \tag{13}$$

Here, the second coordinate is unused. $\hat{F}$ could ignore it via a fixed linear projection, counted in the constant $C_{d,D,T}$.

Now, we give a uniform bound on $S_i$. Take $a_t := \Psi_{i,\delta}(x(t)) \in [0, 1]$ and split the sum into $S_i$ and $S_i^c$. From $r_i(t) = 0$ on $S_i$ and $r_i(t) = -1$ on $S_i^c$, we have

$$\sigma[\rho_i](t) = \frac{e^{\beta(-a_t + r_i(t))}}{\sum_{u \in S_i} e^{-\beta a_u} + \sum_{u \in S_i^c} e^{-\beta(a_u+1)}} \leq \begin{cases} \dfrac{e^{-\beta a_t}}{\sum_{u \in S_i} e^{-\beta a_u}}, & t \in S_i, \\ e^{-\beta} \cdot \dfrac{e^{-\beta a_t}}{\sum_{u \in S_i} e^{-\beta a_u}}, & t \in S_i^c. \end{cases}$$

Hence, we have

$$\tilde{z}_i = \sum_{t \in S_i} \sigma[\rho_i](t) a_t + \sum_{t \in S_i^c} \sigma[\rho_i](t) a_t \leq \underbrace{\frac{\sum_{t \in S_i} a_t e^{-\beta a_t}}{\sum_{u \in S_i} e^{-\beta a_u}}}_{\text{Gibbs mean on } S_i} + e^{-\beta} \frac{\sum_{t \in S_i^c} a_t e^{-\beta a_t}}{\sum_{u \in S_i} e^{-\beta a_u}}. \qquad (14)$$

To simplify, we denote $a_* := \min_{t \in S_i} a_t$ and $b_t := a_t - a_* \in [0, 1]$ for $t \in S_i$. Then, we have

$$\frac{\sum_{t \in S_i} a_t e^{-\beta a_t}}{\sum_{u \in S_i} e^{-\beta a_u}} = a_* + \frac{\sum_{t \in S_i} b_t e^{-\beta b_t}}{\sum_{u \in S_i} e^{-\beta b_u}} \leq a_* + \sum_{t \in S_i} b_t e^{-\beta b_t} \leq a_* + \frac{|S_i| - 1}{e\beta},$$

The inequality comes from $\sup_{b \in [0,1]} b e^{-\beta b} = e^{-1}/\beta$ for $\beta \geq 1$ and one of the $b_t$ is 0, so the denominator in the middle fraction is $\geq 1$. For the $S_i^c$ term in 14, we use $a_t \leq 1$ and $\sum_{u \in S_i} e^{-\beta a_u} \geq 1$ to get

$$e^{-\beta} \frac{\sum_{t \in S_i^c} a_t e^{-\beta a_t}}{\sum_{u \in S_i} e^{-\beta a_u}} \leq e^{-\beta} |S_i^c| \leq e^{-\beta} T.$$

Combining the two bounds, we have the uniform estimate:

$$\min_{t \in S_i} \Psi_{i,\delta}\big(x(t)\big) \leq \tilde{z}_i(X_T) \leq \min_{t \in S_i} \Psi_{i,\delta}\big(x(t)\big) + \frac{|S_i| - 1}{e\beta} + Te^{-\beta} \qquad (\beta \geq 1). \qquad (15)$$

In particular, since $|S_i| \leq T$, there is a constant $C_T$ with

$$0 \leq \tilde{z}_i(X_T) - \min_{t \in S_i} \Psi_{i,\delta}\big(x(t)\big) \leq C_T \Big(\frac{1}{\beta} + e^{-\beta}\Big), \qquad C_T := \max\{T/e, \, T\}. \qquad (16)$$

Thirdly, we need to lift bounds from $\tilde{z}_i$ to $z_i$. From equation 10 and the definition of $z_i$,

$$\Big|\min_{t \in S_i} f_i\big(x(t)\big) - \min_{t \in S_i} \Psi_{i,\delta}\big(x(t)\big)\Big| \leq \delta.$$

Together with equation 16,

$$\big|\tilde{z}_i(X_T) - \bar{z}_i(X_T)\big| \leq \delta + C_T \Big(\frac{1}{\beta} + e^{-\beta}\Big) \qquad \text{for all } X_T \in \mathcal{X}_T, \, i = 1, \dots, D. \qquad (17)$$

Let $L_0 := \sup_{z \in [0,1]^D} \|\nabla F_0(z)\|_1 < \infty$ (compactness and $C^1$). Choose

$$\delta := \frac{\varepsilon}{4L_0 D}, \qquad \beta \geq \beta_\varepsilon := \max\Big\{1, \, \frac{4C_T L_0 D}{\varepsilon}, \, \log\Big(\frac{4C_T L_0 D}{\varepsilon}\Big)\Big\}.$$

Then by equation 17,

$$\|\tilde{z}(X_T) - z(X_T)\|_\infty \leq \frac{\varepsilon}{2L_0 D} \qquad \text{for all } X_T \in \mathcal{X}_T. \qquad (18)$$

Finally, we construct the approximation for $F_0$ and count the number of parameter. By Assumption (A1), there exists a two-layer FFN $\Phi_{\delta_0} : [0,1]^D \to \mathbb{R}$ with width $\leq C_1/\delta_0^{\gamma_0}$ (for some $\gamma_0 > 0$) such that

$$\max_{z \in [0,1]^D} |F_0(z) - \Phi_{\delta_0}(z)| \leq \delta_0.$$

Set $\delta_0 := \varepsilon/2$. Define the model's final feed-forward $\hat{F}$ to project $\mathbb{R}^{2D} \to \mathbb{R}^D$ by keeping the first coordinate of each head (a fixed linear map with entries in $\{0, 1\}$) and apply $\Phi_{\delta_0}$.

Then for all $X_T$, we have

$$\begin{aligned}
\big|\hat{F}(\text{Concat}_i(\cdot)) - F_0\big(z(X_T)\big)\big| &\leq \big|\Phi_{\delta_0}\big(\tilde{z}(X_T)\big) - \Phi_{\delta_0}\big(z(X_T)\big)\big| + \big|\Phi_{\delta_0}\big(z(X_T)\big) - F_0\big(z(X_T)\big)\big| \\
&\leq L_0 \|\tilde{z}(X_T) - z(X_T)\|_1 + \delta_0 \\
&\leq L_0 D \|\tilde{z}(X_T) - z(X_T)\|_\infty + \varepsilon/2 \\
&\leq \varepsilon/2 + \varepsilon/2 = \varepsilon,
\end{aligned}$$

where we used equation 18 in the last inequality.

Here, the trainable components are composed of three parts:

- the $D$ subnetworks $\Psi_{i,\delta}$ inside the encoder blocks equation 11;

- the fixed-size projections $W_{Q,i}, W_{K,i}, W_{V,i}$ (size $O(D)$ and independent of $\varepsilon$);

- the two-layer FFN $\Phi_{\delta_0}$ used inside $\hat{F}$.

Thus

$$M \;\leq\; C' D \cdot \frac{1}{\delta^{\gamma_f}} \;+\; C'' \cdot \frac{1}{\delta_0^{\gamma_0}} \;+\; C''' \qquad \text{for constants } C', C'', C''' = C_{d,D,T}.$$

With $\delta = \Theta(\varepsilon)$ and $\delta_0 = \Theta(\varepsilon)$ chosen above,

$$M \;\leq\; \frac{C_{d,D,T}}{\varepsilon^\gamma}, \qquad \gamma := \max\{\gamma_f, \gamma_0\},$$

and the construction uses $h = D$ heads with per-head dimension $n = 2$ and achieves $\varepsilon$-approximation on $\mathcal{X}_T$. This proves Theorem 2 (1). $\qquad\square$

### A.2.2 Proof of Theorem 2 (2)

**Proof Sketch.** The argument proceeds in two parts. The core idea is to construct two sequences whose representations after the attention layer are indistinguishably close, on the order of $O(\epsilon^{k+1})$, yet whose target outputs differ by at least $3\epsilon$. Lemma 4 then implies the lower bound on the parameter count required for approximation.

Using Lemmas 5, 6, and 7, we obtain $D$ disjoint neighborhoods around the minima $x^{(i)}$. Since $D > s = h$, there exists at least one neighborhood not selected by the $s$ heads. Within this region, the pigeonhole principle guarantees the existence of two distinct subsequences. By carefully designing these subsequences, we ensure that their outputs after the attention layer are nearly indistinguishable, while their target values differ by at least $3\epsilon$. Extending them to full sequences completes the construction.

We now turn to the full proof. To establish Theorem 2 (2), we begin by introducing several auxiliary lemmas that will serve as building blocks for the argument. Lemma 5, 6, and 7 are only to set up the approximation problem into a more tractable form.

**Lemma 4.** *Let $v_1, v_2 \in \mathbb{R}^n$. Suppose*

$$\|v_1 - v_2\|_2 \leq A \quad \text{and} \quad \|\hat{F}(v_1) - \hat{F}(v_2)\| \geq B,$$

*where $\hat{F} : \mathbb{R}^n \to \mathbb{R}^m$ is a two-layer feed-forward network satisfying the constraints stated above. Then $\hat{F}$ must use at least*

$$\Omega\!\left(\frac{B}{A\sqrt{n}}\right)$$

*parameters.*

*Proof of Lemma 4.* Let $\Delta x := v_1 - v_2$ and $\Delta F := \hat{F}(v_1) - \hat{F}(v_2)$. Suppose the two-layer network with width $p$ be

$$\hat{F}(x) \;=\; V\,\sigma(Ux + b) + c,$$

where $U \in \mathbb{R}^{p \times n}$, $V \in \mathbb{R}^{m \times p}$, $b \in \mathbb{R}^p$, $c \in \mathbb{R}^m$, $\sigma$ is 1-Lipschitz acting coordinate-wise and every entry of $U, V, b, c$ has magnitude at most 1.

For the $j$-th output coordinate, we have

$$\Delta F_j = \sum_{r=1}^{p} V_{jr}\big(\sigma(u_r^\top v_1 + b_r) - \sigma(u_r^\top v_2 + b_r)\big),$$

where $u_r^\top$ is the $r$-th row of $U$. Using the 1-Lipschitz property of $\sigma$ and Cauchy–Schwarz inequality, we have

$$|\Delta F_j| \leq \sum_{r=1}^{p} |V_{jr}|\,|u_r^\top \Delta x| \leq \sum_{r=1}^{p} |V_{jr}|\,\|u_r\|_2\,\|\Delta x\|_2.$$

By the entrywise weight bound, $\|u_r\|_2 \le \sqrt{n}$ and $|V_{jr}| \le 1$. Therefore, for all $j$, we have

$$|\Delta F_j| \;\le\; p\sqrt{n}\,\|\Delta x\|_2. \tag{19}$$

Let $\|\cdot\|$ be the norm used in the lemma statement. By norm equivalence in finite dimensions, there exists $c_m \in (0,1]$ depending only on the chosen norm and $m$ such that

$$\|y\| \le \frac{1}{c_m}\|y\|_\infty \quad \text{for all } y \in \mathbb{R}^m.$$

$\|\Delta F\| \ge B$ implies $\|\Delta F\|_\infty \ge c_m B$, so there is some $j^\star$ with

$$|\Delta F_{j^\star}| \;\ge\; c_m B.$$

Combining this with equation 19 and $\|\Delta x\|_2 \le A$, we have

$$c_m B \;\le\; p\sqrt{n}\,A \quad\Longrightarrow\quad p \;\ge\; \frac{c_m B}{A\sqrt{n}}.$$

Finally, let $p_{\mathrm{eff}} \le p$ be the number of hidden units that actually affect the output, i.e., those with a nonzero row in $U$ and a nonzero entry in the $j^\star$-th row of $V$. The above bound holds with $p_{\mathrm{eff}}$ in place of $p$, hence $p_{\mathrm{eff}} \ge c_m B/(A\sqrt{n})$. Each such unit uses at least one nonzero parameter in $U$ and one in $V$, so the parameter counts $k$ satisfy $k \ge p_{\mathrm{eff}}$. Therefore

$$k \;\ge\; \frac{c_m B}{A\sqrt{n}} \;=\; \Omega\!\left(\frac{B}{A\sqrt{n}}\right),$$

which proves the lemma. $\qquad\square$

**Lemma 5.** *There exists $R > 0$ such that for every $i \in \{1,\dots,D\}$ and every $r < R$, there exist constants $\delta_i > 0$ and $L_i > 0$ with the following property: there exists a segment $G_i \subset B(x^{(i)}, r)$ of length $\delta_i$ such that*

$$|f_i(x) - f_i(y)| \;\ge\; L_i\,\|x-y\|_2, \qquad \forall\, x,y \in G_i.$$

*and moreover*

$$f_i(x) > z_i, \qquad \forall\, x \in G_i,$$

*Proof of Lemma 5.* Fix $i \in \{1,\dots,D\}$ and denote $x^\star := x^{(i)}$, $f := f_i$ and $H_\star := \nabla_x^2 f(x^\star)$. By positive definiteness, let $\lambda_i := \lambda_{\min}(H_\star) > 0$. By continuity of $\nabla^2 f$, there exists $R_i^{\mathrm{H}} > 0$ such that

$$\nabla^2 f(x) \;\succeq\; \tfrac{\lambda_i}{2} I \qquad \text{for all } x \in B(x^\star, R_i^{\mathrm{H}}).$$

Set $\mu_i := \lambda_i/2 > 0$. Because the domain is $[0,1]^d$ and $x^\star \in [0,1]^d$, we could choose a unit vector $v_i$ pointing strictly into the cube at $x^\star$ (if $x^\star$ is interior, take any unit vector). Define

$$\tau_i \;:=\; \sup\{\, t > 0 \;:\; x^\star + s v_i \in [0,1]^d \text{ for all } s \in [0,t] \,\} \;>\; 0,$$

and set $R_i := \min\{R_i^{\mathrm{H}}, \tau_i\}$. Take $R := \min_{1 \le i \le D} R_i > 0$. Fix any $r \in (0,R)$ and consider the restriction

$$g(t) \;:=\; f(x^\star + t v_i), \qquad t \in [0,r].$$

Then

$$g''(t) \;=\; v_i^\top \nabla^2 f(x^\star + t v_i)\, v_i \;\ge\; \mu_i \quad \text{for all } t \in [0,r].$$

Since $x^\star$ minimizes $f$ on $[0,1]^d$ and $v_i$ is feasible inward, we have $g(t) \ge g(0)$ for small $t \ge 0$ leading to the one-sided derivative $g'(0+) \ge 0$ (if $x^\star$ is interior then $\nabla f(x^\star) = 0$ so $g'(0) = 0$). Because $g'' \ge \mu_i$, the derivative $g'$ is increasing and thus

$$g'(t) \;\ge\; g'(0+) + \mu_i t \;\ge\; \mu_i t, \qquad t \in [0,r].$$

Let $a := r/4$ and $b := r/2$ and define the segment

$$G_i \;:=\; \{\, x^\star + t v_i : t \in [a,b] \,\} \;\subset\; B(x^\star, r),$$

whose length is $\delta_i := b - a = r/4$. For any $x = x^\star + t v_i$ and $y = x^\star + s v_i$ in $G_i$ with $t > s$, the mean value theorem gives some $\xi \in (s,t) \subset [a,b]$ such that

$$|f(x) - f(y)| \;=\; |g(t) - g(s)| \;=\; |g'(\xi)|\,|t-s| \;\ge\; \mu_i a\,|t-s| \;=\; \left(\frac{\lambda_i r}{8}\right)\|x-y\|_2.$$

Therefore the choice

$$L_i \; := \; \frac{\lambda_i r}{8} \; > \; 0$$

works uniformly for all $x, y \in G_i$. The segment $G_i$ does not contain $x^\star$, for all its points are at distance at least $a = r/4 > 0$ from $x^\star$. By uniqueness of the minimizer, $f(x) > f(x^\star) = z_i$ for all $x \in G_i$.

The lemma holds with $\delta_i = r/4$ and $L_i = (\lambda_{\min}(\nabla_x^2 f_i(x^{(i)})) \, r)/8$.  $\qquad\square$

**Lemma 6.** *Let $(z_1, \ldots, z_D)$ denote the minima defined above. Then there exist constants $r_0 > 0$ and $L_0 > 0$ such that the following holds: for any $i \in \{1, \ldots, D\}$ and any perturbation $\delta_0$ with $|\delta_0| < r_0$,*

$$\left| F_0(z_1, \ldots, z_i + \delta_0, \ldots, z_D) - F_0(z_1, \ldots, z_D) \right| \; \geq \; L_0 \, |\delta_0|.$$

*Proof of Lemma 6.* Denote $e_i$ for the $i$-th standard basis vector of $\mathbb{R}^D$. By assumption, $m_i := \left| \partial_i F_0(z) \right| > 0$ for each $i$. Since $F_0 \in C^1$, the map $u \mapsto \partial_i F_0(u)$ is continuous at $z$. Hence for each $i$, there exists $r_i^{\mathrm{cont}} > 0$ such that

$$\left| \partial_i F_0(u) \right| \; \geq \; \tfrac{1}{2} m_i \quad \text{whenever} \quad \|u - z\|_\infty < r_i^{\mathrm{cont}}.$$

If necessary, shrink $r_i^{\mathrm{cont}}$ so that the line segment $\{ z + t e_i : |t| < r_i^{\mathrm{cont}} \}$ lies in $[0,1]^D$. Define uniform constants

$$L_0 \; := \; \tfrac{1}{2} \min_{1 \leq i \leq D} m_i \; > \; 0, \qquad r_0 \; := \; \min_{1 \leq i \leq D} r_i^{\mathrm{cont}} \; > \; 0.$$

Fix $i$ and $\delta_0$ with $|\delta_0| < r_0$. Consider the one-dimensional slice $g_i(t) := F_0(z + t e_i)$ for $|t| < r_0$. Then $g_i$ is $C^1$ and $g_i'(t) = \partial_i F_0(z + t e_i)$. By the mean value theorem, there exists $\theta \in (0,1)$ such that

$$F_0(z + \delta_0 e_i) - F_0(z) = g_i'(\theta \delta_0) \, \delta_0 = \partial_i F_0\bigl(z + \theta \delta_0 e_i\bigr) \, \delta_0.$$

Taking absolute values and using the lower bound on $\left| \partial_i F_0(\cdot) \right|$ inside the $\ell_\infty$-ball of radius $r_0$ around $z$, we have

$$\left| F_0(z + \delta_0 e_i) - F_0(z) \right| \; \geq \; L_0 \, |\delta_0|,$$

which is the desired inequality.  $\qquad\square$

**Lemma 7.** *Let $z_i = \min_{x \in [0,1]^d} f_i(x)$ and let $x^{(i)}$ denote the unique minimizer of $f_i$ (as assumed above). Then there exist constants $R_0 > 0$ and $\varepsilon_0 > 0$ such that:*

1. *The open balls $\{ B(x^{(i)}, R_0) \}_{i=1}^{D}$ are pairwise disjoint.*

2. *For each $i \in \{1, \ldots, D\}$ and every $x \in [0,1]^d \setminus B(x^{(i)}, R_0)$,*

$$f_i(x) \; > \; z_i + \varepsilon_0.$$

*Proof of Lemma 7.* Since the minimizers $\{x^{(i)}\}_{i=1}^{D}$ are pairwise distinct and finite in number, we have

$$\Delta \; := \; \min_{i \neq j} \left\| x^{(i)} - x^{(j)} \right\|_2 \; > \; 0.$$

Set $R_0 := \tfrac{1}{2} \Delta$. If $i \neq j$ and $x \in B(x^{(i)}, R_0)$, by the triangle inequality, we have

$$\| x - x^{(j)} \|_2 \; \geq \; \| x^{(i)} - x^{(j)} \|_2 - \| x - x^{(i)} \|_2 \; > \; \Delta - R_0 \; = \; R_0,$$

so $x \notin B(x^{(j)}, R_0)$. Hence the balls are pairwise disjoint, proving the first part.

For the second part, fix $i$ and define the compact set $K_i := [0,1]^d \setminus B(x^{(i)}, R_0)$. The continuity of $f_i$ implies that the minimum

$$m_i \; := \; \min_{x \in K_i} f_i(x)$$

is attained on $K_i$. Because $x^{(i)} \notin K_i$ and $x^{(i)}$ is the unique global minimizer on $[0,1]^d$, we have $m_i > z_i$. Let $\varepsilon_i := m_i - z_i > 0$ and set

$$\varepsilon_0 \; := \; \tfrac{1}{2} \min_{1 \leq i \leq D} \varepsilon_i \; > \; 0.$$

| Notation flow (dependency structure) | Meaning |
|---|---|
| $x^{(i)}$ | Point where $f_i$ achieves minimum |
| $\quad \to B(x^{(i)}, r)$ | Basin region for retrieval coordinate $i$ |
| | (Basin around $x^{(i)}$) |
| $\quad\quad \to G_i, K_i$ | Monotone local segment near $x^{(i)}$ |
| | (In $B(x^{(i)}, r)$) |
| | |
| $P_0$ | The set of all candidate points. |
| | (We only choose $x_t \in P_0$) |
| $S_i$ | Index partition for retrieval coordinate $i$ |
| | ($i = 1, \ldots, D$) |
| | |
| Attention head $j$ | Defines response at position $t$ |
| $\quad \to \lambda_j(x, t)$ | Attention score |
| $\quad\quad \to (y_j, t_j)$ | Maximum-attention point selected by |
| | head $j$ in $P_0 \times S_j$ |
| $\quad\quad \to Y = \{y_1, \ldots, y_s\}$ | Chosen maximizers of attention score |
| | (one per head) |
| $\quad \to v_j(x, t)$ | Value embedding |
| | |
| WLOG, suppose $K_1 \cap Y = \emptyset$. | |
| $\quad \to T_0$ | $T_0 \subset S_1$, indices not in $(y_j, t_j)$, $j = 1, \ldots, s$ |
| $\quad\quad \to \eta : [0, 1] \to G_1$ | Coordinate system on $G_1$ |
| $\quad\quad\quad \to q = f_1 \circ \eta$ | Rewriting $f_1\|_{G_1}$ into the coordinate system. |
| $\quad\quad \to U_t$ | Discrete grid on $[0, 1]$ at index $t$ |
| $\quad\quad\quad \to z_\ell(t)$ | Candidate point for subsequence $\ell$, $z_\ell(t) \in \eta(U_t)$ |
| | |
| Adversarial subsequences | |
| $\quad \to Z_\ell = (z_\ell(1), \ldots, z_\ell(T_0))$ | Two subsequences almost |
| | indistinguishable by attention head. |
| $\quad\quad \to W_\ell$ | Full sequence embedding $Z_\ell$ |
| $\quad\quad\quad \to w_\ell(t)$ | Token of $W_\ell$ of index $t$ |
| $\quad\quad\quad \to I_1, I_2, I_3$ | Partition of indices: differ / agree / remaining |
| | |
| Per-head analysis | |
| $\quad \to Q_{j,i}$ | Attention mass on $I_j$ ($j \in \{1, 2, 3\}, i \in \{1, \ldots, s\}$) |
| $\quad \to V_{j,i}$ | Weighted value average on $I_j$ |

Table 1: Flow-style dependency map of notation introduced in the proof of Theorem 2.2.

Then, for every $x \in K_i$, we have

$$f_i(x) \ \geq \ m_i \ = \ z_i + \varepsilon_i \ \geq \ z_i + 2\varepsilon_0 \ > \ z_i + \varepsilon_0,$$

Thus, we have proved this lemma. $\qquad\square$

Before the proof, We also provide a notation table to help with understanding in Table 1.

*Proof of Theorem 2 (2).* Given the target function under the assumptions. For any given single-layer transformer defined in the main context, our goal is to find two different sequences such that their output in the part

$$\text{Concat}_{i=1}^{h}\Big(\sum_{t=1}^{T} \sigma\big[(W_{K,i}\hat{x}(t))^\top W_{Q,i}\hat{c}_0\big] W_{V,i}\hat{x}(t)\Big) \tag{20}$$

are very close (differs by only $O(\epsilon^{k+1})$), but their output from the target function differs by at least $3\epsilon$, then according to lemma 4, we have the required parameter count for the FFN $\hat{F}$ to be at least $\Omega(1/\epsilon^k)$.

**Notations** For each head $i = 1, \ldots, s$, define the attention weight function

$$\lambda_i(x, t) = \exp\big(\gamma \, (W_{K,i} P_\phi(x, t))^\top W_{Q,i} \hat{c}_0\big),$$

and the value mapping

$$v_i(x, t) = W_{V,i} P_\phi(x, t) \in \mathbb{R}^n,$$

where $\gamma > 0$ is the softmax scaling factor, $W_{Q,i}, W_{K,i} \in \mathbb{R}^{n \times E}$ are the query and key projections, and $W_{V,i} \in \mathbb{R}^{n \times E}$ is the value projection for head $i$.

**Notation of sets** Without loss of generality, we assume $x^{(i)}$ belongs to the interior of $[0, 1]^d$, and the other case can be treated with the same method below. From lemma 5, lemma 6 and lemma 7 we have that there exists $R > 0$ and segments $G_i \subset B(x^{(i)}, R), i = 1, \ldots, D$ and $L, \delta_0, r > 0$ satisfying the following:

- $\forall i, \forall x, y \in G_i$, we have $|f_i(x) - f_i(y)| > L \|x - y\|_2$.

- $\forall j \neq i$ and $\forall x \in B(x^{(j)}, R), y \in B(x^{(i)}, R)$, we have $f_i(y) - f_i(x) > \delta_0$.

- The length of $G_i$ is $r$, $\forall i = 1, \ldots, D$.

- For any $i \in \{1, \ldots, D\}$ and any perturbation $\delta_1$ with $|\delta_1| < \max_{x \in B(x^{(i)}, R)}(f_i(x) - z_i)$,

$$\big|F_0(z_1, \ldots, z_i + \delta_1, \ldots, z_D) - F_0(z_1, \ldots, z_D)\big| \geq L \, |\delta_1|.$$

We denote by $K_i := G_i \cup \{x^{(i)}\}, i = 1, \ldots, D$, and $P_0 = \cup_{i=1}^D K_i$. Recall that $k = \frac{\frac{1}{4}T - s - D + 1}{(n+1)s+1} - 1$. We assume without loss of generality that $k > 0$ and $\frac{1}{4}T - s - D + 1 > 0$, otherwise the result would be trivial.

**Max weight for each head** For $j = 1, \ldots, s$, define recursively the pairs $(y_j, t_j)$ as follows:

- For the first head,

$$(y_1, t_1) = \arg \max_{\substack{y \in P_0 \\ t \in S_1}} \lambda_1(y, t).$$

- For $j > 1$,

$$(y_j, t_j) = \arg \max_{\substack{y \in P_0 \\ t \in S_j \\ t \notin \{t_1, \ldots, t_{j-1}\}}} \lambda_j(y, t).$$

- If maximum can be obtained at multiple $(y, t)$, then choose one of them.

Let $Y = \{y_1, \ldots, y_s\}$. Since the sets $K_1, \ldots, K_D$ are pairwise disjoint and $s < D$, there exists at least one index $i \in \{1, \ldots, D\}$ such that

$$K_i \cap Y = \varnothing.$$

Without loss of generality, we assume that $i = 1$. As we have $|S_i| \geq \frac{1}{4}T > s + D - 1, i = 1, \ldots, D$, we have that there exists a set of $(t_2^*, \ldots, t_D^*)$ such that

- $t_j^* \notin \{t_1, \ldots, t_s\}$, for $j = 2, \ldots, D$.

- $t_j^*$ are pairwise distinct.

- $t_j^* \in S_j, j = 2, \ldots, D$.

Let $T_0 = \frac{1}{4}T - s - D + 1 > 0$ and assume that $T_0$ is a integer. Then we have $|S_1 - \{t_1, \ldots, t_s, t_2^*, \ldots, t_D^*\}| \geq T_0 > 0$. Without loss of generality, suppose $\{1, 2, \ldots, T_0\} \subset S_1 - \{t_1, \ldots, t_s, t_2^*, \ldots, t_D^*\}$.

**Sequences to be considered**  As $G_1$ is a segment of length $r$, then it is natural to assign coordinate system $\eta : [0,1] \to G_1$ on $G_1$, with $q := f_1 \circ \eta$ being a monotonically increasing function on $[0,1]$. The monotone property is as a result of $|f_1(x) - f_1(y)| \geq L\|x - y\|_2$.

We denote by $M = T_0 \lfloor \frac{rL^2}{3T_0 \epsilon} \rfloor$. As $T_0 | M$, Construct the following $T_0$ sets:

$$U_j = \frac{j-1}{T_0} + \{\frac{1}{M}, \ldots, \frac{(\frac{T_0}{M})}{M}\}, j = 1, \ldots, T_0 \tag{21}$$

We have $|U_j| = \lfloor \frac{rL^2}{3T_0 \epsilon} \rfloor = O(1/\epsilon)$.

*Claim* 2.1.  Existence of two distinct sub-sequence

There exists two subsequences $z_1(1), \ldots, z_1(T_0)$ and $z_2(1), \ldots z_2(T_0)$ with $z_i(t) \in \eta(U_t)$ satisfying the following conditions.

- $$\left\| \frac{\sum_{t=1}^{T_0} \lambda_i(z_1(t),t) v_i(z_1(t),t)}{\sum_{t=1}^{T_0} \lambda_i(z_1(t),t)} - \frac{\sum_{t=1}^{T_0} \lambda_i(z_2(t),t) v_i(z_2(t),t)}{\sum_{t=1}^{T_0} \lambda_i(z_2(t),t)} \right\|_2 \leq \frac{\epsilon^{k+1}}{3T_0}, \text{ for } i = 1, \ldots, s.$$

  For each $i = 1, \ldots, s$, either of the following holds:

  1. $\frac{\sum_{t=1}^{T_0} \lambda_i(z_1(t),t)}{\sum_{t=1}^{T_0} \lambda_i(z_2(t),t)} \in \left[ 1/(1 + \frac{\epsilon^{k+1}}{12T_0^2}), \, 1 + \frac{\epsilon^{k+1}}{12T_0^2} \right]$.
  2. $\max_{j=1,2} \sum_{t=1}^{T_0} \lambda_i(z_j(t),t) \leq \frac{\epsilon^{k+1}}{4} \sum_{w=1}^{s} \lambda_i(y_w, t_w)$.

*Proof.*  We compare the orders of $1/\epsilon$ appearing on both sides of the conditions.

First, since $|U_t| = O(1/\epsilon)$ for each $t$, the total number of possible choices of subsequences $(z(1), \ldots, z(T_0))$ is at most $O(1/\epsilon^{T_0})$.

Next, to satisfy condition (1), note that both vectors involved are $n$-dimensional with norms bounded by 1. Thus, the discretization required to achieve accuracy $\epsilon^{k+1}/(3T_0)$ in the $\ell_2$ norm leads to at most $O(1/\epsilon^{(k+1)ns})$ distinct possibilities, since there are $s$ heads.

For condition (2), observe that

$$\sum_{w=1}^{s} \lambda_i(y_w, t_w) \geq \frac{1}{T_0} \max_{j=1,2} \sum_{t=1}^{T_0} \lambda_i(z_j(t),t).$$

Hence, for each $i$, the relevant interval can be partitioned into at most $O\left( \frac{-\log \epsilon}{\epsilon^{k+1}} \right)$ sub-intervals.

Taken across $s$ heads, this contributes at most $O\left( (-\log \epsilon)^s / \epsilon^{(k+1)s} \right)$ possibilities.

Combining the two conditions, the total number of distinct admissible cases is bounded above by

$$O\left( \frac{(-\log \epsilon)^s}{\epsilon^{(k+1)ns + (k+1)s}} \right).$$

Since $T_0 \geq (k+1)ns + (k+1)s + 1$, we have

$$O\left( \frac{1}{\epsilon^{T_0}} \right) \gg O\left( \frac{(-\log \epsilon)^s}{\epsilon^{(k+1)ns + (k+1)s}} \right).$$

Therefore, by the pigeonhole principle, there must exist two distinct subsequences $(z_1(1), \ldots, z_1(T_0))$ and $(z_2(1), \ldots, z_2(T_0))$ satisfying all the conditions of Claim 2.1. $\qquad \square$

**Construction of Distinct sequences**  From Claim 2.1, we have constructed two sub-sequences $Z_1, Z_2$ satisfying the given conditions. We now consider the construction of two full input sequence $W_1, W_2$:

- For $t = 1, \ldots, T_0$, if $z_1(t) = z_2(t)$, then $w_1(t) = w_2(t) = x^{(D)}$. Otherwise, $w_1(t) = z_1(t), w_2(t) = z_2(t)$.

- $w_j(t_i) = y_i, i = 1, \ldots, s; \quad j = 1, 2$.

- $w_j(t_i^*) = x^{(i)}, i = 2, \ldots, D; \quad j = 1, 2$.

- For all other $t$, $w_j(t) = x^{(D)}$.

**Difference of** $W1, W2$ **applied to target function** Denote by $I_1$ the set of all indices $t$ with $z_1(t) \neq z_2(t)$, and $I_2 = [T_0] - I_1$, $I_3 = [T] - I_1$. It is clear from the difference of $Z_1, Z_2$ that $I_1 \neq \varnothing$.

We then define the following notations for the simplicity of calculation (defined for each head $i = 1, \ldots, s$):

- $Q_{1,i} = \sum_{t \in I_1} \lambda_i(w_1(t), t)$.

- $Q_{2,i} = \sum_{t \in I_1} \lambda_i(w_2(t), t)$.

- $V_{1,i} = (\sum_{t \in I_1} \lambda_i(w_1(t), t) v_i(w_1(t), t))/Q_{1,i}$.

- $V_{2,i} = (\sum_{t \in I_1} \lambda_i(w_2(t), t) v_i(w_2(t), t))/Q_{2,i}$.

- $Q_{3,i} = \sum_{t \in I_2} \lambda_i(z_1(t), t)$, which is also the same if defined on $Z_2$.

- $V_{3,i} = (\sum_{t \in I_2} \lambda_i(z_1(t), t) v_i(z_1(t), t))/Q_{3,i}$, which is the same if defined on $Z_2$.

- $Q_{4,i} = \sum_{t \in I_3} \lambda_i(w_1(t), t)$, which is the same if defined on $W_2$.

- $V_{4,i} = (\sum_{t \in I_3} \lambda_i(w_1(t), t) v_i(w_1(t), t))/Q_{4,i}$, which is the same if defined on $W_2$.

As $\lambda_i()$ maps to positive values, $V_{j,i}$ are convex combinations of $v_i()$, whose norm is bounded by $1$ according to the constraint section 1. Therefore $\|V_{j,i}\| \leq 1$, $j = 1, 2, 3, 4$.

As $f_1 \circ \eta$ is monotone on $[0, 1]$, let $\tilde{t} = \max_{t \in I_1} t$, then we have

- $\max_{t \in S_1} f_1(w_1(t)) = f_1(w_1(\tilde{t}))$.

- $\max_{t \in S_1} f_1(w_2(t)) = f_1(w_2(\tilde{t}))$.

And by construction we know that

$$\|(w_1(\tilde{t})) - (w_2(\tilde{t}))\| \geq \frac{r}{M} \tag{22}$$

which is the minimal distance for any two points in $U_{\tilde{t}}$. Then we have

$$|f_1(w_1(\tilde{t})) - f_1(w_2(\tilde{t}))| \geq \frac{rL}{M} \tag{23}$$

As we have for $i = 2, \ldots, D$

- $\max_{t \in S_i} f_i(w_1(t)) = z^{(i)}$.

- $\max_{t \in S_i} f_i(w_2(t)) = z^{(i)}$.

Then following the perturbation property of $F_0$ defined above we have that the difference of output between the two sequence to be at least $\frac{rL^2}{M}$, which is greater than $3\epsilon$. Then $\epsilon$-approximation requires that $|Model(W_1) - Model(W_2)| \geq \epsilon$.

$W_1$ **and** $W_2$ **are close after attention layer** For any given head $i$, we consider the the two cases given in 2.1.

**Case 1** Case 1 can be rewritten as follows:

- $\|\frac{Q_{1,i}V_{1,i} + Q_{3,i}V_{3,i}}{Q_{1,i} + Q_{3,i}} - \frac{Q_{2,i}V_{2,i} + Q_{3,i}V_{3,i}}{Q_{2,i} + Q_{3,i}}\|_2 \leq \frac{\epsilon^{k+1}}{3T_0}$.

- $\frac{Q_{1,i} + Q_{3,i}}{Q_{2,i} + Q_{3,i}} \in \left[1/(1 + \frac{\epsilon^{k+1}}{12T_0^2}), 1 + \frac{\epsilon^{k+1}}{12T_0^2}\right]$.

Without loss of generality, we assume $Q_{1,i} \geq Q_{2,i}$. By calculation, we have

$$\frac{Q_{1,i}V_{1,i} + Q_{3,i}V_{3,i}}{Q_{1,i} + Q_{3,i}} - \frac{Q_{2,i}V_{2,i} + Q_{3,i}V_{3,i}}{Q_{2,i} + Q_{3,i}} \tag{24}$$

$$= \frac{Q_{3,i}(Q_{2,i} - Q_{1,i})(V_{3,i} - V_{2,i})}{(Q_{1,i} + Q_{3,i})(Q_{2,i} + Q_{3,i})} + \frac{Q_{1,i}}{Q_{1,i} + Q_{3,i}}(V_{1,i} - V_{2,i}). \tag{25}$$

We have already known that $Q_{4,i} \geq \frac{Q_{1,i} + Q_{2,i}}{T_0}$ (As $Q_{4,i}$ has the max weight of each head in it). Then

$$\left\| \frac{Q_{4,i}(Q_{2,i} - Q_{1,i})(V_{4,i} - V_{2,i})}{(Q_{1,i} + Q_{4,i})(Q_{2,i} + Q_{4,i})} \right\| \tag{26}$$

$$\leq \left\| \frac{(Q_{2,i} - Q_{1,i})(V_{4,i} - V_{2,i})}{(Q_{1,i} + Q_{4,i})} \right\| \tag{27}$$

$$\leq \left\| \frac{T_0(Q_{2,i} - Q_{1,i})(V_{4,i} - V_{2,i})}{(Q_{1,i} + Q_{3,i})} \right\| \tag{28}$$

$$\leq \left\| \frac{T_0 \epsilon^{k+1}(V_{4,i} - V_{2,i})}{12T_0^2} \right\| \tag{29}$$

$$\leq \frac{\epsilon^{k+1}}{6T_0} \tag{30}$$

Similarly, we also have

$$\left\| \frac{Q_{3,i}(Q_{2,i} - Q_{1,i})(V_{3,i} - V_{2,i})}{(Q_{1,i} + Q_{3,i})(Q_{2,i} + Q_{3,i})} \right\| \leq \frac{\epsilon^{k+1}}{6T_0} \tag{31}$$

From inequality 26 and substituting equation 24, we have

$$\left\| \frac{Q_{1,i}}{Q_{1,i} + Q_{3,i}}(V_{1,i} - V_{2,i}) \right\| \leq \frac{\epsilon^{k+1}}{6T_0} + \frac{\epsilon^{k+1}}{3T_0} = \frac{\epsilon^{k+1}}{2T_0} \tag{32}$$

Therefore

$$\left\| \frac{Q_{1,i}}{Q_{1,i} + Q_{4,i}}(V_{1,i} - V_{2,i}) \right\| \tag{33}$$

$$\leq \left\| \frac{T_0 Q_{1,i}}{Q_{1,i} + Q_{3,i}}(V_{1,i} - V_{2,i}) \right\| \tag{34}$$

$$\leq \frac{\epsilon^{k+1}}{2} \tag{35}$$

Thus

$$\left\| \frac{Q_{1,i}V_{1,i} + Q_{4,i}V_{4,i}}{Q_{1,i} + Q_{4,i}} - \frac{Q_{2,i}V_{2,i} + Q_{4,i}V_{4,i}}{Q_{2,i} + Q_{4,i}} \right\|_2 \tag{36}$$

$$\leq \left\| \frac{Q_{4,i}(Q_{2,i} - Q_{1,i})(V_{4,i} - V_{2,i})}{(Q_{1,i} + Q_{4,i})(Q_{2,i} + Q_{4,i})} \right\| + \left\| \frac{Q_{1,i}}{Q_{1,i} + Q_{4,i}}(V_{1,i} - V_{2,i}) \right\| \tag{37}$$

$$\leq \frac{\epsilon^{k+1}}{6T_0} + \frac{\epsilon^{k+1}}{2} \tag{38}$$

$$\leq \epsilon^{k+1} \tag{39}$$

**Case 2** Case 2 can be rewritten as follows:

- $\left\| \frac{Q_{1,i}V_{1,i} + Q_{3,i}V_{3,i}}{Q_{1,i} + Q_{3,i}} - \frac{Q_{2,i}V_{2,i} + Q_{3,i}V_{3,i}}{Q_{2,i} + Q_{3,i}} \right\|_2 \leq \frac{\epsilon^{k+1}}{3T_0}$.

- $Q_{1,i} + Q_{3,i} \leq \frac{\epsilon^{k+1}}{4} \sum_{w=1}^{s} \lambda_i(y_w, t_w) \leq \frac{\epsilon^{k+1}}{4} Q_{4,i}$.

- $Q_{2,i} + Q_{3,i} \leq \frac{\epsilon^{k+1}}{4} \sum_{w=1}^{s} \lambda_i(y_w, t_w) \leq \frac{\epsilon^{k+1}}{4} Q_{4,i}$.

Thus

$$\|\frac{Q_{1,i}V_{1,i} + Q_{4,i}V_{4,i}}{Q_{1,i} + Q_{4,i}} - \frac{Q_{2,i}V_{2,i} + Q_{4,i}V_{4,i}}{Q_{2,i} + Q_{4,i}}\|_2 \tag{40}$$

$$\leq \|\frac{Q_{4,i}(Q_{2,i} - Q_{1,i})(V_{4,i} - V_{2,i})}{(Q_{1,i} + Q_{4,i})(Q_{2,i} + Q_{4,i})}\| + \|\frac{Q_{1,i}}{Q_{1,i} + Q_{4,i}}(V_{1,i} - V_{2,i})\| \tag{41}$$

$$\leq \|\frac{(Q_{2,i} - Q_{1,i})(V_{4,i} - V_{2,i})}{(Q_{2,i} + Q_{4,i})}\| + \|\frac{Q_{1,i}}{Q_{1,i} + Q_{4,i}}(V_{1,i} - V_{2,i})\| \tag{42}$$

$$\leq \|\frac{\epsilon^{k+1}(V_{4,i} - V_{2,i})}{4}\| + \|\frac{\epsilon^{k+1}}{4}(V_{1,i} - V_{2,i})\| \tag{43}$$

$$\leq \epsilon^{k+1} \tag{44}$$

And it can be seen from definition that

- $\frac{Q_{1,i}V_{1,i} + Q_{4,i}V_{4,i}}{Q_{1,i} + Q_{4,i}}$ is the output of the $i$-th head of the attention layer with input sequence $W_1$. (which means $\frac{Q_{1,i}V_{1,i} + Q_{4,i}V_{4,i}}{Q_{1,i} + Q_{4,i}} = \sum_{t=1}^{T} \sigma[(W_{K,i}\hat{w}_1(t))^{\top}W_{Q,i}\hat{c}_0] W_{V,i}\hat{w}_1(t)$).

- $\frac{Q_{2,i}V_{1,i} + Q_{4,i}V_{4,i}}{Q_{2,i} + Q_{4,i}}$ is the output of the $i$-th head of the attention layer with input sequence $W_2$. (which means $\frac{Q_{2,i}V_{2,i} + Q_{4,i}V_{4,i}}{Q_{2,i} + Q_{4,i}} = \sum_{t=1}^{T} \sigma[(W_{K,i}\hat{w}_2(t))^{\top}W_{Q,i}\hat{c}_0] W_{V,i}\hat{w}_2(t)$).

Then for each $i = 1, \ldots, s$, we have that

$$\|\sum_{t=1}^{T} \sigma[(W_{K,i}\hat{w}_1(t))^{\top}W_{Q,i}\hat{c}_0] W_{V,i}\hat{w}_1(t) - \sum_{t=1}^{T} \sigma[(W_{K,i}\hat{w}_2(t))^{\top}W_{Q,i}\hat{c}_0] W_{V,i}\hat{w}_2(t)\| \leq \epsilon^{k+1} \tag{45}$$

Therefore, as $W_O$ have entries bounded by 1, we have

$$\|\left[\hat{c}_0 + W_O \text{Concat}_{i=1}^{h}\left(\sum_{t=1}^{T} \sigma[(W_{K,i}\hat{w}_1(t))^{\top}W_{Q,i}\hat{c}_0] W_{V,i}\hat{w}_1(t)\right)\right] \tag{46}$$

$$-\left[\hat{c}_0 + W_O \text{Concat}_{i=1}^{h}\left(\sum_{t=1}^{T} \sigma[(W_{K,i}\hat{w}_2(t))^{\top}W_{Q,i}\hat{c}_0] W_{V,i}\hat{w}_2(t)\right)\right]\| \tag{47}$$

$$\leq s\epsilon^{k+1} \tag{48}$$

However, it has been proven above that we need $|Model(W_1) - Model(W_2)| \geq \epsilon$ to achieve $\epsilon$-approximation of the target function. According to lemma 4, the required parameter count of the FFN $\hat{F}$ is of order $\Omega(\epsilon/\epsilon^{k+1})$. Thus the parameter count required to achieve $\epsilon$-approximation is $\Omega(1/\epsilon^k)$.

$\square$

*Remark.* **Tightness of Theorem 2 (2)** The lower bound in Theorem 2 (2) remains essentially tight under several relaxations of the feed-forward block $\hat{F}$. If $\hat{F}$ uses Heaviside activations instead of 1-Lipschitz activations, matching upper bounds can be constructed, but this case is impractical since Heaviside activations are rarely used in practice. If parameter norms are permitted to scale as $O(T^{1/\epsilon})$, the parameter count can be reduced to $O(1/\epsilon^{\gamma+1})$, though this scenario is likewise unrealistic in practical settings. Finally, if $\hat{F}$ is allowed up to five layers, the lower bound changes to $\Omega(1/\epsilon^{k/4})$, which does not alter the qualitative conclusion.

### A.3 PROOF OF THEOREM 2 (3)

**Proof Sketch.** The argument is based on an explicit construction. We begin with trivial attention, so that the post-attention output is simply the averaged concatenation $\frac{1}{T}(x(1), \ldots, x(T)) \in \mathbb{R}^{Td}$. The feed-forward block can then be used to compute the transformations $f_i(x(t))$, perform the necessary comparisons, and approximate $F_0$, as ensured by Lemmas 9 and 8.

Having outlined the main idea, we now proceed to the detailed proof. As a first step, we introduce several auxiliary lemmas that will be used in the argument.

**Lemma 8.** *Fix a pointwise activation $\sigma$ (e.g., ReLU or any activation used in this paper). Let $F_1 : \mathbb{R}^{m_1} \to \mathbb{R}^{m_2}$ be a 2-layer fully connected network, $F_2 : \mathbb{R}^{m_2} \to \mathbb{R}^{m_3}$ a 3-layer fully connected network, and $F_3 : \mathbb{R}^{m_3} \to \mathbb{R}$ a 2-layer fully connected network. Let $W_1, W_2, W_3$ denote their respective (maximum) hidden widths, and set $W := \max\{W_1, W_2, W_3\}$. Then there exists a 5-layer fully connected network $G : \mathbb{R}^{m_1} \to \mathbb{R}$ with activation $\sigma$ and hidden width at most $W$ such that*

$$G(x) = F_3\big(F_2\big(F_1(x)\big)\big) \qquad \text{for all } x \in \mathbb{R}^{m_1}.$$

*Proof.* Proof of Lemma 8 Write the three networks in affine–nonlinearity form (with a pointwise activation $\sigma$):

$$
\begin{aligned}
F_1(x) &= A_1\,\sigma(B_1 x + b_1) + a_1, & x &\in \mathbb{R}^{m_1},\ F_1(x) \in \mathbb{R}^{m_2}, \\
F_2(u) &= C_2\,\sigma\big(D_2\,\sigma(E_2 u + e_2) + d_2\big) + c_2, & u &\in \mathbb{R}^{m_2},\ F_2(u) \in \mathbb{R}^{m_3}, \\
F_3(v) &= p_3\,\sigma(Q_3 v + q_3) + r_3, & v &\in \mathbb{R}^{m_3},\ F_3(v) \in \mathbb{R}.
\end{aligned}
$$

Define a 5-layer fully connected network $G : \mathbb{R}^{m_1} \to \mathbb{R}$ by stacking the hidden layers of $F_1$ (one), $F_2$ (two), and $F_3$ (one), keeping their original widths:

$$
\begin{aligned}
h_1(x) &:= \sigma(B_1 x + b_1), \\
u(x) &:= A_1 h_1(x) + a_1, \\
h_2(x) &:= \sigma(E_2 u(x) + e_2), \\
h_3(x) &:= \sigma(D_2 h_2(x) + d_2), \\
v(x) &:= C_2 h_3(x) + c_2, \\
h_4(x) &:= \sigma(Q_3 v(x) + q_3), \\
G(x) &:= p_3 h_4(x) + r_3.
\end{aligned}
$$

By construction,

$$G(x) = p_3\,\sigma\Big(Q_3\big(C_2\,\sigma(D_2\,\sigma(E_2(A_1\,\sigma(B_1 x + b_1) + a_1) + e_2) + d_2) + c_2\big) + q_3\Big) + r_3 = F_3\big(F_2\big(F_1(x)\big)\big).$$

Thus $G$ realizes the composition exactly, has 4 hidden layers (hence 5 layers total), and its hidden widths are precisely those of the constituent hidden layers of $F_1$, $F_2$, and $F_3$. $\qquad\square$

**Lemma 9** (Approximating max with a shallow ReLU network). *Let $f : [0,1]^T \to \mathbb{R}$ be $f(x_1, \dots, x_T) = \max\{x_1, \dots, x_T\}$. For any $\epsilon \in (0,1]$, there exists a fully connected ReLU network $\hat{f}$ with three layers (i.e., two hidden layers and one output layer), whose hidden-layer widths are each at most $2T\lceil 1/\epsilon \rceil$, such that $\hat{f}$ $\epsilon$-approximates $f$.*

*Proof.* Proof of Lemma 9 Let

$$n = \lceil 1/\epsilon \rceil.$$

For each coordinate $t \in [T]$ and each grid index $i = 0, 1, \dots, n-1$, define the first hidden layer neurons by

$$h_1(t, i) = \mathrm{ReLU}\big(x_t - \tfrac{i}{n}\big).$$

For each $j = 0, 1, \dots, n-1$, define the second hidden layer neurons by

$$h_2(j) = \mathrm{ReLU}\left(\sum_{t=1}^{T} h_1(t, j)\right) - \mathrm{ReLU}\left(\sum_{t=1}^{T} h_1(t, j) - \tfrac{1}{n}\right).$$

Finally, the output of the network is given by

$$\hat{f}(x_1, \dots, x_T) = \sum_{j=0}^{n-1} h_2(j).$$

*Claim.* Fix $j \in \{0, \ldots, n-1\}$ and set

$$S_j = \sum_{t=1}^{T} h_1(t, j) = \sum_{t=1}^{T} \text{ReLU}\left(x_t - \frac{j}{n}\right).$$

By definition,

$$h_2(j) = \text{ReLU}(S_j) - \text{ReLU}\left(S_j - \frac{1}{n}\right).$$

1) If $h_2(j) > 0$, then necessarily $S_j > 0$ (since $\text{ReLU}(z) > 0$ iff $z > 0$), hence there exists some $t$ with

$$\text{ReLU}\left(x_t - \frac{j}{n}\right) > 0 \quad \Longleftrightarrow \quad x_t > \frac{j}{n}.$$

Thus $h_2(j) > 0$ only if $\exists t$ with $x_t > j/n$.

2) If there exists $t$ with $x_t > (j+1)/n$, then

$$S_j \geq \text{ReLU}\left(x_t - \frac{j}{n}\right) > \frac{1}{n}.$$

Therefore $S_j \geq \frac{1}{n}$, and we get

$$h_2(j) = S_j - \left(S_j - \frac{1}{n}\right) = \frac{1}{n}.$$

$\square$

Fix $x \in [0, 1]^T$ and let $j$ be such that $\max_t x_t \in (j/n, (j+1)/n]$. By construction,

$$h_2(k) = 0 \quad \text{for } k \geq j+1, \qquad h_2(k) = \frac{1}{n} \quad \text{for } k \leq j-1,$$

and for $k = j$ we have

$$0 \leq h_2(j) = \text{ReLU}(S_j) - \text{ReLU}\left(S_j - \frac{1}{n}\right) \leq \frac{1}{n}, \quad S_j := \sum_{t=1}^{T} \text{ReLU}\left(x_t - \frac{j}{n}\right).$$

Hence

$$\hat{f}(x) = \sum_{k=0}^{n-1} h_2(k) = \sum_{k=0}^{j-1} \frac{1}{n} + h_2(j) \in \left[\frac{j}{n}, \frac{j}{n} + \frac{1}{n}\right] = \left[\frac{j}{n}, \frac{j+1}{n}\right].$$

Since $\max_t x_t \in (j/n, (j+1)/n]$, it follows that

$$0 \leq |\hat{f}(x) - \max_t x_t| \leq \frac{1}{n} \leq \epsilon.$$

Therefore $\hat{f}$ $\epsilon$-approximates $f(x) = \max_t x_t$ on $[0, 1]^T$.

$\square$

*Proof.* Theorem 2 (3)
We begin by fixing the embedding with positional information. Let $P_\phi : [0, 1]^d \times [T] \to \mathbb{R}^{dT}$ be defined by

$$P_\phi(x(t), t) = (0, \ldots, 0, x(t), 0, \ldots, 0),$$

where the vector $x(t)$ occupies the $t$-th block of dimension $d$, and all other blocks are zero. With the classification token $\hat{c}_0 = 0$, the attention layer reduces to a trivial aggregation, and the output (prior to the feed-forward network) is

$$\frac{1}{T} (x(1), \ldots, x(T)) \in [0, 1]^{dT}.$$

Given a target accuracy $\epsilon > 0$, we construct three feed-forward networks $F_1, F_2, F_3$ as follows.

**Step 1: Approximating the component functions.** Define

$$F_1 : \frac{1}{T}[0, 1]^{d \times T} \to \mathbb{R}^{D \times T}, \qquad F_1\left(\frac{1}{T}x(1), \ldots, \frac{1}{T}x(T)\right) = (u(1), \ldots, u(T)),$$

where each $u(t) \in \mathbb{R}^D$ satisfies

$$|u(t)_i - f_i(x(t))| \leq \epsilon \quad \text{for all } i = 1, \ldots, D.$$

By Assumption 3, such an approximation can be implemented by a two-layer FFN with parameter count $O(1/\epsilon^\gamma)$.

**Step 2: Approximating the minimization.** Let $F_2' : \mathbb{R}^{D \times T} \to \mathbb{R}^D$ be defined by

$$F_2'(u(1), \ldots, u(T)) = (u_1, \ldots, u_D), \qquad u_i = \min_{t \in S_i} u(t)_i.$$

By Lemma 9 (which works the same for taking minimum), there exists a three-layer ReLU network with $O(1/\epsilon)$ parameters that $\epsilon$-approximates $F_2'$. We denote this approximation by $F_2$.

**Step 3: Approximating the outer function.** Finally, let $F_3 : \mathbb{R}^D \to \mathbb{R}$ be a two-layer FFN that $\epsilon$-approximates $F_0$, with parameter count $O(1/\epsilon^\gamma)$.

**Composition.** Since $F_0$ is $C^1$ on a compact domain, it is Lipschitz with constant $L$, and the $\min$ operator is 1-Lipschitz. Therefore, the composed network

$$F_3 \circ F_2 \circ F_1$$

provides an $L\epsilon$-approximation of the target function, with total parameter count

$$O(1/\epsilon^{\gamma+1}).$$

According to lemma 8, $F_3 \circ F_2 \circ F_1$ can be written equivalently as a five-layer FFN.

$\square$

### A.4 PROOF OF COROLLARY 1

**Proof Sketch.** It is a direct corollary of Theorem 2 (1) and 2 (2).

*Proof.* Suppose $D_1 < D_2$. Let $M_0$ be the minimal positive integer such that $\mathcal{H}(D_1, 2, d, T, M_0)$ $\epsilon$-approximates $H$. Then with representation $(\{f_i, S_i\}_{i=1}^{D_1}, F_0)$, Theorem 2 (1) suggests that there exists a positive $C_{d,D_1,T}$ such that

$$M_0 \leq \frac{C_{d,D_1,T}}{\epsilon^\gamma} \tag{49}$$

With representation $(\{\tilde{f}_i, \tilde{S}_i\}_{i=1}^{D_2}, \tilde{F}_0)$, Theorem 2 (2) suggests that there exists a positive $C_{d,D_2,T}$ such that

$$M_0 \geq \frac{C_{d,D_2,T}}{\epsilon^k} \text{ for } k = \frac{(\frac{1}{4}T - D_1 - D_2 + 1)}{3D_1 + 1} - 1 \tag{50}$$

As $f_i$ and $F_0$ are at least $C^1$ smooth, we have $\gamma \leq \max(D_1, D_2)$. Thus with $D_1^2 + D_2^2 \leq \frac{1}{50}T$, we have $k > \gamma$. Then there exist $\epsilon > 0$ such that

$$\frac{C_{d,D_2,T}}{\epsilon^k} > \frac{C_{d,D_1,T}}{\epsilon^\gamma} \tag{51}$$

This leads to a contradiction. Thus $D_1 = D_2$.

$\square$

# B EXPERIMENT DETAILS

## B.1 DETAILS FOR EXPERIMENT 1

### B.1.1 EXPERIMENTAL DETAILS FOR SECTION 6.1

**Data generation.** The intrinsic dimension of the synthetic task is $D = 4$. For each sequence length $T \in \{8, 16, 32, 64, 128\}$ we generate 8000 training and 2000 validation examples. The inputs are i.i.d. Gaussian samples $x(t) \sim \mathcal{N}(0, I_4)$.

**Model architecture.** Each input vector $x(t)$ is first mapped to $\mathbb{R}^{8h}$ by a two-layer feed-forward network with hidden dimension $N$ and ReLU activations, ensuring a per-head embedding dimension of 8. A trainable classification token $c_0$ is appended, and no positional encoding is used since the task is permutation invariant. The sequence is processed by a single-layer multi-head attention block without residual connections or layer normalization, consistent with the theoretical setting. The output is concatenated and passed through a two-layer GELU-activated feed-forward network with hidden dimension $N$, yielding the final scalar prediction. The fixed hidden size ensures comparability of parameter counts across different $h$.

**Training protocol.** Each configuration $(h, T)$ is trained separately under multiple random seeds. To reduce the effect of optimization variance, we report the *minimal validation error* achieved across seeds. This choice isolates expressivity limitations of the architecture from randomness in training dynamics.

**Evaluation metric.** We adopt the normalized mean squared error (NMSE), defined as mean squared error divided by the variance of the targets. As $T$ increases, maxima of Gaussian samples concentrate, shrinking target variance and making trivial predictors appear competitive under raw MSE. (An intuition is that suppose the target output be $Y_T = \max_{1 \le t \le T} x_t$ with input tokens $x_t \sim \mathcal{N}(0, 1)$ independently, then $\mathrm{Var}(Y_T)$ *decreases* as $T$ increases, because $Y_T$ concentrates more tightly around its growing mean.) Normalization by variance corrects this effect and ensures comparability across lengths. NMSE is also equivalent to $1 - R^2$, where $R^2$ is the standard coefficient of determination.

**Variance across seeds.** While mean performance across seeds is also informative, reporting the minimal validation NMSE highlights the best achievable accuracy for a given architecture. This emphasizes limitations due to model capacity rather than training noise. Tables showing seed variance are included for completeness (Table 2).

*Remark.* When $h \ge D = 4$, we also observe that the validation NMSE first decreases rapidly and then increases slowly as $T$ grows. For shorter sequences, the model with enough heads can either capture the pattern through attention (Theorem 2 (1)) or rely on a memorization-based strategy with the feed-forward network (Theorem 2 (3)). Both approaches generalize reasonably well, but the memorization-based one does so less effectively. For longer sequences, memorization becomes infeasible and the model relies on attention, which generalizes better; however, longer sequences may also be more sensitive to parameterization, and the observed curve likely reflects a tradeoff between these effects. See Figure 3 in Appendix.

### B.1.2 FIGURES AND TABLES FOR SYNTHETIC EXPERIMENT 6.1

| Heads | T=8 | T=16 | T=32 | T=64 | T=128 |
|---|---|---|---|---|---|
| 1 | $7.01 \times 10^{-2} \pm 5.99 \times 10^{-2}$ | $1.09 \times 10^{-1} \pm 9.93 \times 10^{-2}$ | $1.10 \times 10^{-1} \pm 9.36 \times 10^{-2}$ | $1.14 \times 10^{-1} \pm 8.53 \times 10^{-2}$ | $1.45 \times 10^{-1} \pm 1.05 \times 10^{-1}$ |
| 2 | $7.31 \times 10^{-3} \pm 4.75 \times 10^{-4}$ | $8.41 \times 10^{-3} \pm 7.97 \times 10^{-4}$ | $9.42 \times 10^{-3} \pm 6.42 \times 10^{-4}$ | $1.31 \times 10^{-2} \pm 1.16 \times 10^{-2}$ | $1.47 \times 10^{-2} \pm 1.21 \times 10^{-2}$ |
| 3 | $6.94 \times 10^{-4} \pm 2.90 \times 10^{-4}$ | $6.40 \times 10^{-4} \pm 3.87 \times 10^{-4}$ | $9.09 \times 10^{-4} \pm 4.31 \times 10^{-4}$ | $1.31 \times 10^{-3} \pm 5.10 \times 10^{-4}$ | $1.58 \times 10^{-3} \pm 5.21 \times 10^{-4}$ |
| 4 | $6.10 \times 10^{-5} \pm 1.52 \times 10^{-4}$ | $4.36 \times 10^{-5} \pm 1.93 \times 10^{-4}$ | $4.80 \times 10^{-5} \pm 2.30 \times 10^{-4}$ | $8.75 \times 10^{-6} \pm 5.58 \times 10^{-5}$ | $5.23 \times 10^{-6} \pm 5.67 \times 10^{-6}$ |
| 5 | $3.35 \times 10^{-5} \pm 5.84 \times 10^{-5}$ | $1.10 \times 10^{-5} \pm 2.36 \times 10^{-5}$ | $4.91 \times 10^{-6} \pm 6.32 \times 10^{-6}$ | $4.19 \times 10^{-6} \pm 8.39 \times 10^{-6}$ | $3.99 \times 10^{-6} \pm 4.29 \times 10^{-6}$ |

Table 2: Error bar for synthetic dataset. NMSE(Mean ± Standard Deviation) for different sequence lengths $T$ and number of heads.

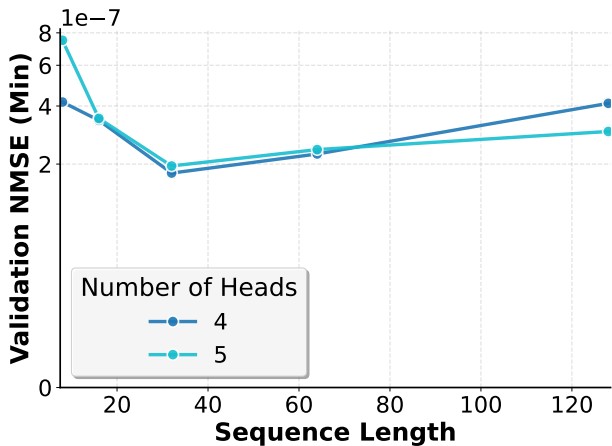

Figure 3: A zoom in plot of Figure1, which shows that when the number of head is enough, the loss first decreases and then increases, as explained in the remark B.1.1

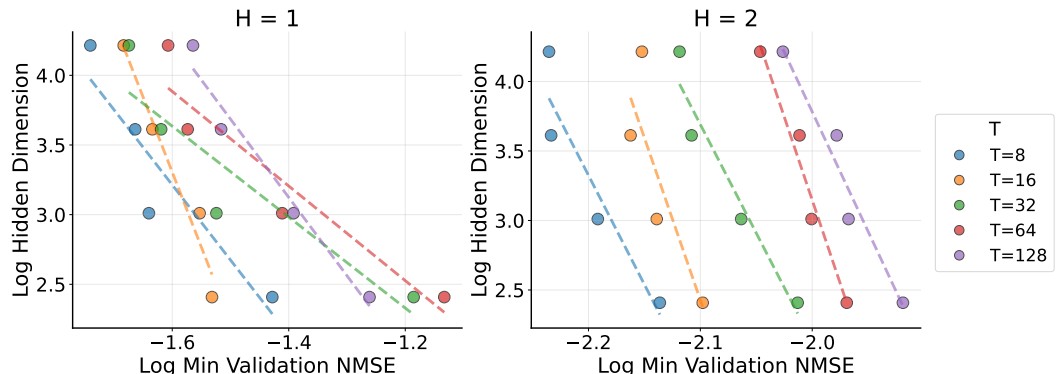

Figure 4: Additional plot of 1b for $H = 1$ and $H = 2$.

| Heads | T=8 | T=16 | T=32 | T=64 | T=128 |
|---|---|---|---|---|---|
| 1 | $1.75 \times 10^{-2}$ | $1.98 \times 10^{-2}$ | $2.06 \times 10^{-2}$ | $2.54 \times 10^{-2}$ | $3.03 \times 10^{-2}$ |
| 2 | $7.17 \times 10^{-3}$ | $7.39 \times 10^{-3}$ | $7.82 \times 10^{-3}$ | $8.57 \times 10^{-3}$ | $1.02 \times 10^{-2}$ |
| 3 | $2.11 \times 10^{-4}$ | $2.17 \times 10^{-4}$ | $2.73 \times 10^{-4}$ | $3.71 \times 10^{-4}$ | $4.77 \times 10^{-4}$ |
| 4 | $1.32 \times 10^{-6}$ | $5.59 \times 10^{-7}$ | $3.40 \times 10^{-7}$ | $3.46 \times 10^{-7}$ | $5.70 \times 10^{-7}$ |
| 5 | $2.19 \times 10^{-6}$ | $4.33 \times 10^{-7}$ | $3.22 \times 10^{-7}$ | $2.73 \times 10^{-7}$ | $2.66 \times 10^{-7}$ |

Table 3: Validation NMSE under fixed total embedding dimension $E = nh = 32$.

| Heads | T=8 | T=16 | T=32 | T=64 | T=128 |
|---|---|---|---|---|---|
| 1 | $1.38 \times 10^{-2}$ | $1.63 \times 10^{-2}$ | $1.84 \times 10^{-2}$ | $2.17 \times 10^{-2}$ | $2.31 \times 10^{-2}$ |
| 2 | $1.09 \times 10^{-3}$ | $7.08 \times 10^{-4}$ | $7.24 \times 10^{-4}$ | $7.76 \times 10^{-4}$ | $1.11 \times 10^{-3}$ |
| 3 | $4.18 \times 10^{-7}$ | $1.72 \times 10^{-7}$ | $1.17 \times 10^{-7}$ | $3.58 \times 10^{-7}$ | $2.11 \times 10^{-7}$ |
| 4 | $5.56 \times 10^{-7}$ | $1.22 \times 10^{-7}$ | $6.89 \times 10^{-8}$ | $1.85 \times 10^{-7}$ | $3.48 \times 10^{-7}$ |

Table 4: Approximation error for the $D = 3$ retrieval task under fixed total embedding dimension $E = nh$.

| Heads | T=8 | T=16 | T=32 | T=64 | T=128 |
|---|---|---|---|---|---|
| 1 | $2.12 \times 10^{-4}$ | $1.85 \times 10^{-4}$ | $2.22 \times 10^{-4}$ | $3.13 \times 10^{-4}$ | $4.28 \times 10^{-4}$ |
| 2 | $7.22 \times 10^{-6}$ | $2.69 \times 10^{-6}$ | $3.50 \times 10^{-6}$ | $3.07 \times 10^{-6}$ | $3.83 \times 10^{-6}$ |
| 3 | $8.16 \times 10^{-6}$ | $1.83 \times 10^{-6}$ | $1.73 \times 10^{-6}$ | $3.86 \times 10^{-6}$ | $3.50 \times 10^{-6}$ |
| 4 | $3.68 \times 10^{-6}$ | $1.87 \times 10^{-6}$ | $2.60 \times 10^{-6}$ | $4.32 \times 10^{-6}$ | $5.98 \times 10^{-6}$ |
| 5 | $6.15 \times 10^{-6}$ | $3.02 \times 10^{-6}$ | $3.31 \times 10^{-6}$ | $3.78 \times 10^{-6}$ | $5.34 \times 10^{-6}$ |

Table 5: Two-layer transformer on the synthetic task ($D = 4$, NoPE, NoLN, fixed total embedding dimension $E = nh = 32$).

## B.2 MS MARCO TEXT RETRIEVAL

### B.2.1 EXPERIMENT DETAILS FOR MS MARCO (TEXT RETRIEVAL) EXPERIMENT

**Dataset construction.** We construct retrieval-style datasets from the MS MARCO passage ranking collection (Bajaj et al., 2016). Since the original dataset associates each query with only a few candidate passages, we enlarge the candidate set by mining hard negatives. Specifically, BM25 (Robertson & Zaragoza, 2009) is used to mine local negatives and FAISS (Johnson et al., 2019) similarity search to retrieve global negatives, reducing redundancy across queries. For each query, the sequence length $T$ is defined as the total number of candidates (one positive and $T - 1$ negatives), with $T \in \{8, 16, 32, 64\}$. We build datasets containing 28,000 training queries and 2,000 validation queries for each $T$.

**Model and training setup.** We evaluate a two-layer Transformer encoder with per-head embedding dimension fixed at 32, while varying the number of heads across $\{1, 2, 4, 6, 8, 10, 12, 14, 16\}$. Tokenization and input embeddings follow the BERT tokenizer and frozen BERT word, position, and segment embeddings (Devlin et al., 2019), projected to the model hidden size $h = \text{heads} \times 32$. Only the projection and Transformer layers are trained. We report training top-1 accuracy, focusing on training performance since MS MARCO with BM25-mined negatives is particularly challenging for validation, and the difference can be seen in training metrics. Training MRR is also reported in Fig 5, with similar trend as training accuracy.

### B.2.2 FIGURES AND TABLES FOR EXPERIMENT

| Heads | T=8 | T=16 | T=32 | T=64 |
|---|---|---|---|---|
| 1 | $0.597 \pm 0.003$ | $0.450 \pm 0.005$ | $0.303 \pm 0.003$ | $0.154 \pm 0.002$ |
| 2 | $0.771 \pm 0.003$ | $0.647 \pm 0.003$ | $0.486 \pm 0.002$ | $0.286 \pm 0.002$ |
| 4 | $0.956 \pm 0.002$ | $0.900 \pm 0.002$ | $0.793 \pm 0.002$ | $0.580 \pm 0.004$ |
| 6 | $0.992 \pm 0.000$ | $0.977 \pm 0.001$ | $0.937 \pm 0.001$ | $0.814 \pm 0.002$ |
| 8 | $0.998 \pm 0.000$ | $0.995 \pm 0.000$ | $0.983 \pm 0.001$ | $0.932 \pm 0.002$ |
| 12 | $1.000 \pm 0.000$ | $0.999 \pm 0.000$ | $0.998 \pm 0.000$ | $0.991 \pm 0.001$ |
| 16 | $1.000 \pm 0.000$ | $1.000 \pm 0.000$ | $0.999 \pm 0.000$ | $0.996 \pm 0.000$ |

Table 6: Error bar for MS Marco dataset. Accuracy (Mean ± Standard Deviation) for different sequence lengths $T$ and number of heads.

## B.3 CIFAR-10 IMAGE CLASSIFICATION

### B.3.1 DATASET CONSTRUCTION

We create image classification datasets from the CIFAR-10 dataset using a padded preprocessing approach. The original CIFAR-10 images have dimensions of $32 \times 32$ pixels. To generate datasets with larger image sizes, we apply padding to achieve sizes in the set $\{32, 48, 64, 96, 128\}$. The original image is randomly positioned within the enlarged frame, with padding filled using the colors of the border pixels. An illustration is provided in Figure 6. By apply this padding method we are creating tasks with increasing difficulty. The background is enlarged, making models need more

| Heads | T=8 | T=16 | T=32 | T=64 |
|---|---|---|---|---|
| 1 | $0.5107 \pm 0.0069$ | $0.3917 \pm 0.0071$ | $0.2542 \pm 0.0049$ | $0.1257 \pm 0.0057$ |
| 2 | $0.5221 \pm 0.0102$ | $0.4205 \pm 0.0056$ | $0.2712 \pm 0.0067$ | $0.1369 \pm 0.0038$ |
| 4 | $0.5076 \pm 0.0112$ | $0.4048 \pm 0.0070$ | $0.2547 \pm 0.0093$ | $0.1139 \pm 0.0061$ |
| 6 | $0.5153 \pm 0.0112$ | $0.3865 \pm 0.0103$ | $0.2397 \pm 0.0098$ | $0.1018 \pm 0.0057$ |
| 8 | $0.5058 \pm 0.0082$ | $0.3801 \pm 0.0084$ | $0.2308 \pm 0.0068$ | $0.0983 \pm 0.0050$ |
| 12 | $0.5184 \pm 0.0073$ | $0.3721 \pm 0.0107$ | $0.2219 \pm 0.0091$ | $0.0902 \pm 0.0054$ |
| 16 | $0.5021 \pm 0.0123$ | $0.2816 \pm 0.0418$ | $0.2170 \pm 0.0075$ | $0.0878 \pm 0.0058$ |

Table 7: MS Marco Validation Accuracy (Mean ± Standard Deviation) for different sequence lengths $T$ and number of heads.

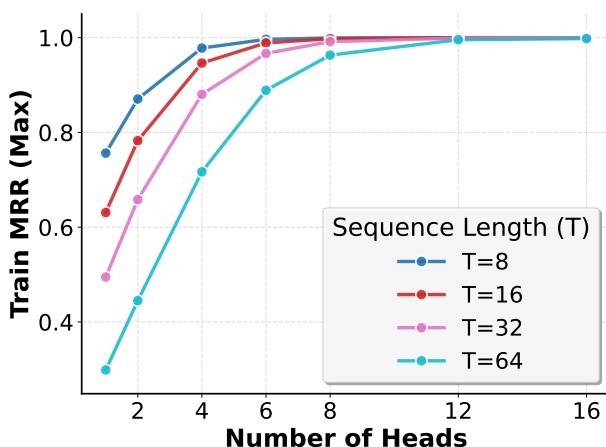

Figure 5: Plot of training mrr for MS MARCO dataset.

effort to learn how to extract useful information. The random placement make sure the padded outside aera cannot be simply ignored by position encodings.

Each image is divided into non-overlapping patches of size $8 \times 8$ pixels, resulting in a sequence of patches for each image. For each image size, the sequence length $T$ is defined as the total number of patches plus one additional class token, with $T = \{17, 37, 65, 145, 257\}$. We adopt the standard CIFAR-10 data splits, which include $50,000$ training images and $10,000$ test images across $10$ classes.

### B.3.2 MODEL TRAINING SETUP

We evaluate a Vision Transformer (ViT) with four layers and a per-head embedding dimension of 16, while varying the number of attention heads in different configurations. Each image patch is embedded through a linear projection, and positional embeddings are added along with a learnable class token. No global convolutional embedding is used.

Input processing follows the standard ViT procedure, including patch embedding of size $8 \times 8$, positional encoding, and aggregation of the class token for final classification. The model is trained using the AdamW optimizer with cosine annealing learning rate scheduling. Standard architectural techniques, such as layer normalization, residual connections, and dropout, are employed for regularization.

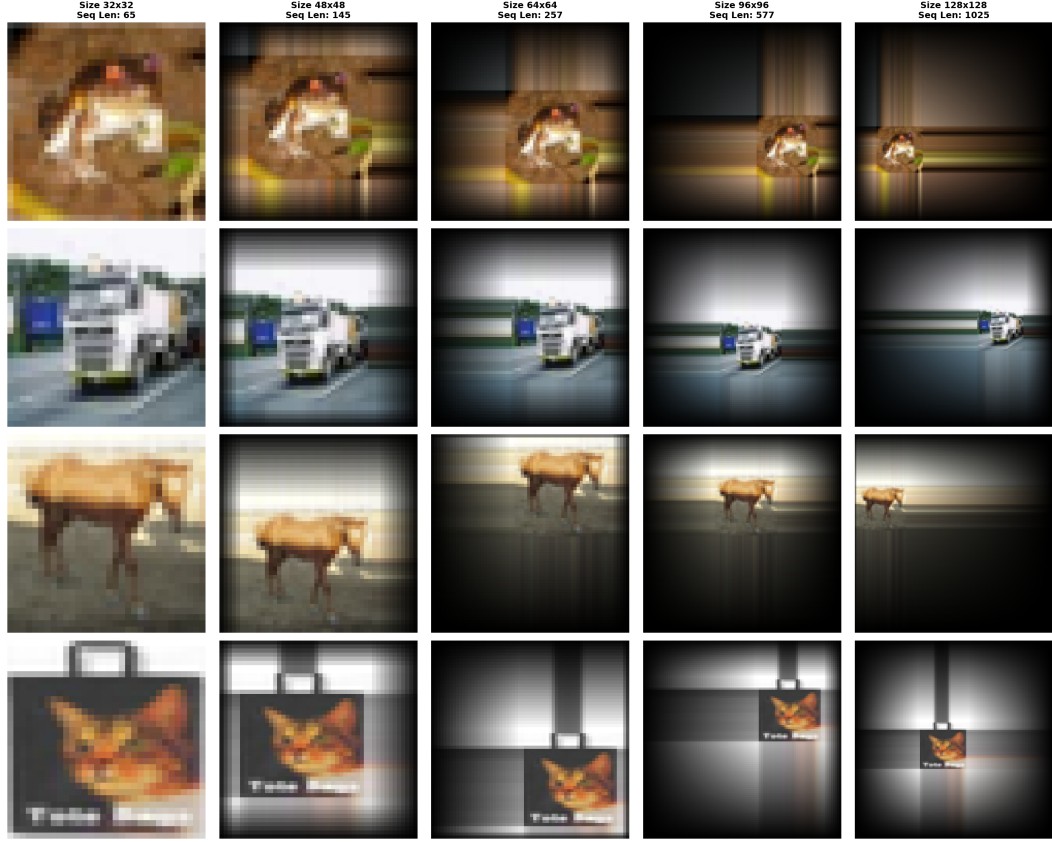

Figure 6: Examples of the padded images from the dataset.

| Heads | Seq=65 | Seq=145 | Seq=257 | Seq=577 | Seq=1025 |
|---|---|---|---|---|---|
| 1 | $4.78 \times 10^1 \pm 4.50 \times 10^{-1}$ | $4.37 \times 10^1 \pm 4.70 \times 10^{-1}$ | $4.20 \times 10^1 \pm 5.00 \times 10^{-1}$ | $4.08 \times 10^1 \pm 6.50 \times 10^{-1}$ | $4.00 \times 10^1 \pm 7.20 \times 10^{-1}$ |
| 2 | $5.97 \times 10^1 \pm 4.50 \times 10^{-1}$ | $5.52 \times 10^1 \pm 3.80 \times 10^{-1}$ | $5.34 \times 10^1 \pm 4.40 \times 10^{-1}$ | $5.15 \times 10^1 \pm 4.70 \times 10^{-1}$ | $5.08 \times 10^1 \pm 7.40 \times 10^{-1}$ |
| 4 | $7.55 \times 10^1 \pm 2.10 \times 10^{-1}$ | $7.03 \times 10^1 \pm 3.20 \times 10^{-1}$ | $6.85 \times 10^1 \pm 7.70 \times 10^{-1}$ | $6.62 \times 10^1 \pm 6.00 \times 10^{-1}$ | $6.58 \times 10^1 \pm 7.20 \times 10^{-1}$ |
| 8 | $9.51 \times 10^1 \pm 1.50 \times 10^{-1}$ | $9.26 \times 10^1 \pm 4.70 \times 10^{-1}$ | $9.14 \times 10^1 \pm 6.20 \times 10^{-1}$ | $9.07 \times 10^1 \pm 1.02 \times 10^{0}$ | $9.02 \times 10^1 \pm 1.00 \times 10^{0}$ |
| 10 | $9.81 \times 10^1 \pm 5.00 \times 10^{-2}$ | $9.73 \times 10^1 \pm 2.40 \times 10^{-1}$ | $9.67 \times 10^1 \pm 4.60 \times 10^{-1}$ | $9.65 \times 10^1 \pm 3.20 \times 10^{-1}$ | $9.67 \times 10^1 \pm 7.30 \times 10^{-1}$ |
| 11 | $9.88 \times 10^1 \pm 1.20 \times 10^{-1}$ | $9.83 \times 10^1 \pm 1.20 \times 10^{-1}$ | $9.81 \times 10^1 \pm 2.40 \times 10^{-1}$ | $9.77 \times 10^1 \pm 2.10 \times 10^{-1}$ | $9.79 \times 10^1 \pm 2.20 \times 10^{-1}$ |
| 12 | $9.92 \times 10^1 \pm 3.00 \times 10^{-2}$ | $9.89 \times 10^1 \pm 1.60 \times 10^{-1}$ | $9.86 \times 10^1 \pm 2.40 \times 10^{-1}$ | $9.86 \times 10^1 \pm 1.90 \times 10^{-1}$ | $9.86 \times 10^1 \pm 2.90 \times 10^{-1}$ |
| 13 | $9.94 \times 10^1 \pm 6.00 \times 10^{-2}$ | $9.93 \times 10^1 \pm 6.00 \times 10^{-2}$ | $9.91 \times 10^1 \pm 1.10 \times 10^{-1}$ | $9.90 \times 10^1 \pm 2.00 \times 10^{-1}$ | $9.91 \times 10^1 \pm 2.50 \times 10^{-1}$ |
| 14 | $9.96 \times 10^1 \pm 3.00 \times 10^{-2}$ | $9.94 \times 10^1 \pm 9.00 \times 10^{-2}$ | $9.93 \times 10^1 \pm 1.00 \times 10^{-1}$ | $9.93 \times 10^1 \pm 1.70 \times 10^{-1}$ | $9.93 \times 10^1 \pm 2.30 \times 10^{-1}$ |
| 16 | $9.97 \times 10^1 \pm 3.00 \times 10^{-2}$ | $9.96 \times 10^1 \pm 2.00 \times 10^{-2}$ | $9.95 \times 10^1 \pm 7.00 \times 10^{-2}$ | $9.96 \times 10^1 \pm 1.40 \times 10^{-1}$ | $9.96 \times 10^1 \pm 1.70 \times 10^{-1}$ |
| 20 | $9.98 \times 10^1 \pm 1.00 \times 10^{-2}$ | $9.97 \times 10^1 \pm 5.00 \times 10^{-2}$ | $9.97 \times 10^1 \pm 8.00 \times 10^{-2}$ | $9.98 \times 10^1 \pm 6.00 \times 10^{-2}$ | $9.99 \times 10^1 \pm 7.00 \times 10^{-2}$ |
| 24 | $9.99 \times 10^1 \pm 2.00 \times 10^{-2}$ | $9.98 \times 10^1 \pm 2.00 \times 10^{-2}$ | $9.98 \times 10^1 \pm 4.00 \times 10^{-2}$ | $9.99 \times 10^1 \pm 4.00 \times 10^{-2}$ | $9.99 \times 10^1 \pm 5.00 \times 10^{-2}$ |

Table 8: Error bar for Image task. Accuracy (Mean ± Standard Deviation) for different sequence lengths and number of heads.

## C  LARGE LANGUAGE MODEL USAGE

Large language models were used only for linguistic refinement (e.g., polishing sentences and checking grammar). The core ideas, theoretical results, experimental design, and analyses presented in this paper were entirely developed by the authors without assistance from large language models.

| Heads | T=65 | T=145 | T=257 | T=577 | T=1025 |
|---|---|---|---|---|---|
| 1 | $50.50 \pm 0.44$ | $45.85 \pm 0.58$ | $43.53 \pm 0.58$ | $41.43 \pm 0.97$ | $39.95 \pm 0.75$ |
| 2 | $60.01 \pm 0.57$ | $55.01 \pm 0.25$ | $53.12 \pm 0.69$ | $49.95 \pm 0.80$ | $48.53 \pm 0.83$ |
| 4 | $67.98 \pm 0.43$ | $63.49 \pm 0.70$ | $61.42 \pm 0.61$ | $57.19 \pm 0.54$ | $55.31 \pm 0.85$ |
| 8 | $69.65 \pm 0.55$ | $66.06 \pm 0.53$ | $62.43 \pm 0.95$ | $57.58 \pm 0.59$ | $56.18 \pm 1.14$ |
| 10 | $69.70 \pm 0.24$ | $65.64 \pm 0.37$ | $62.45 \pm 0.49$ | $57.21 \pm 1.03$ | $54.44 \pm 1.49$ |
| 11 | $69.84 \pm 0.36$ | $65.26 \pm 0.43$ | $61.97 \pm 0.29$ | $56.95 \pm 0.63$ | $53.77 \pm 2.91$ |
| 12 | $69.66 \pm 0.39$ | $65.79 \pm 0.56$ | $62.63 \pm 0.36$ | $56.57 \pm 0.66$ | $53.17 \pm 1.06$ |
| 13 | $69.72 \pm 0.18$ | $65.30 \pm 0.59$ | $61.66 \pm 0.58$ | $54.81 \pm 2.23$ | $52.90 \pm 1.45$ |
| 14 | $69.49 \pm 0.48$ | $65.25 \pm 0.48$ | $61.32 \pm 0.95$ | $54.01 \pm 2.13$ | $50.42 \pm 2.48$ |
| 16 | $69.69 \pm 0.33$ | $64.24 \pm 0.33$ | $59.29 \pm 1.12$ | $49.68 \pm 1.39$ | $48.51 \pm 2.59$ |
| 20 | $67.99 \pm 0.35$ | $61.49 \pm 1.07$ | $55.25 \pm 2.12$ | $48.65 \pm 0.85$ | $46.89 \pm 1.07$ |
| 24 | $65.12 \pm 0.68$ | $56.80 \pm 1.09$ | $52.14 \pm 0.56$ | $48.74 \pm 0.60$ | $48.39 \pm 0.76$ |

Table 9: Error bar for Image task. Validation Accuracy (Mean ± Standard Deviation) for different sequence lengths and number of heads.

Table 10: Hyperparameter settings of popular transformer models. Only d (embedding dimension), L (number of layers), and H (number of attention heads) are shown for brevity.

| H | Model | d | L | Year |
|---|---|---|---|---|
| 8 | Attention is all you need | 512 | 6 | 2017 |
| 8 | Gemma 2B | 2,048 | 18 | 2024 |
| 12 | GPT | 768 | 12 | 2018 |
| 16 | BERT-Large | 1,024 | 24 | 2019 |
| 16 | ViT-Huge | 1,280 | 32 | 2021 |
| 16 | Gemma 7B | 3,072 | 28 | 2024 |
| 28 | Turing-NLG | 4,256 | 78 | 2020 |
| 32 | LLaMA-7B | 4,096 | 32 | 2023 |
| 32 | Baichuan 2-7B | 4,096 | 32 | 2023 |
| 32 | Mistral 7B | 4,096 | 32 | 2023 |
| 32 | Yi-6B | 4,096 | 32 | 2023 |
| 32 | LLaMA 3-8B | 4,096 | 32 | 2024 |
| 32 | Mixtral 8x7B | 4,096 | 32 | 2024 |
| 40 | LLaMA-13B | 5,120 | 40 | 2023 |
| 40 | Baichuan 2-13B | 5,120 | 40 | 2023 |
| 56 | Yi-34B | 7,168 | 60 | 2023 |
| 64 | LLaMA-65B | 8,192 | 80 | 2023 |
| 64 | Llama-2-70B | 8,192 | 80 | 2023 |
| 64 | LLaMA 3-70B | 8,192 | 80 | 2024 |
| 96 | GPT-3 | 12,288 | 96 | 2020 |
| 96 | Jurassic-1 | 13,824 | 76 | 2021 |
| 128 | MT-NLG | 20,480 | 105 | 2021 |
| 128 | LaMDA | 8,192 | 64 | 2022 |
| 128 | LLaMA 3.1-405B | 16,384 | 126 | 2024 |
| 128 | DeepSeek-V2 | 5,120 | 60 | 2024 |

