# OpenReview forum: "The Effect of Attention Head Count on Transformer Approximation"
_ICLR.cc/2026/Conference — ICLR 2026 Poster_

### Official Review · Reviewer_vXWe · 2025-10-25

**Soundness:** 4
**Presentation:** 3
**Contribution:** 2
**Rating:** 6
**Confidence:** 4

**Summary:**

The paper shows that target functions that can be concisely expressed in a given form can be efficiently approximated with transformers with a sufficient amount of heads, and inefficiently so when the number of heads is lower than a given threshold.

**Strengths:**

1. The paper is generally well-written and explained. Intuition is provided after most mathematical definitions and results which makes it easy to follow the reasoning and the motivation.
2. The introduction of the class of generalized D-retrieval tasks and proving that they are dense in the space of continuous functions is quite neat. It does remind me of the Kolmogorov-Arnold representation theorem, although there are a few details that might be of concern (see Weaknesses).
3. The experiments demonstrate the predictions from theory indeed do appear in practice.

**Weaknesses:**

1. Theorem 1 is purely a density result. It says nothing as to how the intrinsic dimensionality $D$ and the smoothness of $f$ depend on the target function $F$.
2. This is a problem because the authors then go on to study how $D$ affects the required number of heads for a transformer to approximate $F$ well. However, if the link between the $F$ and $D$ is not established, it is difficult to be convinced whether the conclusions of the paper actually say anything of practical value. For instance, there could be trivial constructions that require a very large $D$ and $f$ with very high Lipschitz constant. Then all that the inner functions $f_i$ would do is to embed the sequence into a $D$-long vector and then let the MLP do the heavy lifting. But then that MLP might have to be extremely large. Overall, this interplay does not seem to be taken good care of.
3. The experiments on real datasets do not study the interplay between the number of layers and the number of heads. How were the depths used (two and four layers) chosen? Additional experiments with varying depth possibly depth vs number of heads plot could be nice to have.
4. On line 131 you say that “we omit layer normalization, noting that its removal does not harm the approximation capacity of the model”. However, layer norm (depending on where it is applied) might limit the ability of softmax to be “picky” and to approximate hardmax. Similarly, on line 151 you say that $\beta$ can be chosen “arbitrarily large in order to make the softmax attention mechanism approximate a hardmax.” However, for both numerical stability and learnability, we do not want arbitrarily large activations in models. I understand that these are necessary assumptions for the theory to work in the limit but I think you should be more upfront with that they are significant and an important departure from how models work in practice.

**Questions:**

Minor comments:

1. On line 182 you use the term “intrinsic dimension” for the first time before it is defined, it feels a bit handwavy at this point. Perhaps you can explain what it means.
2. There seems to be a mix in the notation in Figure 1 and the experiments section: both $h$ and $H$ seem to refer to the number of heads in the model.

---

> ### Author Response · Authors · 2025-11-20
> **Reply to Weakness (Part 1)**
>
> **W1: Theorem 1 is purely a density result. It says nothing as to how the intrinsic dimensionality $D$ and the smoothness of $f$ depend on the target function $F$.**
>
> Answer: (W1 and W2 are answered together).
>
> **W2: This is a problem because the authors then go on to study how $D$ affects the required number of heads for a transformer to approximate $F$ well. However, if the link between the $F$ and $D$ is not established, it is difficult to be convinced whether the conclusions of the paper actually say anything of practical value. For instance, there could be trivial constructions that require a very large $D$ and $f$ with very high Lipschitz constant. Then all that the inner functions $f_i$ would do is to embed the sequence into a $D$-long vector and let the MLP do the heavy lifting. But then that MLP might have to be extremely large. Overall, this interplay does not seem to be taken good care of.**
>
>
> Answer: We appreciate the reviewer’s comments. Theorem 1 is only used to ensure that the target class defined by $F_0$ and $\{f_i\}_{i=1}^D$ is expressive enough. While achieving density may require large $D$, these are rather the extreme cases that is not the primary focus of the paper.
>
> The target $H$ is determined by $f_i$, $S_i$ and $F_0$ together, so the smoothness of $f_i$ does not depend on $F_0$. In Theorem 2, we study targets where $f_i$ and $F_0$ are arbitrarily smooth and do not depend on $T$, under the setting $d, D, n \ll T \ll 1/\epsilon$. The practical value here is that this target class provides explicit examples that cannot be well approximated by transformers with too few heads ($h<D$), but can be efficiently approximated when $h \ge D$.
>
> In particular, even for the simple and smooth case
>
> $f_i(x) = \|x - x^{(i)}\|^2,
> \qquad
> F_0(z_1,\dots,z_D)=\sum_{i=1}^D z_i$,
>
> a transformer with $D$ heads can approximate well (Theorem 2.1), while a transformer with $D-1$ heads provably suffers from the bottleneck (Theorem 2.2). This shows that the lower bound does not rely on highly non-smooth $f_i$ and $F_0$ as in the Kolmogorov–Arnold superposition theorem, where the inner functions may be only Hölder continuous. Instead, the separation we prove comes purely from the architectural limitation in the number of heads.
>
> Uniqueness: The representation $(f_i, S_i, F_0)$ of a given target $H$ may not be unique (for instance, one can replace $f_i$ by $2f_i$ and compensate in $F_0$ without changing $H$). However, under our regime $D \ll T$ and within the class of decompositions satisfying our assumptions, we view $D$ as an intrinsic quantity: if two such representations $(f_i, S_i, F_0)$, $i=1,\dots,D_1$, and $(\tilde f_i, \tilde S_i, \tilde F_0)$, $i=1,\dots,D_2$, define the same target $H$ for input lengths $T \gg D_1, D_2$, then their intrinsic dimensions must coincide ($D_1 = D_2$). In other words, while the specific choice of $f_i, S_i$ and $F_0$ is not unique, the corresponding $D$ is. This can be formalized as a corollary of theorem 2, and we will sketch the argument in the revised version.

---

> ### Author Response · Authors · 2025-11-20
> **Reply to Weakness (Part 2)**
>
> **W3: The experiments on real datasets do not study the interplay between the number of layers and the number of heads. How were the depths used (two and four layers) chosen? Additional experiments with varying depth possibly depth vs number of heads plot could be nice to have.**
>
> Answer: We thank the reviewer for bringing up this important aspect. There is no special reason to choose 2 or 4 layers in the experiments, just to examine if our theory can work at least qualitatively on multilayer transformers. Extending the theory to multi-layer architectures is indeed a natural next step. While we leave a rigorous analysis for future work, our preliminary results indicate that the head bottleneck persists: insufficient total head count cannot be fully compensated by depth alone.
>
> In particular, our observations suggest that a necessary condition for efficient approximation is approximately
>
> $l \cdot h \ge D$,
>
> meaning that the intrinsic dimension must be adequately represented across layers to avoid information loss.
>
> At the same time, we conjecture that depth becomes increasingly beneficial when the target involves more entangled or hierarchical interactions among coordinates—structures not present in our current synthetic setup. In such cases, multiple layers may help resolve dependencies that a single layer cannot disentangle, while the head count still governs the capacity to represent independent retrieval directions.
>
> To empirically support the above intuition, we include an additional result on the same $D=4$ synthetic task, using a 2-layer Transformer (no positional encoding, no layer normalization). The performance still improves when $h$ increases, consistent with $lh \ge D$ acting as a threshold:
>
> **Two-layer transformer on the synthetic task ($D = 4$, NoPE, NoLN)**
>
> | **Number of Heads $h$** | **$T=8$** | **$T=16$** | **$T=32$** | **$T=64$** | **$T=128$** |
> |------------------------:|:---------:|:----------:|:----------:|:----------:|:-----------:|
> | 1 | 2.12e-04 | 1.85e-04 | 2.22e-04 | 3.13e-04 | 4.28e-04 |
> | 2 | 7.22e-06 | 2.69e-06 | 3.50e-06 | 3.07e-06 | 3.83e-06 |
> | 3 | 8.16e-06 | 1.83e-06 | 1.73e-06 | 3.86e-06 | 3.50e-06 |
> | 4 | 3.68e-06 | 1.87e-06 | 2.60e-06 | 4.32e-06 | 5.98e-06 |
> | 5 | 6.15e-06 | 3.02e-06 | 3.31e-06 | 3.78e-06 | 5.34e-06 |
>
> We will incorporate this discussion in the revised manuscript to make the role of depth clearer.
>
>
> **W4: On line 131 you say that “we omit layer normalization, noting that its removal does not harm the approximation capacity of the model”. However, layer norm (depending on where it is applied) might limit the ability of softmax to be “picky” and to approximate hardmax. Similarly, on line 151 you say that $\beta$ can be chosen “arbitrarily large in order to make the softmax attention mechanism approximate a hardmax.” However, for both numerical stability and learnability, we do not want arbitrarily large activations in models. I understand that these are necessary assumptions for the theory to work in the limit but I think you should be more upfront with that they are significant and an important departure from how models work in practice.**
>
> Answer: Thank you for pointing this out. We agree that omitting layer normalization and allowing large attention logits are idealized assumptions that simplify the analysis but differ from practical training constraints. These assumptions are used only to isolate approximation capacity in the limit.
>
> That said, we conjecture that the key lower-bound result (Theorem 2.2) continues to hold even when layer normalization is present or when the scaling of weight matrices is constrained, since the fundamental bottleneck arises from insufficient head specialization rather than from the particular normalization strategy. We will clarify these assumptions and this conjecture explicitly in the revised manuscript.

---

> ### Author Response · Authors · 2025-11-20
> **Reply to Questions**
>
> **Q1: On line 182 you use the term “intrinsic dimension” for the first time before it is defined, it feels a bit handwavy at this point. Perhaps you can explain what it means.**
>
> Answer: Thank you for pointing this out. We will clarify that the “intrinsic dimension” refers to the smallest $D$ (Or as shown in the answer W2, the unique $D \ll T$) such that the target can be written in the $D$-generalized retrieval task form (i.e., depending effectively on only $D$ latent values extracted from the sequence) and satisfying all assumptions. We will add this explanation at its first use.
>
>
> **Q2: There seems to be a mix in the notation in Figure 1 and the experiments section: both $h$ and $H$ seem to refer to the number of heads in the model.**
>
> Answer: We thank the reviewer for catching this inconsistency. In the revision, we will uniformly use the lowercase $h$ to denote the number of heads throughout the paper, including Figure 1 and the experimental section.

---

> > ### Comment · Reviewer_vXWe · 2025-11-22
> >
> > Thank you for your response.
> >
> > I do agree that Theorem 1 being purely a density result, is not really a problem for the claims you are making in the paper. I am nevertheless still concerned about potentially offloading the heavy lifting to the MLP.
> >
> > Thank you for the clarification on the other questions, I am happy with all that. With that in mind, I will maintain my score.

---

### Official Review · Reviewer_YCdB · 2025-11-01

**Soundness:** 3
**Presentation:** 3
**Contribution:** 3
**Rating:** 8
**Confidence:** 3

**Summary:**

This paper studies how the approximation properties of single-layer transformers on sequence-to-vector tasks vary with the number of attention heads. To investigate this question, they first introduce a new 'generalized D-retrieval' task, which they prove to be dense in the space of continuous sequence-to-vector functions. They then theoretically find that the number of parameters necessary to approximate their D-retrieval task is small when the number of attention heads $\geq D$ and scales at least with $O\left(1 / \epsilon^{c T}\right)$ where $T$ is the sequence length if the number of attention heads $< D$.
They validate their theoretical findings empirically on toy as well as more realistic datasets.

**Strengths:**

The story of the paper was clear and the paper was well-organized. The introduction  of the generalized D-retrieval task and the connection to the necessary number of heads of a single-layer transformer to efficiently approximate it is original. In particular, the lower bound on the number of parameters necessary to achieve a certain accuracy is interesting and useful. The experiments were well designed and presented.

**Weaknesses:**

The presentation of the experimental results could be improved. In particular, it is not clear to me why they used the minimal validation NMSE achieved across multiple random seeds on the synthetic tasks, while using the training accuracy on the real datasets and the analysis of the results on the real datasets was kept very short, making it hard to evaluate the applicability of these results to real-world problems.

**Questions:**

1. In line 190, why do you make the assumption $\left|S_i\right| \geq \frac{1}{4} T$? Where is this assumption needed?
2. Should the input space of $F$ in line 195 depend on $T$? If yes, could you quickly explain this.
3. Could you expand a bit on Assumption 2? Why or why not do you think they are reasonable assumptions?
4. Could you explain the following sentence in slightly more detail:
> As T increases, the number of relevant elements to distinguish grows linearly with T , yet they are compressed into an n-dimensional vector whose size does not scale with T.
5. Could you explain the following sentence in more detail:
> NMSE, equivalent to $1−R^2$, corrects for the variance shrinkage of maxima as T grows, thus enabling fair comparison across lengths.
6. Could you expand on why you used the minimal validation NMSE achieved across multiple random seeds on the synthetic tasks, while using the training accuracy on the real datasets?
7. Do you have further thoughts on how one might be able to verify that the experiments on real datasets indeed identify the 'correct' intrinsic dimension of the task? Might it be useful to train multiple slightly different transformers (e.g., different numbers of layers) on the same task to check whether the discovered intrinsic dimension is constant?

---

> ### Author Response · Authors · 2025-11-20
> **Reply to Weakness**
>
> **W1: The presentation of the experimental results could be improved. In particular, it is not clear to me why they used the minimal validation NMSE achieved across multiple random seeds on the synthetic tasks, while using the training accuracy on the real datasets and the analysis of the results on the real datasets was kept very short, making it hard to evaluate the applicability of these results to real-world problems.**
>
> Answer: We thank the reviewer for this observation. We provide a detailed justification of our evaluation protocol and its relevance to real-world settings in our response to Question 6 below. To avoid redundancy, we will incorporate that explanation directly into the revised manuscript and ensure that the presentation of both synthetic and real-data results is made more consistent and easier to interpret.

---

> ### Author Response · Authors · 2025-11-20
> **Reply to Questions (Part 1)**
>
> **Q1: In line 190, why do you make the assumption $|S_i| \ge \frac{1}{4}T$? Where is this assumption needed?**
>
> Answer: The assumption $|S_i| \ge \frac{1}{4}T$ is used in the proof of Theorem 2.2 to ensure that each ``max-selection’’ over the set $S_i$ still involves a number of elements proportional to $T$. This proportionality allows us to construct sufficiently many candidate subsequences to carry out the pigeonhole argument and derive the lower bound.
>
> In fact, the constant $\frac{1}{4}$ is not essential: any fixed $\alpha \in (0, \frac{1}{2})$ would suffice, changing the expression of $k$ in Theorem 2.2 from
> $k = \frac{\left(\frac{1}{4}T - s - D + 1\right)}{(n+1)s+1} - 1$ to $k =
> \frac{\left(\alpha T - s - D + 1\right)}{(n+1)s+1} - 1$.
> We chose $\frac{1}{4}$ simply to keep the constants readable and avoid introducing additional notations that may distract from the main theoretical message.
>
>
> **Q2: Should the input space of $F$ in line 195 depend on $T$? If yes, could you quickly explain this.**
>
> Answer: We thank the reviewer for catching this notational ambiguity. The expression in line 195 should indeed be revised. A clearer and correct statement is:
>
> > For vector-valued targets $H:[0,1]^{d\times T} \to \mathbb{R}^\ell$ defined through the same functions $f_i$, subsets $S_i$, and an outer map $F_0 : [0,1]^D \to \mathbb{R}^\ell$, the extension is applied coordinate-wise, since each coordinate function of $F_0$ can be approximated independently.
>
>
> Under this corrected formulation, $F_0$ itself does not depend on $T$; only the domain of $H$ scales with the input length. We will update the manuscript accordingly to avoid confusion.
>
>
> **Q3: Could you expand a bit on Assumption 2? Why or why not do you think they are reasonable assumptions?**
>
> Answer: Thank you for raising this question. Assumption 2 is only required for establishing the lower bound in Theorem 2.2, and we believe these conditions are reasonable because they exclude only degenerate cases while preserving broad generality for both the functions $f_i$ and the outer map $F_0$. More specifically:
>
> - Assumptions (2.1) and (2.4) ensure that each $f_i$ behaves regularly around its minimizer, enabling a clean geometric separation of the $D$ basins into which inputs may fall. A representative example is $f_i(x)=\|x-x^{(i)}\|^2$. A degenerate example excluded by these assumptions is $f_i(x) \equiv c_0$ for constant $c_0$, which is totally independent of the input.
> - Assumption (2.2) requires distinct minimizers, which allows partitioning the space into $D$ disjoint minima basin around each minimizer $x^{(i)}$. Since an attention head can effectively specialize to at most one such region, the condition $h < D$ then guarantees an unattended region—crucial for the lower-bound construction. A degenerate example excluded by this assumption is $f_1 = f_2 = \dots = f_D$.
> - Assumption (2.3) enforces sensitivity of the target to small perturbations near the minimizers, ruling out trivial flat cases (such as $F_0 \equiv c_0$ for constant $c_0$) where no meaningful separation can be made. A simple example satisfying this is $F_0(z_1,\dots,z_D) = \sum_{i=1}^D z_i$.
>
> We will incorporate this intuition into the revised manuscript to clarify that Assumption 2 serves only to rule out pathological scenarios while maintaining a wide applicability of the result.
>
> **Q4: Could you explain the following sentence in slightly more detail:**
>
> > As T increases, the number of relevant elements to distinguish grows linearly with T, yet they are compressed into an n-dimensional vector whose size does not scale with T.
>
>
> Answer: We appreciate the reviewer’s request for clarification. The quoted sentence refers to the intuition behind the synthetic example, where the model must ultimately compute a quantity of the form $\max_{1\le t\le T} x_t$. With inadequate head count, the information about all $T$ input elements must be preserved through the attention layer. However, as each head has only $n$ embedding dimensions, this entire set of values is compressed into an $n$-dimensional vector whose size does not scale with $T$.
>
> A simple way to visualize this compression is to imagine storing multiple input values in a single coordinate of the embedding vector, for example by packing several $x_t$'s into the decimal expansion of one number. If we pack $T/n$ values into one coordinate, then accurately recovering them later (to accuracy $\epsilon$) would require precision on the order of $\epsilon^{T/n}$. Thus, as $T$ increases while $n$ remains fixed, the representation must become exponentially more precise in order to retain the necessary information, revealing the core bottleneck that drives the lower bound.

---

> ### Author Response · Authors · 2025-11-20
> **Reply to Questions (Part 2)**
>
> **Q5: Could you explain the following sentence in more detail:**
>
> >NMSE, equivalent to $1 - R^2$, corrects for the variance shrinkage of maxima as T grows, thus enabling fair comparison across lengths.
>
> Answer: Thank you for the question. The intuition comes from the behavior of maxima of i.i.d. random variables. Let the target output be $Y_T = \max_{1 \le t \le T} x_t$ with input tokens $x_t \sim \mathcal{N}(0,1)$ independently, as similar to synthetic experiment setting. (which means input sequence $x_{[T]} \sim \mathcal{N}(0, I_T)$). It is known that $\operatorname{Var}(Y_T)$ decreases as $T$ increases, because $Y_T$ concentrates more tightly around its growing mean. Therefore, using the standard MSE would artificially suggest better performance at larger $T$ simply due to reduced variance of the target.
>
> Normalized MSE (NMSE), defined as $1 - R^2$, compensates for this effect by normalizing with the variance of the target. This ensures that a trivial predictor (e.g. always outputting $\mathbb{E}[Y_T]$) yields the same loss value (namely, 1) for all $T$. Thus, NMSE enables fair comparison of approximation quality across different sequence lengths.
>
>
> **Q6: Could you expand on why you used the minimal validation NMSE achieved across multiple random seeds on the synthetic tasks, while using the training accuracy on the real datasets?**
>
> Answer: Thank you for the opportunity to clarify. Our choice of evaluation metrics reflects the differing goals and target structures of the synthetic and real datasets.
>
> On the synthetic tasks, the output range is $\mathbb{R}$, and the focus is strictly on approximation ability. As discussed above (see response to Q5), NMSE is used to allow fair comparison across varying sequence lengths $T$, and the model exhibits stable generalization on this controlled setting. Therefore, the validation NMSE is the appropriate metric.
>
> In contrast, for the real datasets, the output range is a discrete set (e.g., identifying the most relevant input token), and there is no natural NMSE-like normalization that reflects task difficulty across different instances. Accuracy is the standard metric in such retrieval settings. Since the primary objective of these experiments is to demonstrate that the theoretical head-count effect also appears at the level of approximation on real data, we report the training accuracy to isolate architectural expressivity from issues related to optimization or data scarcity. For completeness, we will additionally report test accuracy in the appendix of the revised manuscript; in our current setup, generalization is noticeably weaker and the trends are less stable, so we avoid drawing strong conclusions about head-count effects from the test metrics.
>
>
> In the revision, we will make these distinctions explicit to improve clarity and consistency in the experimental presentation.
>
>
> **Q7: Do you have further thoughts on how one might be able to verify that the experiments on real datasets indeed identify the ‘correct’ intrinsic dimension of the task? Might it be useful to train multiple slightly different transformers (e.g., different numbers of layers) on the same task to check whether the discovered intrinsic dimension is constant?**
>
> Answer: We appreciate this thoughtful suggestion. Beyond fully training models to detect the transition, one potential approach would be to analyze head contributions early in training to estimate how many heads meaningfully affect the output. While this idea is promising, we do not yet have sufficient empirical evidence to draw firm conclusions.
>
> We also agree that your  proposed strategy of training multiple transformers with varying depths and head counts on the same task, and then comparing how the error scales with $T$, is a valid way to probe whether the inferred intrinsic dimension is stable across architectures. We will mention both of these ideas as possible directions for future investigation in the revised version.

---

> > ### Comment · Reviewer_YCdB · 2025-11-28
> >
> > I thank the authors for answering my questions, addressing all of my concerns, and making updates to the paper as necessary.
> >
> > I will therefore maintain my score and continue to recommend acceptance.

---

### Official Review · Reviewer_SJw1 · 2025-11-03

**Soundness:** 3
**Presentation:** 3
**Contribution:** 3
**Rating:** 8
**Confidence:** 4

**Summary:**

This paper studies the task of approximating sequence-to-vector mappings with a particular structure, where the output is a function of $D$ summary statistics of the input sequence with length $T$. The authors prove that with a number of heads $h \geq D$, a Transformer with a single attention layer can efficiently approximate this class of mappings. On the other hand, with $h < D$, and an embedding dimension $n \ll T$, approximating up to $\epsilon$ error requires $(1/\epsilon)^{\Omega(T)}$ parameters. When the embedding dimension is $n \geq Td$ where $d$ is the input token dimension, the Transformer can efficiently approximate these mappings by memorizing the input sequence.

**Strengths:**

The paper presents an interesting setting where one can clearly prove the benefits of multiple heads for certain long-context tasks. It shows a nice tradeoff where one either needs to use as many heads as the intrinsic dimension of the mapping, or needs to use an embedding dimension that grows with the sequence length; in practical settings the former is always preferred for long-context problems. The authors also demonstrate the relevance of their introduced task by showing that it is a dense subset of the class of continuous sequence-to-vector mappings.

**Weaknesses:**

* A key weakness of the results is that they only apply to a single-layer Transformer, i.e. a Transformer using only one layer of attention. It is unclear how the lower bounds would change if one were to use multiple layers with fewer heads.

* It would be nice if the authors could provide some intuition on the necessity of Assumption 2. Some of the items in this assumption seem a bit specific, e.g. requiring non-zero coordinates in the gradient of $F\_0$.

* It would also be nice if the authors could include a brief intuition of the proof strategy, especially the strategy for proving the lower bound, in the main text.

**Questions:**

* There seems to be a sudden transition in between cases 1 and 2 of Theorem 2, i.e. even with $h = D - 1$ one has a lower bound $M \geq 1/\epsilon^{\Omega(T/(nD))}$, while with $h = D$ we suddenly get $M \leq 1/\epsilon^{\gamma}$. Is there an intuition on this sudden change?

* Similarly, what is the reason for the difference between $M \leq 1/\epsilon^{\gamma}$ for Case 1 and $M \leq 1/\epsilon^{1 + \gamma}$ for Case 3 in Theorem 2?

* A relevant work that could be discussed is [1]. The authors there consider a sparse sequence-to-sequence task, and in one of their results they separate the approximation power of a single-head and multi-head attention in Transformers. The task studied in [1] could also be of interest here, as it allows to move from sequence-to-vector mappings to sequence-to-sequence ones where the length of the output sequence can grow with the input.

* It would be nice if $C_{d,D,T}$ could be made explicit, at least in terms of $T$, since long inputs are of particular interest here.


[1] A. Mousavi-Hosseini et al. "When Do Transformers Outperform Feedforward and Recurrent Networks? A Statistical Perspective". NeurIPS 2025.

---

> ### Author Response · Authors · 2025-11-20
> **Reply to Weakness (Part 1)**
>
> **W1: A key weakness of the results is that they only apply to a single-layer Transformer; i.e., a Transformer using only one layer of attention. It is unclear how the lower bounds would change if one were to use multiple layers with fewer heads.**
>
> Answer: We thank the reviewer for pointing this out. Extending the theory to multi-layer architectures is indeed a natural next step. While a full analysis is left for future work, our current intuition and experiments suggest that the head bottleneck persists: when the total number of heads across layers is insufficient to cover the intrinsic dimension $D$, approximation remains challenging.
>
> In particular, based on our observations, a necessary condition for efficient approximation in an $l$-layer Transformer is approximately
>
> $l \cdot h \ge D$,
>
> indicating that the task's intrinsic dimension must be sufficiently represented across layers.
>
> To empirically support this intuition, we include below an additional 2-layer experiment on the same synthetic task ($D=4$), without positional encoding and layer normalization (both of which introduce inductive biases irrelevant to the theoretical focus). The accuracy still improves significantly as $h$ increases, consistent with $lh \ge D$ acting as a performance threshold.
>
> **Two-layer transformer on the synthetic task ($D = 4$, NoPE, NoLN)**
>
> | **Number of Heads $h$** | **$T=8$** | **$T=16$** | **$T=32$** | **$T=64$** | **$T=128$** |
> |------------------------:|:---------:|:----------:|:----------:|:----------:|:-----------:|
> | 1 | 2.12e-04 | 1.85e-04 | 2.22e-04 | 3.13e-04 | 4.28e-04 |
> | 2 | 7.22e-06 | 2.69e-06 | 3.50e-06 | 3.07e-06 | 3.83e-06 |
> | 3 | 8.16e-06 | 1.83e-06 | 1.73e-06 | 3.86e-06 | 3.50e-06 |
> | 4 | 3.68e-06 | 1.87e-06 | 2.60e-06 | 4.32e-06 | 5.98e-06 |
> | 5 | 6.15e-06 | 3.02e-06 | 3.31e-06 | 3.78e-06 | 5.34e-06 |
>
> We will add this discussion and the above table in the revised version to clarify how the theoretical picture extends beyond the single-layer setting.
>
>
>
>
> **W2: It would be nice if the authors could provide some intuition on the necessity of Assumption 2. Some of the items in this assumption seem a bit specific, e.g. requiring non-zero coordinates in the gradient of $F_0$.**
>
>
> Answer: We thank the reviewer for highlighting this point. Assumption 2 is required only for Theorem 2.2, and its major intention is to exclude degenerate target functions for which a meaningful lower bound cannot be established. The intuition behind each component is as follows:
>
> - Assumptions (2.1) and (2.4) guarantee that each $f_i$ has a well-behaved landscape around its minimizer, which enables the geometric separation used in the analysis.
>   A canonical example is $f_i(x)=\|x-x^{(i)}\|^2$.
>   A degenerate example excluded by these assumptions is $f_i(x) \equiv c_0$ for constant $c_0$, which is totally independent of the input.
>
> - Assumption (2.2) ensures that the minimizers of $\{f_i\}$ are distinct, allowing us to partition the space into $D$ disjoint basins.
>   Since one attention head can effectively specialize to at most one basin, having $h<D$ ensures that at least one basin remains “unattended,” which is crucial for constructing the adversarial sequences.
>   A degenerate example excluded by this assumption is $f_1 = f_2 = \dots = f_D$.
>
> - Assumption (2.3) requires non-zero gradient near each minimizer so that perturbations in the input lead to changes in the target value, ruling out trivial flat regions where no separation can be established (such as $F_0 \equiv c_0$ for constant $c_0$).
>   A canonical example of $F_0$ is $F_0(z_1, \dots, z_D) = \sum_{i=1}^D z_i$.
>
> We will clarify this intuition in the revision to make the role of Assumption 2 more transparent.

---

> ### Author Response · Authors · 2025-11-20
> **Reply to Weakness (Part 2)**
>
> **W3: It would also be nice if the authors could include a brief intuition of the proof strategy, especially the strategy for proving the lower bound, in the main text.**
>
> Answer: We appreciate the reviewer’s suggestion. We will include a concise intuition of the lower bound proof in the main text. In brief, the strategy consists of four steps:
> - **(i)** Each attention head tends to focus on one of the $D$ disjoint minima basins $B(x^{(i)}, r)$ around $x^{(i)}$.
>
> - **(ii)** When $h < D$, at least one basin receives no attention; we select a monotone segment $G_i \subset B(x^{(i)}, r)$ (assume $i=1$ without loss of generality).
>
> - **(iii)** Along $G_1$, we construct many candidate subsequences and apply a pigeonhole argument to obtain two subsequences $Z_1$ and $Z_2$ that induce nearly identical attention outputs but different values of $f_1$.
>
> - **(iv)** We extend $Z_1, Z_2$ into full sequences $W_1, W_2$ such that the attention layer maps them within $O(\epsilon^{k+1})$ distance of each other, while the target function separates them by at least $3\epsilon$.
>   By Lemma 4, the feed-forward network must therefore contribute a large portion of the approximation power, yielding the lower bound on its size.
>
> We will incorporate this intuition in the revised manuscript to make the core idea of the proof more accessible.

---

> ### Author Response · Authors · 2025-11-20
> **Reply to Questions (Part 1)**
>
> **Q1: There seems to be a sudden transition in between cases 1 and 2 of Theorem 2, i.e., even with $h = D-1$ one has a lower bound $M \ge 1/\varepsilon^{\Omega(T/(nD))}$, while with $h = D$ we suddenly get $M \le 1/\varepsilon^\gamma$. Is there an intuition on this sudden change?**
>
> Answer: Thank you for the thoughtful question. The sudden change is rooted in the role of attention heads in allocating representational capacity across the $D$ minima basins of the target function. When $h = D$, each head can specialize to one distinct basin $B(x^{(i)}, r)$, enabling the model to preserve the necessary information locally within the attention block. In contrast, when $h < D$, at least one basin remains unattended. The information from that basin must then be compressed through the shared embedding dimension $E = n h$, forcing the feed-forward block to reconstruct fine-grained variations from a significantly reduced representation. This compression bottleneck leads to a fundamentally worse approximation rate and produces the sharp transition observed between the cases $h < D$ and $h \ge D$.
>
>
> **Q2: Similarly, what is the reason for the difference between $M \le 1/\varepsilon^{\gamma}$ for Case 1 and $M \le 1/\varepsilon^{1+\gamma}$ for Case 3 in Theorem 2?**
>
> Answer: We appreciate the reviewer’s question. The distinction arises from the functional structure required in the two cases.
>
> In Case 1, the approximation relies only on Assumption 3, where both the inner functions $f_i$ and the outer aggregation $F_0$ can each be approximated up to error $\epsilon$ using $\Omega(1/\epsilon^{\gamma})$ parameters. Thus the overall bound remains $M \le 1/\epsilon^{\gamma}$.
>
> In Case 3, the model must additionally approximate a ''max-like’’ operation to correctly extract the dominant basin values. Implementing such a selection mechanism requires an extra $\Omega(1/\epsilon)$ factor in the parameter count (beyond the $\Omega(1/\epsilon^{\gamma})$ needed to approximate $f_i$ and $F_0$). Combining these contributions yields
> $M \le \frac{1}{\epsilon^{\max(1,\gamma)}}$, and we use
> $M \le \frac{1}{\epsilon^{1+\gamma}}$ for notation simplicity.
>
> We will add a brief remark after Theorem 2 in the revised manuscript to clarify this choice of exponent.

---

> ### Author Response · Authors · 2025-11-20
> **Reply to Questions (Part 2)**
>
> **Q3: A relevant work that could be discussed is [1]. The authors there consider a sparse sequence-to-sequence task, and in one of their results they separate the approximation power of a single-head and multi-head attention in Transformers. The task studied in [1] could also be of interest here, as it allows moving from sequence-to-vector mappings to sequence-to-sequence ones where the length of the output sequence can grow with the input.**
>
> Answer: We thank the reviewer for pointing out this very relevant work. We will add a discussion of [1] in the revised manuscript. Their $q$-sparse token regression model is a sparse sequence-to-sequence retrieval task, where the output is a length-$N$ sequence and each position depends only on $q$ relevant input tokens indicated in the prompt. This can be viewed as a natural sequence-to-sequence analogue of our sequence-to-vector “generalized retrieval’’ formulation, allowing the output length to grow with the input.
>
> In contrast to our focus on approximation rates and intrinsic dimension, [1] mainly studies sample complexity and shows that a single-layer Transformer with at least $q$ heads can learn with sample complexity almost independent of the sequence length, while RNNs and feedforward networks require sample complexity polynomial in $N$. They also prove an approximation lower bound showing that at least $q$ heads are needed in their setting, which is conceptually aligned with our head-count threshold results. We see these statistical results as complementary to our approximation-theoretic guarantees, and extending our analysis to $q$STR-type sequence-to-sequence targets is an interesting direction for future work.
>
> _Reference:_  [1] A. Mousavi-Hosseini et al., “When Do Transformers Outperform Feedforward and Recurrent Networks? A Statistical Perspective”. NeurIPS 2025.
>
>
>
>
> **Q4: It would be nice if $C_{d, D, T}$ could be made explicit, at least in terms of $T$, since long inputs are of particular interest here.**
>
> Answer: Thank you for the suggestion. We can make the dependence on $T$ explicit. In particular, the constant appearing in Theorem 2.2 can be written as
>
> $C_{d,D,T} = C_{d,D}(rT)^{-\alpha T}$,
>
> where $r>0$ is determined by the local behavior of each $f_i$ around its minimizer $x^{(i)}$ and by $\nabla F_0(z_1,\dots,z_D)$, and is therefore independent of $T$. The exponent $\alpha>0$ depends only on the dimension $d$ and the intrinsic dimension $D$. This form comes from tracking the constants in the proof of Theorem 2.2 in the appendix, where the construction introduces an $(rT)^{-\alpha}$ factor when applying the Pigeonhole theorem on $O(T)$ terms.
>
> Regarding the regime of large $T$, our results are stated under the condition $d, D \ll T \ll 1/\epsilon$. The expression above should therefore be interpreted for $T$ in this regime: for each fixed accuracy level $\epsilon$, the bound remains meaningful as long as $T$ is not taken beyond the scale $T \ll 1/\epsilon$.
>
> We will add a brief remark in the revised manuscript to clarify both the origin of $C_{d,D,T}$ in the proof and how to interpret the dependence on $T$.

---

### Official Review · Reviewer_FaEJ · 2025-11-03

**Soundness:** 3
**Presentation:** 2
**Contribution:** 3
**Rating:** 6
**Confidence:** 4

**Summary:**

This paper provides a theoretical analysis of the expressivity and approximation efficiency of single-layer Transformer networks, focusing specifically on the role of the number of attention heads, $h$. It introduces a function class, the generalized $D$-retrieval task, which models problems that combine multiple sequence elements determined by minimizing different component functions. The authors establish the generality of this task in Theorem 1 by proving that the class is dense in all continuous sequence-to-vector mappings.

The core of the paper is then captured in Theorem 2, which provides a set of sharp approximation bounds:
* Positive Result (Thm 2.1): A transformer with $h=D$ heads can efficiently ($\epsilon$-approximation in $L_\infty$ norm) approximate $D$-retrieval tasks with constant per-head embedding dimension ($n=2$) and MLP size $M$ scaling independently of sequence length $T$.
* Negative Result (Thm 2.2): If the number of heads $h$ is insufficient ($h<D$), and the sequence length $T$ is much larger than the per-head embedding dimension $n$, the MLP size $M$ must grow exponentially in $T$.
* Positive Result (Thm 2.3): If the embedding dimension $n$ is scales linearly with the $T$, the full sequence can be effectively "memorized" in the attention output, and the task can be completed efficiently without an exponentially large FFN.

The theoretical findings are validated with experiments. The synthetic task clearly demonstrates a sharp phase transition in learning difficulty at $h=D=4$. Experiments on simple image and text retrieval tasks show that performance increases as the number of heads (with fixed parameters per head) increases.

**Strengths:**

The work provides a fundamental theoretical basis that cleanly captures the trade-offs on the expressive power of attention heads. The model is deliberately structured to reveal that when $h<D$, heads are forced to encode multiple retrieval roles simultaneously, creating an information bottleneck that MLP can only overcome with exponential complexity in $T$. Taken together, the bounds demonstrate sharp trade-offs between difference scaling regimes depending on head and embedding dimension capacity. The use of a simple, illustrative toy example greatly aids in explaining the bounds.

A key achievement is how the work uses the concept of intrinsic dimension $D$ to distinguish performance at $h$ vs. $h+1$ heads. This analysis moves beyond earlier approximation results that either focused on universal capacity (where one large head is enough) or analyzed rank restrictions in isolation. The demonstration that $h<D$ leads to a $T$-dependent performance scaling is a useful distinction.

The synthetic experiments offer strong validation. They demonstrate a real threshold in model performance exactly at the theoretical intrinsic dimension D=4. This empirical finding strongly suggests that the theoretical representational gap translates directly into a learning difficulty gap in practice.

**Weaknesses:**

The theoretical section lists model assumptions (Assumption 1) and target function constraints (Assumption 2) that appear to apply to the entire set of theorems (Thm 2.1, 2.2, 2.3). While it is appreciated that the bounds are in a consistent setting, it would improve the presentation and clarity of the paper if the authors would specify the minimal subset of assumptions required for each individual bound. For example, does Theorem 2.1 strictly require the Hessian constraints of Assumption 2.3? This small clarification would make the theoretical scope of each result easier to grasp.

The derivation of the lower bound in Theorem 2.2 relies on Lemma 4, which bounds all weights in the MLP by 1 (Assumption 1.3). This weight constraint simplifies the analysis, but is (to my knowledge) nonstandard in theoretical work or practical models. It would be valuable to discuss whether the lower bound can be maintained by instead considering a joint bound on the network's width and its weight norm in the lower bound, instead of a strict magnitude bound on every entry. This would align the result more closely with well-known complexity bounds for feed-forward networks (e.g., those from the line of work including [Yehudai-Shamir '19](https://arxiv.org/abs/1904.00687)).

While the proofs appear largely correct upon detailed inspection, the proof of Theorem 2.2 is dense and hard to follow, involving numerous variables and recursive definitions that must be kept in the reader's head. I recommend the authors consider adding visual aids or diagrams that illustrate the construction of the adversarial sequences $W_1$ and $W_2$ and the partitioning of weights across the sequence indices. More descriptive notation would also help.

The experiments on real tasks show that increasing the number of heads $h$ (with a constant embedding dimension $n$ per head) helps performance. However, this is a key confounder, as the total embedding dimension $E=nh$ increases linearly with $h$, and thus the raw parameter count of the attention layers increases. Given the theoretical claim is about the specialization enabled by multiple heads, is it possible to design an experiment to disentangle the effect of the number of heads $h$ from the total parameter count?

**Questions:**

Why did the synthetic experiments focus exclusively on the case where $D=4$ (4-retrieval task)? It would be valuable to know how the critical performance threshold changes as a function of $D$. Did preliminary experiments suggest the sharpness of the transition is invariant to $D$?

The restriction to single-layer transformers is a noted limitation. While a rigorous multi-layer theory is beyond the scope of this paper, it would be highly insightful to share a conjecture or intuition about what is expected in that regime. Is the intrinsic dimension D of a task expected to be distributed across layers, or does the bottleneck effect remain a strong limiting factor at every layer? Insights based on the real-world experiments, which use deeper architectures, would be helpful to include in the discussion.

---

> ### Author Response · Authors · 2025-11-20
> **Reply to Weakness (part 1)**
>
> **W1: The theoretical section lists model assumptions (Assumption 1) and target function constraints (Assumption 2) that appear to apply to the entire set of theorems (Thm 2.1, 2.2, 2.3). While it is appreciated that the bounds are in a consistent setting, it would improve the presentation and clarity of the paper if the authors would specify the minimal subset of assumptions required for each individual bound. For example, does Theorem 2.1 strictly require the Hessian constraints of Assumption 2.3? This small clarification would make the theoretical scope of each result easier to grasp.**
>
> Answer: We thank the reviewer for this suggestion and will clarify in the revised version which assumptions are required for each theorem. Specifically:
>
>     Theorems 2.1 and 2.3 (the upper bound) do not require Assumption 1 (model constraints) or Assumption 2 (target constraints). These assumptions are only used for lower bound-type statements and can be dropped without affecting the validity of the results. Both theorems rely solely on Assumption 3,(A1) and (A2), which specify the approximation properties of the target.
>
>     Theorem 2.2 (the lower bound) does require all conditions in Assumption 1 and Assumption 2, since both the model structure and the non-degenerate properties of the target class are essential for the lower bound. Assumption 3 is not used in the proof of Theorem 2.2.
>
> **W2: The derivation of the lower bound in Theorem 2.2 relies on Lemma 4, which bounds all weights in the MLP by 1 (Assumption 1.3). This weight constraint simplifies the analysis, but is (to my knowledge) nonstandard in theoretical work or practical models. It would be valuable to discuss whether the lower bound can be maintained by instead considering a joint bound on the network’s width and its weight norm in the lower bound, instead of a strict magnitude bound on every entry. This would align the result more closely with well-known complexity bounds for feed-forward networks (e.g., those from the line of work including Yehudai-Shamir ’19).**
>
> Answer: The lower bound can still be maintained in this setting. Here we restate Theorem 2.2 in a more general way, assuming $M_0$ to be the max weight norm of the feedforward block $\hat{F}$ and $l$ be the number of hidden layers of $\hat{F}$, $w$ be the width of $\hat{F}$. Then for $h=s<D$, define $k=\frac{(\frac{1}{4} T-s -D +1)}{l[(n+1)s+1]}-1$, then to achieve $\epsilon$-approximation, we have $w = \Omega\left(\frac{1}{M_0^2\epsilon^k}\right)$.
>
> We note that the more general formulation involving the maximum weight norm $M_0$ and the number of hidden layers $l$ is presented here as a conjecture, it is consistent with our current proof framework (when $M_0 \le 1$ ans $l=1$), but not formally proved. Therefore, we do not intend to modify the statement of Theorem 2.2 in the revised manuscript. The general version is included here only to clarify how our result is expected to relate to standard width–norm complexity measures.
>
>
> Relation to Yehudai--Shamir ’19 :  The line of work initiated by Yehudai and Shamir studies expressivity gaps between neural networks and restricted feature-based models, such as random-feature networks. Their results show that learning certain simple target functions requires exponentially many random features unless the model allows very large weight scaling. The hardness comes from margin/geometry arguments on the representation power of such feature maps. Our result is different in its mechanism: the feedforward neural network needs to either have large enough parameters or be wide enough to approximate functions with extremely large Lipschitz constant.
>
> We will incorporate this  discussion of network expressiveness into the revised manuscript to better connect with existing theoretical literature.

---

> ### Author Response · Authors · 2025-11-20
> **Reply to Weakness (Part 2)**
>
> **W3: While the proofs appear largely correct upon detailed inspection, the proof of Theorem 2.2 is dense and hard to follow, involving numerous variables and recursive definitions that must be kept in the reader’s head. I recommend the authors consider adding visual aids or diagrams that illustrate the construction of the adversarial sequences $W_1$ and $W_2$ and the partitioning of weights across the sequence indices. More descriptive notation would also help.**
>
> Answer: We appreciate the reviewer’s careful reading of Theorem 2.2 and the suggestion to improve its readability. In the revision, we will make the structure and intuition of the proof more transparent without changing the underlying argument. Concretely:
>
> High-level roadmap (Proof sketch): At the beginning of the proof, we will add a brief overview that explains the main steps in words: (i) each head selects its most responsive locations $(y_j,t_j)$ on the sets $K_i$, which is in the minima basin around $x^{(i)}$; (ii) because $s<D$, there is at least one segment $G_i \subset K_i$ that no head focuses on, we then consider the segment $G_i$ in it; (iii) along this segment (suppose it is $G_1$), we construct many candidate subsequences and, by a pigeonhole argument, obtain two subsequences $Z_1,Z_2$ whose post-attention representations are almost identical but whose contribution to $f_1$ differs; (iv) these subsequences are then embedded into full sequences $W_1,W_2$, which the target function separates by at least $3\epsilon$, while the attention block maps them within $O(\epsilon^{k+1})$, forcing a large feed-forward network by Lemma 4.
>
> We also create a notation table to help with understanding.
> | **Notation flow (dependency structure)** | **Meaning** |
> |------------------------------------------|-------------|
> | $x^{(i)}$ | Point where $f_i$ achieves minimum |
> | $\rightarrow B(x^{(i)},r)$ | Basin region for retrieval coordinate $i$ (Basin around $x^{(i)}$) |
> | $\rightarrow \rightarrow G_i, K_i$ | Monotone local segment near $x^{(i)}$ (In $B(x^{(i)},r)$) |
> | $P_0$ | The set of all candidate points. (We only choose $x_t \in P_0$) |
> | $S_i$ | Index partition for retrieval coordinate $i$ ($i=1, \dots, D$) |
> | Attention head $j$ | Defines response at position $t$ |
> | $\rightarrow \lambda_j(x,t)$ | Attention score |
> | $\rightarrow \rightarrow (y_j,t_j)$ | Maximum‐attention point selected by head $j$ in $P_0\times S_j$ |
> | $\rightarrow \rightarrow Y=\{y_1,\dots,y_s\}$ | Chosen maximizers of attention score (one per head) |
> | $\rightarrow v_j(x,t)$ | Value embedding |
> | WLOG, suppose $K_1 \cap Y = \emptyset$. |  |
> | $\rightarrow T_0$ | $T_0 \subset S_1$, indices not in $(y_j, t_j)$, $j=1, \dots, s$ |
> | $\rightarrow \eta:[0,1]\to G_1$ | Coordinate system on $G_1$ |
> | $\rightarrow q=f_1\circ\eta$ | Rewriting $f_1$ into the coordinate system. |
> | $\rightarrow U_t$ | Discrete grid on $[0,1]$ at index $t$ |
> | $\rightarrow z_\ell(t)$ | Candidate point for subsequence $\ell$, $z_\ell(t) \in \eta(U_t)$ |
> | Adversarial subsequences and extension to full sequences |  |
> | $\rightarrow Z_\ell = (z_\ell(1),\dots,z_\ell(T_0))$ | Two subsequences almost indistinguishable by attention head. |
> | $\rightarrow W_\ell$ | Full sequence embedding $Z_\ell$ |
> | $\rightarrow w_\ell(t)$ | Token of $W_\ell$ of index $t$ |
> | $\rightarrow I_1,I_2,I_3$ | Partition of indices: differ / agree / remaining |
> | Per-head aggregate statistics for analysis |  |
> | $\rightarrow Q_{j,i}$ | Attention mass on $I_j$ ($j\in\{1,2,3\}, i\in \{1, \dots, s\}$) |
> | $\rightarrow V_{j,i}$ | Weighted value average on $I_j$ |

---

> ### Author Response · Authors · 2025-11-20
> **Reply to Weakness (Part 3)**
>
> **W4: The experiments on real tasks show that increasing the number of heads $h$ (with a constant embedding dimension $n$ per head) helps performance. However, this is a key confounder, as the total embedding dimension $E = n h$ increases linearly with $h$, and thus the raw parameter count of the attention layers increases. Given the theoretical claim is about the specialization enabled by multiple heads, is it possible to design an experiment to disentangle the effect of the number of heads $h$ from the total parameter count?**
>
> Answer: We thank the reviewer for the valuable suggestion. We conducted an additional set of experiments on the synthetic dataset in Section 6.1, where we fix the total embedding dimension $E = nh$ and vary the number of heads $h$, using the same kind of single-layer transformer setting. The results below demonstrate a consistent trend. (For $h=3,5$, we choose per-head embedding dimension to be $\lceil 32/h\rceil$, and total embedding dimension becomes $E=33, 35$. )
>
> **Validation NMSE under fixed total embedding dimension $E = nh = 32$**
>
> | **Number of Heads $h$** | **$T=8$** | **$T=16$** | **$T=32$** | **$T=64$** | **$T=128$** |
> |------------------------:|:---------:|:----------:|:----------:|:----------:|:-----------:|
> | 1 | 1.75e-02 | 1.98e-02 | 2.06e-02 | 2.54e-02 | 3.03e-02 |
> | 2 | 7.17e-03 | 7.39e-03 | 7.82e-03 | 8.57e-03 | 1.02e-02 |
> | 3 | 2.11e-04 | 2.17e-04 | 2.73e-04 | 3.71e-04 | 4.77e-04 |
> | 4 | 1.32e-06 | 5.59e-07 | 3.40e-07 | 3.46e-07 | 5.70e-07 |
> | 5 | 2.19e-06 | 4.33e-07 | 3.22e-07 | 2.73e-07 | 2.66e-07 |

---

> ### Author Response · Authors · 2025-11-20
> **Reply to Questions**
>
> **Q1: Why did the synthetic experiments focus exclusively on the case where $D = 4$ (4-retrieval task)? It would be valuable to know how the critical performance threshold changes as a function of $D$. Did preliminary experiments suggest the sharpness of the transition is invariant to $D$?**
>
> Answer: We appreciate the reviewer’s insightful suggestion. Our initial choice of $D=4$ was based on its ability to clearly highlight the transition predicted by the theory while maintaining a reasonable computational cost. Following the reviewer’s recommendation, we additionally conducted experiments on the case $D=3$. The results exhibit the same qualitative behavior.
>
> The corresponding results for $D=3$ are presented below. (We also keep the $E=nh$ constant as above in Weakness 4)
>
> **Approximation error for the $D = 3$ retrieval task with fixed total embedding dimension $E = nh$**
>
> | **Number of heads $h$** | **$T=8$** | **$T=16$** | **$T=32$** | **$T=64$** | **$T=128$** |
> |------------------------:|:---------:|:----------:|:----------:|:----------:|:-----------:|
> | 1 | 1.38e-02 | 1.63e-02 | 1.84e-02 | 2.17e-02 | 2.31e-02 |
> | 2 | 1.09e-03 | 7.08e-04 | 7.24e-04 | 7.76e-04 | 1.11e-03 |
> | 3 | 4.18e-07 | 1.72e-07 | 1.17e-07 | 3.58e-07 | 2.11e-07 |
> | 4 | 5.56e-07 | 1.22e-07 | 6.89e-08 | 1.85e-07 | 3.48e-07 |
>
> **Q2: The restriction to single-layer transformers is a noted limitation. While a rigorous multi-layer theory is beyond the scope of this paper, it would be highly insightful to share a conjecture or intuition about what is expected in that regime. Is the intrinsic dimension $D$ of a task expected to be distributed across layers, or does the bottleneck effect remain a strong limiting factor at every layer? Insights based on the real-world experiments, which use deeper architectures, would be helpful to include in the discussion.**
>
> Answer: We appreciate the reviewer’s insightful question regarding the multi-layer setting. Developing a full theoretical characterization for deep transformers is an important direction of our ongoing work. While a rigorous extension is beyond the scope of this paper, we summarize here the intuition that is most consistent with our theoretical arguments and empirical observations.
>
> **Conjecture (multi-layer expressivity).**
> Consider an $l$-layer transformer with $h$ attention heads per layer, operating on a $D$-retrieval task.
> A necessary condition for efficient approximation is the head–dimension condition:
>
> $ l \cdot h \ge D$,
>
> which reflects that the intrinsic task dimension must be sufficiently “covered’’ across layers.
>
> More precisely, when $lh < D$, we conjecture the lower-bound scaling:
>
> $\log(\mathrm{ParamCount})
>     = \Omega \Bigl(|\log \epsilon| \cdot \frac{a_l \, T^{\, b_l}}{n h} \Bigr)$,
>
> for some constants $a_l, b_l > 0$ depending only on depth $l$.
>
> This formulation reduces to our proven bound in the single-layer case $(l = 1)$, and the MS MARCO experiment suggests $b_l = 0.25$ for $l = 2$.
>
>
>
> Two-layer experiment (NoPE, NoLN): To further support the above picture, we trained a 2-layer transformer on the same synthetic task without positional encoding and without layer normalization, as LN noticeably harms generalization in this toy setting experiment. As shown in the table below, increasing $h$ leads to consistent improvement, again aligning with the $lh \approx D$ threshold. (We also keep $E=nh$ constant in this experiment. )
>
> **Two-layer transformer on the synthetic task ($D = 4$, NoPE, NoLN)**
>
> | **Number of Heads $h$** | **$T=8$** | **$T=16$** | **$T=32$** | **$T=64$** | **$T=128$** |
> |------------------------:|:---------:|:----------:|:----------:|:----------:|:-----------:|
> | 1 | 2.12e-04 | 1.85e-04 | 2.22e-04 | 3.13e-04 | 4.28e-04 |
> | 2 | 7.22e-06 | 2.69e-06 | 3.50e-06 | 3.07e-06 | 3.83e-06 |
> | 3 | 8.16e-06 | 1.83e-06 | 1.73e-06 | 3.86e-06 | 3.50e-06 |
> | 4 | 3.68e-06 | 1.87e-06 | 2.60e-06 | 4.32e-06 | 5.98e-06 |
> | 5 | 6.15e-06 | 3.02e-06 | 3.31e-06 | 3.78e-06 | 5.34e-06 |
>
> We will incorporate these expanded discussions and the new experiments into the revised manuscript. We thank the reviewer once again for prompting us to explore this insightful extension.

---

### Author Response · Authors · 2025-11-25
**Revised version uploaded**

We thank all reviewers for their thoughtful and constructive feedback. We have uploaded a revised version of the manuscript with changes highlighted in blue, and we are grateful for the suggestions that helped us improve the clarity and scope of the paper.

---

> ### Author Response · Authors · 2025-11-30
> **Summary of the revisions**
>
> **Paper Summary**
>
> We study how the number of attention heads affects the approximation power of Transformers. Using a generalized $D$-retrieval task, we prove upper and lower bounds showing a head-count bottleneck (too few heads ($h < D$) require parameters scaling as $O(1/\epsilon^{cT})$), and we support these results with synthetic and real-data experiments.
>
>
> **Summary of main revisions**
>
> To address the concerns raised during the review process and to improve the clarity and scope of the manuscript, we have made the following revisions:
>
>
> ### 1. Theory
>
> #### 1.1 Theoretical setup
>
> - Clarified the intuition behind Assumption 2, showing that its role is to exclude degenerate cases (e.g., trivial or overlapping minima) while preserving the generality of the target class. This makes explicit why these are reasonable assumptions for our setting.
>
> - Added Corollary 2 (a direct corollary of Theorem 2) to show that the intrinsic dimension $D$ is unique and does not depend on the particular representation of the target via $(f_i, S_i, F_0)$. This clarifies the definition of $D$ as an intrinsic property of the task.
>
> - Connected our target class, the generalized $D$-retrieval task, to existing work such as Mousavi-Hosseini et al. (2025), highlighting how their $q$-sparse sequence-to-sequence setting is a natural analogue of our sequence-to-vector formulation.
>
>
>
>
>
>
>
>
> #### 1.2 Theorems and their intuition
>
> - Added a brief proof-strategy explanation for Theorem 2.2, outlining the construction of adversarial subsequences $Z_1, Z_2$ and the role of the head bottleneck. We also added a notation-flow summary in the appendix, placed immediately before the proof of Theorem 2.2, to make the argument easier to follow.
>
> - Clarified the dependency of Theorem 2 on the assumptions, made explicit the form of the constant $C_{d,D,T}$ in Theorem 2 with respect to $T$, and explained how the bound should be interpreted under the regime $d, D \ll T \ll 1/\epsilon$.
>
> - Compared our approach with existing work such as Yehudai & Shamir (2019), highlighting the differences in focus and techniques.
>
>
>
>
> #### 1.3 Conjectural extensions and insights
>
> - Added a short conjectural discussion of the multi-layer regime, proposing that a necessary condition for efficient approximation is $l h \ge D$ and conjecturing a corresponding scaling of $\log(\mathrm{ParamCount})$ with $h$, $T$, and depth $l$.
>
> - Added a discussion immediately before the Limitations section on potential ways to probe the intrinsic dimension of a given task, outlining how one might empirically investigate $D$ beyond fully training models.
>
>
>
>
>
> ### 2. Experiments
>
> - Added fixed-$E$ experiments that keep the total embedding dimension $E = n h$ constant while varying $h$, showing that the performance improvements persist even when parameter count is controlled.
>
> - Added new synthetic experiments for $D = 3$, confirming that the sharp transition in approximation quality with respect to $h$ is not specific to $D = 4$.
>
> - Included two-layer synthetic experiments (NoPE, NoLN) on the $D = 4$ task, revealing an $l h \ge D$ threshold behavior that supports our conjecture in the multi-layer case.
>
> - Clarified the reason for using different evaluation metrics: NMSE for synthetic tasks and accuracy for real-world tasks, and reported test accuracies for the real-world tasks for completeness.
>
>
> ### 3. Presentation and clarity
>
> - Harmonized notation in the experimental section, consistently using lowercase $h$ for the number of heads.
>
> - In the task setup (Mathematical Formulation), restated the sentence explaining how to extend from scalar-valued targets to vector-valued targets, making the construction clearer to follow.
>
> - In addition, we made several minor editorial and formatting improvements throughout the main text and appendix, which we omit here for brevity.
>
> **Concluding remarks**
>
> We thank the reviewers for their careful reading and constructive feedback. The revised manuscript incorporates the changes summarized above. We hope that these revisions and clarifications adequately address the concerns raised during the review process and provide a clearer picture of how the attention head count $h$ and the intrinsic dimension $D$ of sequence tasks interact  to determine approximation behavior.

---

### Meta-Review · Area_Chair_Xk6v · 2026-01-08

**Summary:**

The paper provides a novel analysis of Transformer expressive power and the role of number of heads. All reviewers find the paper's contributions meaningful and interesting. Reviewers raised several clarification questions about the proof structure and details, authors addressed it in their response. Overall I think the paper gives a clean setting to study the role of number of heads in attention layers and I am happy to suggest acceptance.

**Reviewer Concerns:**

Main concerns were around proof structure and clarity of the manuscript.

**Reviewer Scores:**

N/A

---

### Decision · Program_Chairs · 2026-01-26

Accept (Poster)